# Oncoprotein SET-associated transcription factor ZBTB11 triggers lung cancer metastasis

Wenbin Xu[1,2], Han Yao[1,2], Zhen Wu[1], Xiaojun Yan[1], Zishan Jiao[1], Yajing Liu[1], Meng Zhang[1] & Donglai Wang [1] ✉

Metastasis is the major cause of lung cancer-related death, but the mechanisms governing lung tumor metastasis remain incompletely elucidated. SE translocation (SET) is overexpressed in lung tumors and correlates with unfavorable prognosis. Here we uncover SET-associated transcription factor, zinc finger and BTB domain-containing protein 11 (ZBTB11), as a prometastatic regulator in lung tumors. SET interacts and collaborates with ZBTB11 to promote lung cancer cell migration and invasion, primarily through SET-ZBTB11 complex-mediated transcriptional activation of matrix metalloproteinase-9 (MMP9). Additionally, by transcriptional repression of proline-rich Gla protein 2 (PRRG2), ZBTB11 links Yes-associated protein 1 (YAP1) activation to drive lung tumor metastasis independently of SET-ZBTB11 complex. Loss of ZBTB11 suppresses distal metastasis in a lung tumor mouse model. Overexpression of ZBTB11 is recapitulated in human metastatic lung tumors and correlates with diminished survival. Our study demonstrates ZBTB11 as a key metastatic regulator and reveals diverse mechanisms by which ZBTB11 modulates lung tumor metastasis.

The latest global cancer statistics reveal that lung cancer remains the leading cause of cancer death, although its incidence has slightly declined into the second[1]. Lung cancer can be histologically categorized into small cell lung cancer (SCLC) and non-small cell lung cancer (NSCLC), the latter of which can be further divided into adenocarcinoma (LUAD), squamous cell carcinoma (LUSC), large cell carcinoma and others[2]. Among all types of lung cancer, more than 40% are adenocarcinomas, and metastasis, which represents a hallmark of an advanced stage of malignancy, serves as the major cause of lung cancer-associated death in patients[2–4]. However, the regulatory network of lung cancer metastasis remains incompletely understood.

The oncoprotein SET was originally identified in a study of acute undifferentiated leukemia (AUL), where a novel oncogenic *SET-CAN* fusion gene was discovered upon investigating DNA breakpoints caused by chromosomal translocation[5]. Complete loss of SET expression in a conditional knockout mouse model reveals an embryonically lethal phenotype, demonstrating a pivotal role of SET in regulating normal development[6,7]. More strikingly, SET overexpression is a frequent event among different types of cancer including lung carcinoma[8–12], suggesting that the aberrant expression of SET in adult cells contributes to tumor initiation and/or progression. Indeed, the following evidence indicates that the high levels of SET in NSCLC are markedly correlated with the progression of clinical stages, lymph node metastasis, and, consequently, a poor survival rate[13]. As a multifaceted protein, SET can be distributed in both the cytosol and nucleus and physically interact with diverse key factors to promote tumor-related behaviors of the cell[14,15]. For example, cytosolic SET primarily binds the catalytic subunit of serine/threonine-protein phosphatase 2 A (PP2A) and inhibits PP2A enzymatic activity and tumor suppressive actions, by which SET promotes cell proliferation, anti-apoptosis and metastasis[16].

[1]State Key Laboratory of Common Mechanism Research for Major Diseases & Department of Medical Genetics, Institute of Basic Medical Sciences & School of Basic Medicine, Chinese Academy of Medical Sciences & Peking Union Medical College, Beijing 100005, China. [2]These authors contributed equally: Wenbin Xu, Han Yao. ✉e-mail: dwang@ibms.pumc.edu.cn

In addition, nuclear SET mainly serves as a chaperone that intimately associates with histones for nucleosome assembly or acts as a cofactor that binds transcription factors (TFs) for gene-specific transcriptional regulation[6,17,18]. Thus, targeting aberrant SET by individual or combined strategies may trigger both cytoplasm- and nucleus-involved mechanisms for tumor, including lung carcinoma, intervention[12,19]. However, the major TFs within the nucleus dictating SET for metastatic regulation in lung cancer are still obscure.

ZBTB11 belongs to an evolutionarily conserved ZBTB protein family that consists of approximately 60 unique members in vertebrate cells[20]. Similar to the majority of the family members, ZBTB11 is characterized as harboring an N-terminal BTB domain and tandem C-terminal zinc finger motifs that are responsible for mediating protein–protein interactions and DNA binding, respectively[21,22]. Based on such structural features, ZBTB11 was considered a putative transcription factor[23], and this notion was recently validated by integrating ChIP-seq and RNA-seq datasets in studies of both human fibroblasts and mouse embryonic stem cells (mESCs)[24,25]. Under physiological conditions, ZBTB11 contributes to neutrophil development and emergency granulopoiesis through a Pu.1-ZBTB11-p53 regulatory pathway[26]. In addition, ZBTB11 may also coordinate with zinc finger protein 131 (ZFP131) to maintain the pluripotency of ESCs by preventing aberrant expression of pro-differentiation genes[27]. More importantly, dysfunction of ZBTB11 has emerged as a key event during disease onset. For example, genetic evidence from analyzing consanguineous families indicated that biallelic missense mutations of *ZBTB11* in its zinc finger motifs were critically involved in the pathogenesis of autosomal recessive intellectual disability (ID)[25,28,29]. In addition, ZBTB11 was spatially enriched within the sites of active fibro-fatty replacement of myocardium, which is involved in cardiomyopathy progression[30]. Furthermore, the reduced expression of ZBTB11 by mi-548j upon chronic hepatitis B (CHB) infection was related to the blunted antiviral effect by impairing type I interferon production[31]. Downregulation of ZBTB11 was also observed in hepatocellular carcinoma[32], whereas overexpression of ZBTB11 was reported in bladder cancer[33] and different types of leukemia[34,35]. More recently, ZBTB11 was reported to participate in the long noncoding RNA (lncRNA) DUBR-mediated migration and invasion of lung cancer cells, suggesting a potential role of ZBTB11 during lung cancer progression[36]. However, the precise transcriptional profiles, regulatory networks, and biological consequences controlled by ZBTB11 in lung cancer remain largely unknown.

In this study, we identify ZBTB11 as a binding partner of the oncoprotein SET in the nucleus. Integrated analyses of ChIP-seq and RNA-seq reveal that ZBTB11 acts as a transcription factor in lung cancer cells. SET cooperates with ZBTB11 in transcriptional regulation by acting as a cofactor, and the interplay between SET and ZBTB11 is critically involved in modulating the metastatic behaviors of lung cancer cells. Mechanistically, we uncover MMP9 as a primary downstream effector that contributes to SET-ZBTB11 complex-mediated regulation of metastasis. In addition, ZBTB11 exhibits a mode of action through the transcriptional repression of PRRG2, establishing a connection to the activation of YAP1 that promotes lung tumor metastasis in a SET-ZBTB11 complex-independent manner. More importantly, conditional knockout of ZBTB11 dramatically inhibits distal metastasis of the primary lung tumors induced by genetic lesions in vivo. Taken together, our study demonstrates that the SET-binding partner ZBTB11 plays critical roles in modulating lung tumor metastasis.

## Results

### Identification of ZBTB11 as a binding partner of the oncoprotein SET

To identify the proteins that functionally interact with the oncoprotein SET, we generated a stable cell line that constitutively expresses ectopic Flag-HA double-tagged SET in p53-null H1299 lung cancer cells. The SET-containing protein complex in the nuclear fraction was obtained by Flag-HA tandem purification and visualized by silver staining (Fig. 1a). Mass spectrometry (MS) analysis revealed 66 proteins from the control purification (H1299-EV) and 225 proteins from the H1299-FH-SET purification (Supplementary Fig. 1a and Supplementary Data 1). After excluding 36 proteins shared by these two purifications, we focused on the remaining 189 proteins that represent potential SET-specific binding partners. Of note, transcription intermediary factor 1-beta (TRIM28, also known as KAP1), a validated SET direct binding protein in the nucleus[37], was ranked No. 1 among the identified proteins (Supplementary Fig. 1b), suggesting that our MS results can reflect the cellular profiles of SET-binding proteins. Since we particularly focused on SET-mediated transcriptional regulation, we noticed that ZBTB11, a putative transcription factor, was enriched as a major SET-binding protein with top rank No. 10 (Fig. 1a and Supplementary Fig. 1b), even ranking higher than CBX3 (No. 30) and histone H1 (No. 35), two other known binding proteins of SET[38,39] (Supplementary Data 1).

The existence of ZBTB11 in the SET-containing protein complex purified from H1299-FH-SET stable cells was further confirmed by coimmunoprecipitation (Co-IP) with a ZBTB11-specific antibody (Fig. 1b). In addition, a reciprocal Co-IP assay also validated the formation of the SET-ZBTB11 complex in HEK293T cells expressing ectopic SET and/or ZBTB11 (Fig. 1c). Moreover, we successfully detected the interaction between endogenous SET and ZBTB11 in H1299 cells (Fig. 1d, e), indicating that the SET-ZBTB11 protein complex can form under physiological conditions in cancer cells. Notably, the presence of chromatin or DNA was dispensable for the interaction between SET and ZBTB11, since digestion of DNA by benzonase still maintained the cellular endogenous SET-ZBTB11 interaction (Supplementary Fig. 1c). Cellular fractionation showed that ZBTB11 was primarily localized in the nucleus, while SET was distributed in both the nucleus and cytoplasm (Fig. 1f). As expected, the colocalization of SET and ZBTB11 was mainly observed in the nucleus (Fig. 1g), suggesting a potential role of the SET-ZBTB11 complex in regulating nucleus-based biological processes. To further evaluate the binding properties of the SET-ZBTB11 complex, we performed an in vitro pull-down assay with purified proteins and revealed a direct interaction between SET and ZBTB11 (Fig. 1h). Structurally, SET can be divided into the N-terminal dimerization domain (DD), the middle earmuff domain (ED) and the C-terminal acidic domain (AD)[40]. As shown in Fig. 1i, AD was essentially responsible for binding with ZBTB11, as the loss of AD completely abolished the SET-ZBTB11 interaction. On the other hand, we also mapped the linker region (amino acids 313–568) of ZBTB11 as the primary region that was critical for interacting with SET (Fig. 1j). Taken together, our data characterize ZBTB11 as a binding partner of the oncoprotein SET in lung cancer cells.

Next, we investigated why the AD of SET specifically interacts with the linker region of ZBTB11. Due to the nature of enrichment with aspartic acid (D) and glutamic acid (E), the AD of SET exhibits a highly negative charge, which, in turn, enables mediation of the protein–protein interactions by recognizing the lysine/arginine (K/R)-rich regions that harbor a positive charge[6,41]. Since we noticed that the 522-561 aa of ZBTB11 is a K/R-rich region (Supplementary Fig. 1d), we then divided the linker region into two fragments (313–521 aa and 522–568 aa) and evaluated the binding affinity of each fragment to SET. As shown in Supplementary Fig. 1e, the fragment containing 522–568 aa was necessary and sufficient for ZBTB11 binding with SET, indicating that the K/R-rich region is indeed critically involved in the ZBTB11-SET interaction. To further evaluate whether the positive charge is required for K/R-rich region binding with SET, we expressed mutant K/R-rich regions (Mut) where all lysine and arginine residues were replaced with alanine residues (A) to mimic a charge-neutralized status (Supplementary Fig. 1d). As shown in Supplementary Fig. 1f, the charge-neutralized mutant fragment completely lost the ability to bind with SET, confirming that the maintenance of the positive charge within the

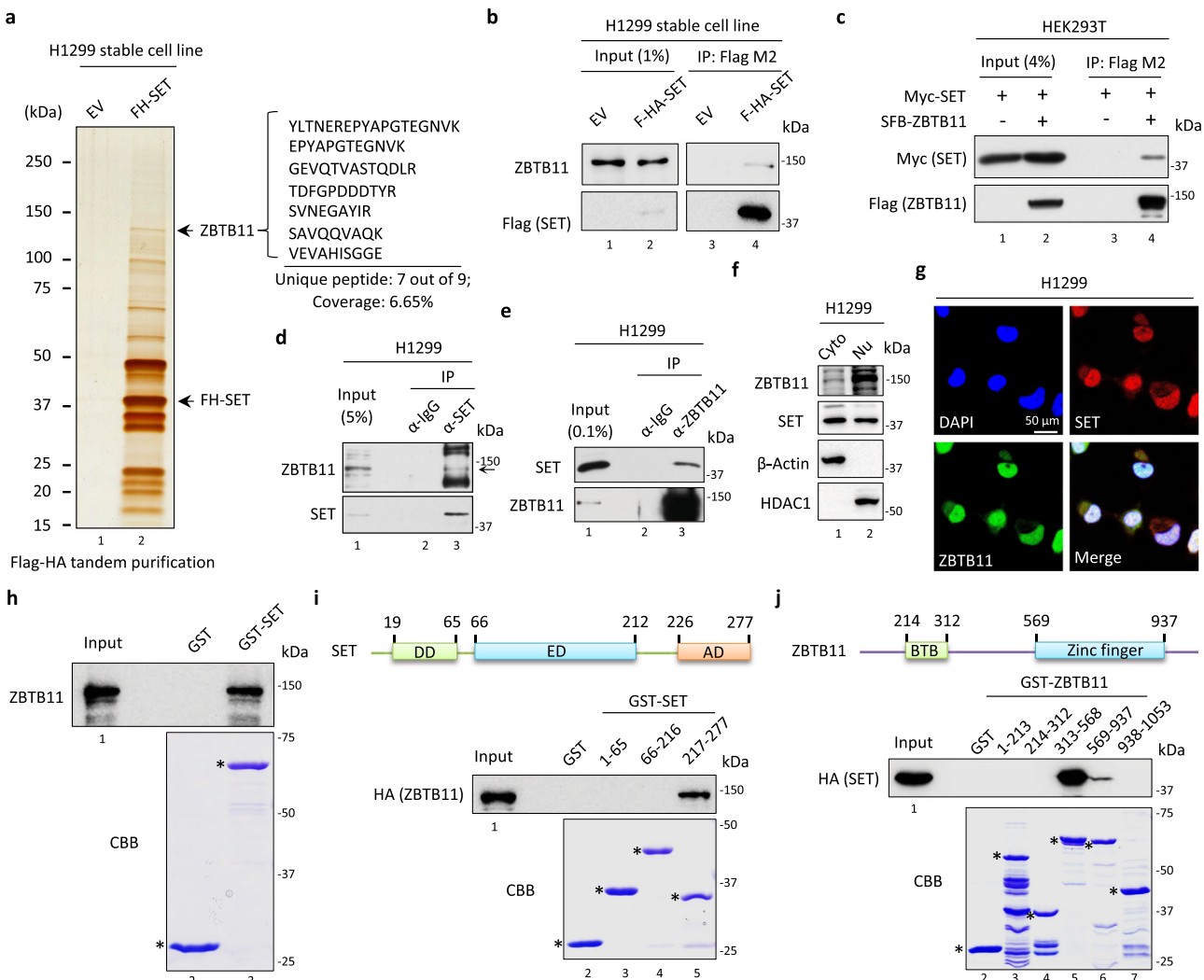

**Fig. 1 | Identification of the SET-ZBTB11 protein complex. a** Silver staining and mass spectrometry (MS) analysis of the protein complex purified from control- or SET-overexpressing stable H1299 cells identified ZBTB11. An empty vector (EV) or a Flag-HA-tagged SET (FH-SET)-expressing construct was stably transfected into H1299 cells, and the protein complex from the nuclear fraction of the indicated cells was tandemly purified by immobilized anti-Flag and anti-HA agarose. 7 unique out of 9 peptides corresponding to ZBTB11 were identified from the FH-SET-containing protein complex. **b** Coimmunoprecipitation (Co-IP)-Western blot (WB) analysis of the SET-ZBTB11 complex in H1299-EV or H1299-FH-SET cells purified by anti-Flag agarose. **c** Co-IP-WB analysis of the SET-ZBTB11 interaction in

HEK293T cells transiently transfected with Myc-tagged SET (Myc-SET) with or without S protein-Flag-streptavidin-binding protein-tagged ZBTB11 (SFB-ZBTB11). Co-IP-WB analysis of the interaction between endogenous SET and ZBTB11 in H1299 cells by anti-SET (**d**) or anti-ZBTB11 (**e**) antibody. **f** WB analysis of ZBTB11 and SET in the cytoplasmic or nuclear fraction of H1299 cells. **g** Immunofluorescence assay of endogenous SET and ZBTB11 in H1299 cells. DAPI was used to counterstain the nucleus. **h** In vitro pull-down analysis of the direct binding between purified full-length SET and ZBTB11. In vitro pull-down analysis of the domain(s) of SET (**i**) or ZBTB11 (**j**) responsible for mediating their physical interaction. Source data are provided as a Source Data file. * indicates GST or GST-fusion protein.

K/R-rich region is critical for the interaction of ZBTB11 with SET. Our data reveal the "charge effect" as a physical basis for mediating the SET-ZBTB11 interaction, where the negative charge within the acidic domain of SET attracts the positive charge within the K/R-rich region of ZBTB11 (Supplementary Fig. 1g).

### ZBTB11 acts as a transcription factor in lung cancer cells

The molecular implications of ZBTB11 in cancer cells are poorly reported. As ZBTB family members may serve as transcription factors, we were then intrigued to investigate whether ZBTB11 participated in gene transcriptional regulation. To this end, we interfered with ZBTB11 expression and evaluated the whole transcriptional profiles through RNA-sequencing (RNA-seq) analysis in H1299 cells. Knockdown was conducted by two individual oligos against *ZBTB11* for 96 h, and the knockdown efficiency was confirmed (Fig. 2a). We identified 78 upregulated genes and 125 downregulated genes upon ZBTB11 depletion

with the criteria of fold change ≥ 2 and $p < 0.05$ (Fig. 2b). Approximately 70% (142) of these genes were protein-coding, leaving the remaining minority further classified into noncoding RNA (~10%, 20), pseudogene (~9%, 19) and uncategorized groups (~11%, 22) (Fig. 2c).

To investigate whether ZBTB11 serves as a transcription factor in lung cancer cells, we next performed chromatin immunoprecipitation-sequencing (ChIP-seq) to evaluate the genome-wide distribution of chromatin-bound ZBTB11. Notably, the loci with ZBTB11 binding showed a significant enrichment around transcription starting sites (TSSs, range across −3 kb ~ +3 kb of TSS) genome wide with a consensus binding motif CC/AGGAAG (Fig. 2d, e), suggesting that ZBTB11 most likely functions as a transcription factor (TF) in terms of its regulation of gene expression in cancer cells. Indeed, the ZBTB11-specific ChIP-seq data revealed 23632 genes, more than 7-fold in number compared to the control ChIP-seq assay by normal IgG, which only defined 3369 genes (Fig. 2f). After ruling out the

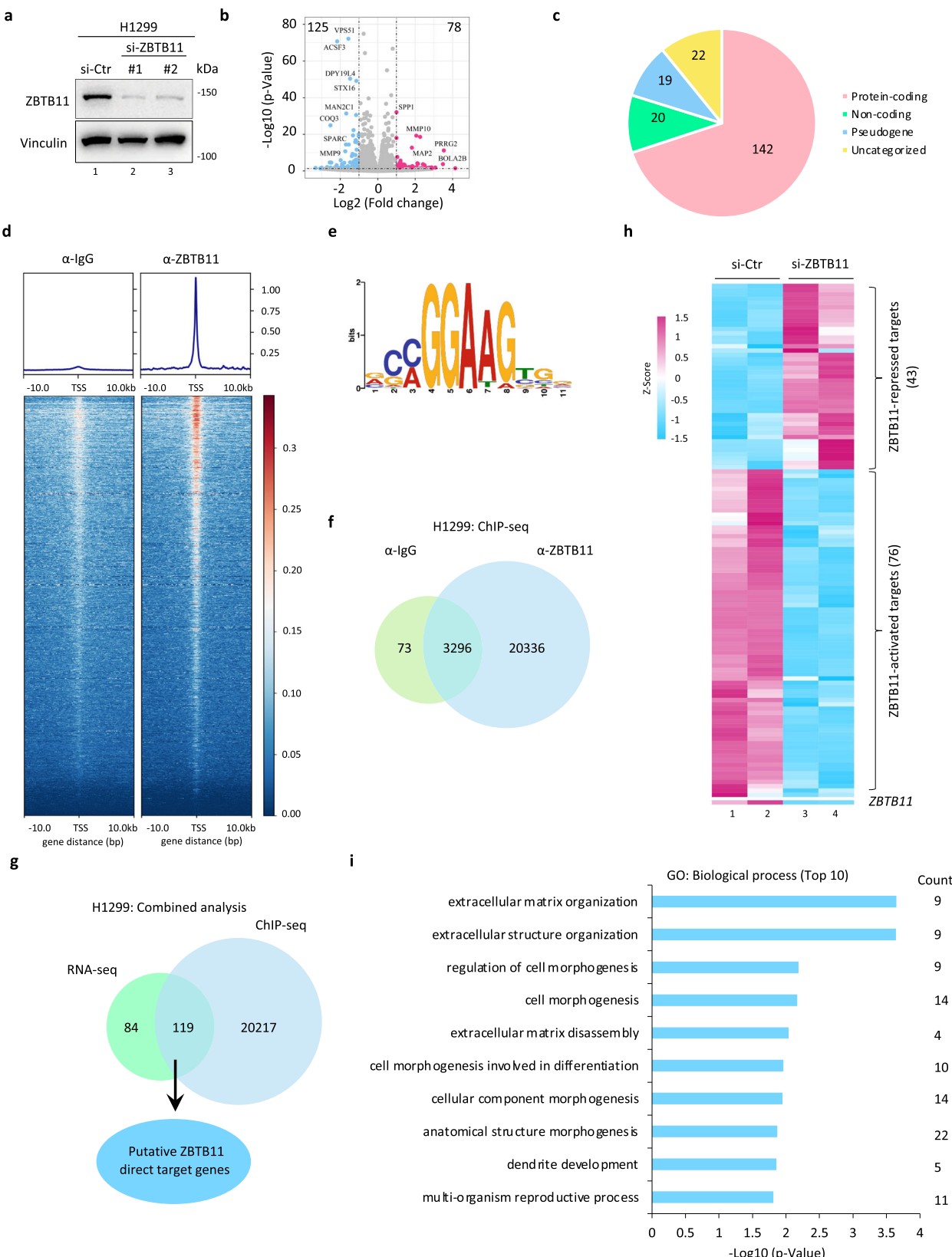

overlapping genes (3296), we collected 20336 genes with specific occupancy by ZBTB11 (Fig. 2f). The ChIP-seq results were further validated with ChIP–qPCR by monitoring ZBTB11 recruitment to a subgroup of identified loci that correspond to the promoters of genes including *ATPAF1, ANKRD40, COQ3, DUS1L, HEMK1, STX16, TACO1* and *TTC26* (Supplementary Fig. 2a–h).

By integral analysis of both the RNA-seq dataset and ChIP-seq dataset, we identified 119 putative ZBTB11 direct target genes (Fig. 2g). Among these putative targets, 43 genes were upregulated and 76 genes were downregulated upon ZBTB11 depletion (Fig. 2h), suggesting that ZBTB11 serves as either a transcriptional activator or repressor in lung cancer cells, probably depending on the regulatory context of a

**Fig. 2 | ZBTB11 acts as a transcription factor in lung cancer cells. a** WB analysis of ZBTB11 knockdown efficiency in H1299 cells transiently transfected with control siRNA (si-Ctr) or two individual siRNAs targeting ZBTB11 (si-ZBTB11) for 96 h. **b** Volcano plot revealing the differentially expressed genes upon ZBTB11 knockdown in H1299 cells (*n* = 2 biologically independent samples). The *p*-values were determined by Wald test. **c** Pie-plot showing the category of the differentially expressed genes upon ZBTB11 knockdown in H1299 cells. **d** Distribution of ZBTB11 ChIP-seq reads and heatmap of binding signals around the 10-kb windows centered on the transcription start site (TSS) of genes. **e** The ZBTB11 binding motif discovered de novo from ZBTB11-high peaks in ChIP-seq. **f** Venn diagram of the genes with ZBTB11 enrichment analyzed by ChIP-seq. ChIP with normal IgG served as a negative control. **g** Venn diagram showing putative ZBTB11 direct target genes by combinational analyses of both RNA-seq and ChIP-seq datasets. **h** Heatmap of putative ZBTB11 direct target genes with upregulation or downregulation upon ZBTB11 depletion. **i** Gene Ontology (GO) analysis of the top 10 biological processes enriched by the differentially expressed putative ZBTB11 direct target genes. The *p*-value was determined by one-sided hypergeometric test. Source data are provided as a Source Data file.

given target gene. Moreover, gene ontology (GO) analysis revealed that these putative targets of ZBTB11 were involved in the regulation of multiple biological processes, such as extracellular matrix organization, cell morphogenesis, dendrite development and multiorganism reproductive processes (Fig. 2i). Taken together, our data indicate that ZBTB11 is a transcription factor in lung cancer cells.

## SET cooperates with ZBTB11 in transcriptional regulation

To investigate the functional interplay between SET and ZBTB11 in lung cancer cells, we sought to identify SET-mediated regulation of ZBTB11 direct target genes. To this aim, we first evaluated differentially expressed genes by RNA-seq in H1299 cells upon SET depletion for 96 h (Fig. 3a). As shown in Fig. 3b, we identified 164 and 145 genes that were upregulated and downregulated, respectively, upon SET knockdown. Similarly, the majority of SET-regulated genes were also enriched in the protein-coding category (~67%, 207, Fig. 3c). Based on these datasets, we found 35 putative ZBTB11 direct target genes that were coregulated by SET (Fig. 3d). Interestingly, these genes can be clearly divided into two groups: 18 genes that were transcriptionally repressed upon both SET and ZBTB11 depletion and 17 genes that were transcriptionally activated upon both SET and ZBTB11 depletion (Fig. 3e). This transcriptional profile of SET/ZBTB11-coregulated target genes suggests that SET may functionally synergize with ZBTB11 for transcriptional regulation. Notably, GO analysis revealed that the top biological process enriched by the coregulated target genes was "extracellular matrix organization", exactly the same as the biological process enriched by ZBTB11-regulated genes (Figs. 3f vs. 2i). In addition, the RT−qPCR assay using target-specific primers further confirmed that the genes including *MMP9*, *MMP19*, *SPARC*, *PECAM1*, *SPP1* and *COL14A1*, which are involved in extracellular matrix organization, were indeed coregulated by both SET and ZBTB11 in H1299 cells (Fig. 3g). Similar transcriptional regulation of these genes upon SET and/or ZBTB11 knockdown was also recapitulated in H1975 lung cancer cells (Supplementary Fig. 2i). Collectively, these data suggest that SET cooperates with ZBTB11 in transcriptional regulation and prompted us to focus on the potential roles of the SET-ZBTB11 protein complex in the regulation of extracellular matrix-related biological processes upon tumor initiation and progression.

## SET-ZBTB11 complex promotes lung cancer cell metastasis

Dysfunction of extracellular matrix organization within tumor microenvironments (TMEs) is a key event during the tumor metastatic process[42]. In addition, the genes coregulated by ZBTB11 and SET, such as *MMP9* and *SPARC*, have been well validated to participate in the regulation of cancer cell migration and invasion[43–45]. Therefore, we hypothesized that the SET-ZBTB11 complex is involved in modulating the metastatic behaviors of cancer cells. To this end, we first evaluated cell migration and invasion in H1299-EV and H1299-FH-SET stable cells with or without endogenous ZBTB11 depletion (Fig. 4a). Knockdown of ZBTB11 markedly repressed cell migration and invasion, while SET overexpression modestly increased these metastatic behaviors (Fig. 4b–e). Notably, the SET overexpression-mediated effect on cell migration and invasion was largely abolished by concomitant knockdown of ZBTB11 (Fig. 4b–e). Next, we generated control or SFB-tagged

ZBTB11 stable cells (H1299-EV or H1299-ZBTB11-SFB) and evaluated the potential roles of ZBTB11 overexpression in regulating cell migration and invasion in these cells with or without SET depletion (Fig. 4f). As expected, knockdown of SET decreased cell migration and invasion (Fig. 4g–j). In addition, compared with the SET overexpression-mediated modest increase in cell migration and invasion (Fig. 4b–e), ZBTB11 overexpression dramatically boosted these metastatic behaviors (Fig. 4g–j). More importantly, knockdown of SET significantly abrogated ZBTB11 overexpression-induced upregulation of both cell migration and invasion (Fig. 4g–j).

To further investigate the role of the SET-ZBTB11 interaction in the metastatic regulation of cancer cells, we evaluated whether re-expression of RNAi-resistant full-length SET (re-FH-SET-FL) or acidic domain-deleted SET (re-FH-SET-ΔAD) is able to rescue the reduced cell migration and invasion caused by depletion of endogenous SET in H1299-ZBTB11-SFB stable cells (Supplementary Fig. 3a). As shown in Supplementary Fig. 3b, c, re-expression of SET-FL, but not SET-ΔAD, which does not bind with ZBTB11, obviously elevated cell migration and invasion, suggesting that the interaction between SET and ZBTB11 is critical for SET/ZBTB11-mediated metastatic regulation of cancer cells.

To validate the regulatory role of the SET-ZBTB11 complex under more physiological conditions, we disrupted the SET-ZBTB11 interaction by interfering with endogenous SET and/or ZBTB11 to evaluate potential alterations in cancer cell behaviors in H1299 cells (Fig. 4k). Knockdown of SET or ZBTB11 showed no obvious effects on cell proliferation, the cell cycle, the cellular response to the chemotherapeutic drug camptothecin (Cpt), or colony formation under our experimental conditions (Supplementary Fig. 4a–e). However, cell migration and invasion were markedly reduced up to 45% and 54%, respectively, upon SET depletion (Fig. 4l–o). In addition, depletion of ZBTB11 exhibited a more robust suppressive effect on the migration (~94%) and invasion (~87%) of H1299 cells (Fig. 4l–o). More importantly, this effect on cancer cell migration and invasion by SET depletion was abolished upon concomitant knockdown of ZBTB11 (Fig. 4l–o), suggesting that SET-mediated metastatic regulation of lung cancer cells is probably dependent on the presence of ZBTB11. Similar results were observed in another lung cell line H1975 (Supplementary Fig. 5a–e), supporting that SET-ZBTB11 complex-mediated regulation of migration and invasion in lung cancer cells represents a general phenomenon.

To corroborate SET-ZBTB11 complex-mediated modulation of cancer cell behaviors in vivo, we employed H1299-Luc2-tdT-2 reporter cells with stable knockdown of SET or ZBTB11 and monitored distal metastasis of the indicated cancer cells through a subcutaneous tumor xenograft mouse model (Fig. 4p). The stable knockdown of SET or ZBTB11 in H1299-reporter cells was validated (Fig. 4q). Notably, under our experimental conditions, stable knockdown of SET and/or ZBTB11 had no obvious effects on cell viability and cell proliferation (Supplementary Fig. 5f–g). In addition, depletion of SET and/or ZBTB11 did not impair primary tumor growth (Supplementary Fig. 5h–i). However, under the same conditions, knockdown of either SET or ZBTB11 markedly reduced distal lung metastasis of the primary tumors, as evidenced by in vivo bioluminescent imaging (Fig. 4r–s). More importantly, concomitant knockdown of SET and ZBTB11 in cancer

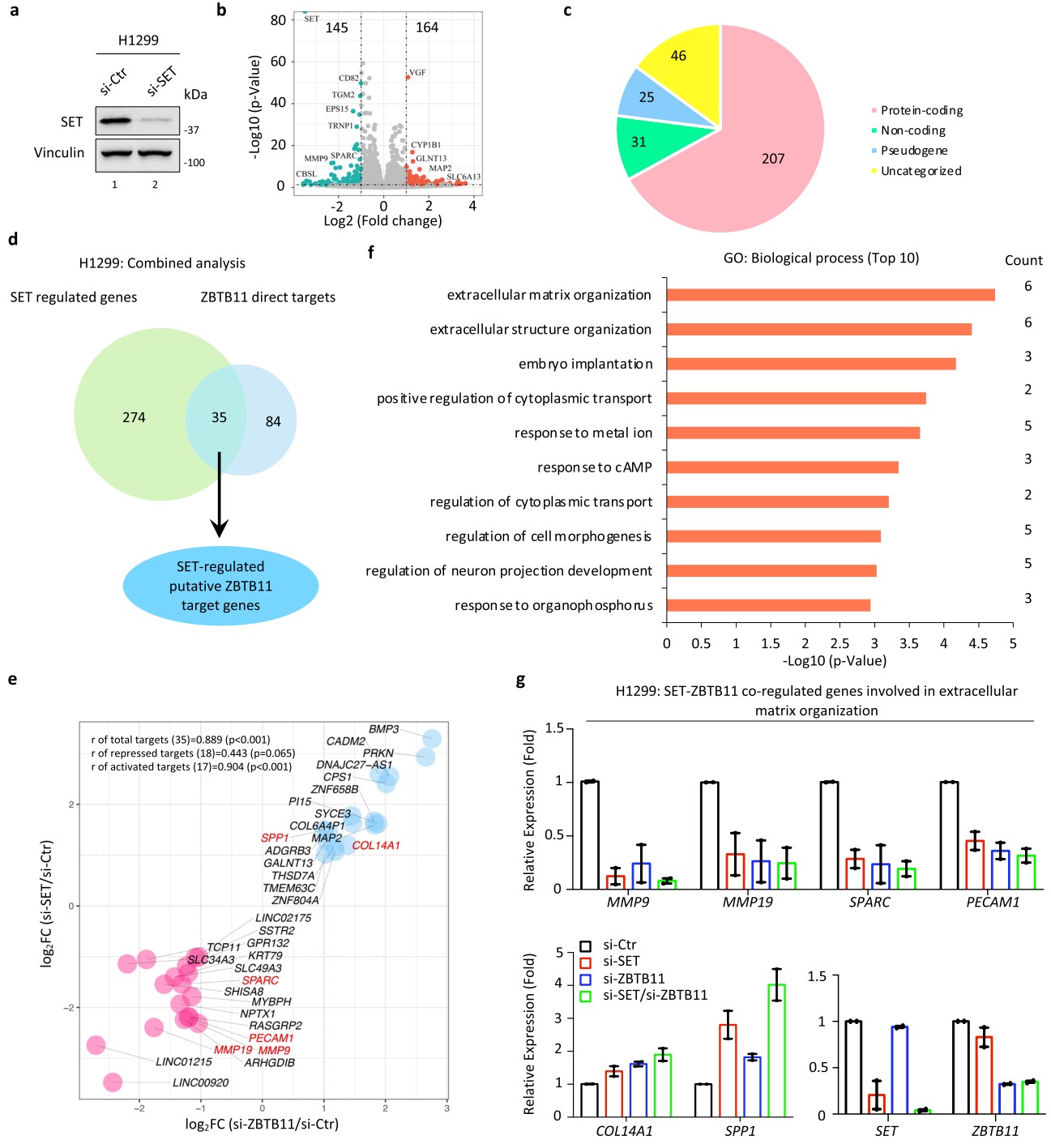

**Fig. 3 | SET cooperates with ZBTB11 in transcriptional regulation. a** WB analysis of SET knockdown efficiency in H1299 cells transiently transfected with control siRNA (si-Ctr) or siRNAs targeting SET (si-SET) for 96 h. **b** Volcano map of the differentially expressed genes upon SET knockdown in H1299 cells ($n = 2$ biologically independent samples). The *p*-value was determined by Wald test. **c** Pie-plot showing the category of the differentially expressed genes upon SET knockdown in H1299 cells. **d** Venn diagram of the ZBTB11 target genes coregulated by SET. **e** Correlation analysis of SET/ZBTB11-regulated genes revealing that SET and ZBTB11 synergistically modulate transcription. Pearson's correlation analysis was performed to calculate correlation coefficients and *p*-values. The genes related to extracellular matrix organization were highlighted with red. **f** GO analysis of the top 10 biological processes enriched by the SET-regulated ZBTB11 direct target genes. The *p*-values were determined by hypergeometric test. **g** RT−qPCR analysis of representative genes in H1299 cells with SET or ZBTB11 knockdown, individually or together. Data were shown as the mean ± S.E.M., $n = 2$ experimental replicates. Source data are provided as a Source Data file.

cells exhibited no further inhibitory effect on distal lung metastasis compared with SET or ZBTB11 depletion alone (Fig. 4r–s), supporting that both SET and ZBTB11, as well as their interaction, play important roles in promoting tumor metastasis in vivo. Taken together, our data indicate that the SET-ZBTB11 complex is critically involved in regulating lung cancer cell metastasis.

## MMP9 is a downstream effector of the SET-ZBTB11 complex in metastatic regulation

To obtain mechanistic insights into SET-ZBTB11 complex-regulated cancer cell metastasis, we focused on the coregulated genes involved in extracellular matrix organization (Fig. 3g), as they are functionally related to or even directly contribute to cancer cell metastatic

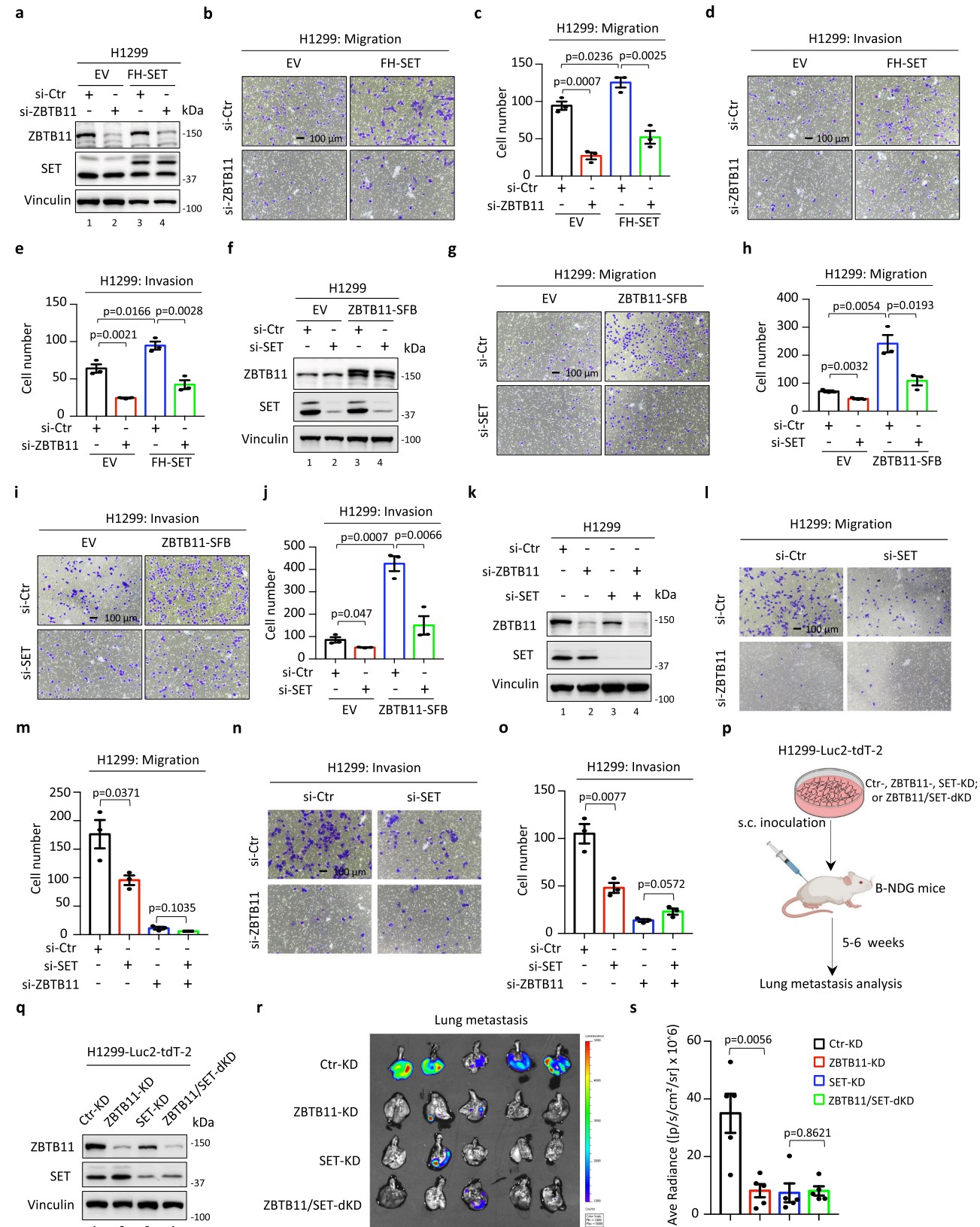

regulation. Since *SPP1* has been verified to promote metastasis[46,47] and the transcriptional changes in *COL14A1* were relatively weak (Fig. 3g), we speculated that these two genes might not be involved in metastatic inhibition upon SET or ZBTB11 depletion in our current experimental conditions. Therefore, we focused on *MMP9*, *MMP19*, *SPARC* and *PECAM1*. First, we evaluated whether depletion of these genes

recapitulated the changes in cancer cell metastatic behaviors observed upon SET or ZBTB11 knockdown. As shown in Supplementary Fig. 6a–c, knockdown of MMP19 or SPARC in H1299 cells had no obvious effect on cell migration and invasion. In contrast, depletion of MMP9 or PECAM1 resulted in a dramatic reduction in both cell migration and invasion (Supplementary Fig. 6d–m). Similar changes in

**Fig. 4 | SET-ZBTB11 complex promotes lung cancer cell metastasis. a** WB analysis of ZBTB11 knockdown efficiency in H1299-EV or H1299-FH-SET stable cells transiently transfected with control siRNA (si-Ctr) or siRNA targeting ZBTB11 (si-ZBTB11) for 96 h. Cell migration (**b**, **c**) or invasion (**d**, **e**) of H1299-EV or H1299-FH-SET stable cells depleted with or without ZBTB11. Data were shown as the mean ± S.E.M., *n* = 3 biologically independent samples. The *p*-value was determined by two-sided *t*-test. **f** WB analysis of SET knockdown efficiency in H1299-EV or H1299-ZBTB11-SFB stable cells transiently transfected with control siRNA (si-Ctr) or siRNA targeting SET (si-SET) for 96 h. Cell migration (**g**, **h**) or invasion (**i**, **j**) of H1299-EV or H1299-ZBTB11-SFB cells depleted with or without SET. Data were shown as the mean ± S.E.M., *n* = 3 biologically independent samples. The *p*-value was determined by two-sided *t*-test. **k** WB analysis of SET and ZBTB11 knockdown efficiency in H1299 cells transiently transfected with control siRNA (si-Ctr) or siRNA against SET or ZBTB11 (si-SET or si-ZBTB11) for 96 h. Cell migration (**l**, **m**) or invasion (**n**, **o**) of H1299 cells upon SET and/or ZBTB11 depletion. Data were shown as the mean ± S.E.M., *n* = 3 biologically independent samples. The *p*-value was determined by two-sided *t*-test. **p** Schematic diagram (created with BioRender) of the workflow for analyzing tumor metastasis in vivo. **q** WB analysis of SET and ZBTB11 knockdown efficiency in H1299-Luc2-tdT-2 cells stably transfected with control shRNA (sh-Ctr) or shRNA targeting SET and/or ZBTB11 (sh-SET and/or sh-ZBTB11). **r** Bioluminescent image of lung metastasis from the primary tumors in a mouse model where H1299-Luc2-tdT-2 cells with or without SET/ZBTB11 knockdown were subcutaneously inoculated into the flanks of immunodeficient B-NDG (NSG) mice. **s** Quantitative analysis of the metastasis of cancer cells in the lung based on (**r**). Data were shown as the mean ± S.E.M., *n* = 5 mice per group. The *p*-value was determined by two-sided *t*-test. Source data are provided as a Source Data file.

cell migration and invasion upon MMP9 or PECAM1 depletion were also observed in H1975 cells (Supplementary Fig. 6n–p). These results suggest that multiple downstream targets, such as MMP9 and PECAM1, contribute together to SET-ZBTB11 complex-mediated metastatic regulation.

MMP9 is well studied in promoting metastasis by degrading the extracellular matrix, while PECAM1 is mainly expressed on vascular endothelial cells and contributes to intercellular junctions between endothelial cells[43,48]. It has been reported that PECAM1 may act as a substrate of MMP9, which cleaves the extracellular portion of PECAM1 and facilitates leukocyte migration[49]. Thus, SET-ZBTB11 complex-mediated transcriptional activation of MMP9 is likely involved in the functional regulation of PECAM1 in certain biological contexts. Therefore, we focused on MMP9 for further study and investigated whether MMP9 serves as one of effectors that directly contribute to SET-ZBTB11 complex-mediated metastatic regulation. To this end, we generated control or MMP9 stably expressing H1299 cells by using a retrovirus infection method and evaluated whether MMP9 rescues ZBTB11 depletion-mediated changes in cell migration and invasion (Fig. 5a). As expected, ectopic MMP9 profoundly promoted cancer cell migration and invasion (Fig. 5b–e). In addition, ZBTB11 depletion-mediated reduction of cell migration and invasion can be largely reversed by MMP9 overexpression (Fig. 5b–e). Next, we interfered with SET expression in control or MMP9 stably expressing H1299 cells to evaluate whether MMP9 is involved in SET-mediated metastatic regulation (Fig. 5f). MMP9 markedly reversed the reduction in both cell migration and invasion induced by SET knockdown (Fig. 5g–j). On the one hand, these results reveal that MMP9 is critically involved in SET-ZBTB11 complex-mediated regulation of cancer cell metastatic behaviors. On the other hand, the partially rescued cell migration and invasion by MMP9 overexpression upon SET or ZBTB11 depletion further support the notion that multiple downstream targets may contribute together to SET-ZBTB11 complex-mediated metastatic regulation.

To elucidate the mechanism by which the SET-ZBTB11 complex regulates MMP9 expression, we performed a ChIP assay to evaluate the recruitment of ZBTB11 and SET to the *MMP9* locus. As ChIP-seq data suggested that ZBTB11 may bind within the gene body of *MMP9*, we then designed the corresponding primers and successfully detected both ZBTB11 and SET occupancy on the *MMP9* gene body (Fig. 5k). In addition, the re-ChIP assay of SET after the primary ChIP of ZBTB11 indicated that SET-ZBTB11 indeed formed a protein complex that colocalized within *MMP9* loci (Fig. 5l). Furthermore, the inducible knockout (iKO) of ZBTB11 by using the doxycycline (Doxy)-driven CrispR/Cas9 technique proved a remarkable dissociation of SET on *MMP9* loci, indicating a high dependency of ZBTB11 for SET recruitment on *MMP9* loci (Fig. 5m). Interestingly, we also found that SET overexpression promoted ZBTB11 recruitment to *MMP9* loci (Fig. 5n). This evidence suggests that SET-ZBTB11 complex formation might increase the binding affinity of ZBTB11 to its cognate DNA element.

Next, we performed a luciferase assay to evaluate whether SET is functionally involved in ZBTB11-mediated *MMP9* expression. As expected, overexpression of ZBTB11 was able to activate the luciferase reporter that contains the ZBTB11-binding element of *MMP9* (*MMP9*-reporter) (Fig. 5o). In addition, ZBTB11-mediated activation of the *MMP9* reporter was further increased by SET overexpression (Fig. 5o). Notably, this enhancement effect on *MMP9*-reporter activation was largely dependent on the interaction between SET and ZBTB11, since overexpression of SET-ΔAD that does not bind ZBTB11 failed to enhance ZBTB11-mediated activation of the *MMP9* reporter (Supplementary Fig. 6q). Our data indicate that MMP9 is a direct target of the SET-ZBTB11 complex, where SET promotes ZBTB11-mediated transcriptional activation of *MMP9* by serving as a co-factor of ZBTB11.

Finally, we generated a xenograft tumor metastasis model to evaluate the role of MMP9 in metastasis in vivo (Fig. 5p). Since SET acts as a co-factor that facilitates ZBTB11-dependent transcriptional activation of *MMP9*, we decided to simply overexpress ZBTB11 to drive *MMP9* expression in our model. As expected, overexpression of ZBTB11 readily upregulated the expression of MMP9, and this regulatory effect on MMP9 expression can be successfully blocked by shRNA targeting *MMP9* (Fig. 5q). Overexpression of ZBTB11 markedly promoted distal lung metastasis of the primary tumors that were inoculated subcutaneously on the flanked back (Fig. 5r, s). More importantly, knockdown of MMP9 remarkably suppressed ZBTB11-induced tumor metastasis (Fig. 5r, s). Taken together, based on both in vitro and in vivo analyses, our data reveal that MMP9 represents one of the key downstream effectors that contribute to SET-ZBTB11 complex-mediated regulation of tumor metastasis.

## Transcriptional repression of PRRG2 contributes to ZBTB11-induced metastasis independently of the SET-ZBTB11 complex

Since knockdown of ZBTB11 displayed a more robust effect on cancer cell migration and invasion than SET depletion (Fig. 4l–o and Supplementary Fig. 5b–e), we speculated that ZBTB11 likely maintains additional ways to modulate metastasis independent of SET-ZBTB11 complex formation. To this end, we focused on the target genes whose expression was regulated only by ZBTB11 but not by SET. Among these genes, 26 and 58 were upregulated and downregulated upon ZBTB11 knockdown, respectively (Supplementary Fig. 7a). We noticed that *PRRG2*, a previously less characterized gene, displayed the most significant and reliable changes in its expression (considering both fold change and *p*-value) upon ZBTB11 knockdown (Supplementary Fig. 7a). Although the functions of PRRG2 are poorly characterized, we found that PRRG4, a member of the PRRG family of proteins, was recently reported to be involved in metastatic regulation[50]. Thus, we were prompted to investigate whether PRRG2 transcriptionally regulated by ZBTB11 also participates in metastatic modulation.

We generated H1299 cells stably expressing PRRG2 (H1299-PRRG2-Flag, Fig. 6a). There was a minimal effect on cell proliferation upon PRRG2 overexpression (Supplementary Fig. 7b). However, the expression ectopic PRRG2 profoundly inhibited cell migration and

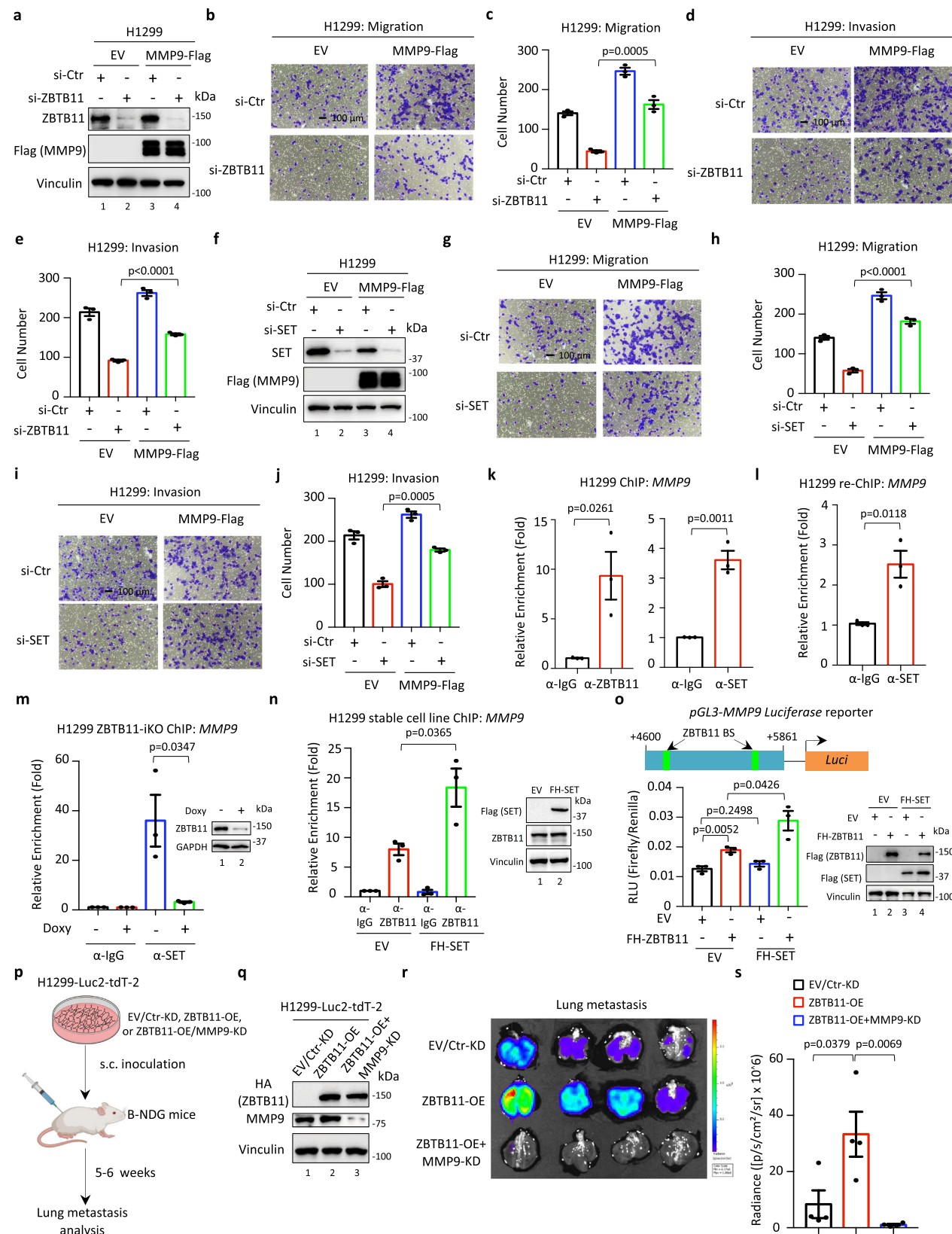

invasion (Fig. 6b–e), suggesting that the upregulation of PPRG2 has an antimetastatic effect on lung cancer cells. As expected, knockdown of ZBTB11 strongly elevated the expression of *PRRG2*, which can be largely blocked by concomitantly transfecting siRNA against *PRRG2* (Fig. 6f). Knockdown of PRRG2 not only increased cell migration under normal conditions but also dramatically rescued the ZBTB11

knockdown-mediated reduction of cell migration (Fig. 6g–h). For cell invasion, we did not find an obvious change upon PRRG2 depletion alone (Fig. 6i–j). However, knockdown of PRRG2 was able to reverse the reduction of cell invasion caused by ZBTB11 depletion (Fig. 6i–j). In contrast, depletion of SET did not increase *PRRG2* expression in H1299 cells (Supplementary Fig. 7c), consistent with the observations from

**Fig. 5 | MMP9 is a downstream effector of the SET-ZBTB11 complex in metastatic regulation. a** WB analysis of ZBTB11 in H1299-EV or H1299-MMP9-Flag stable cells upon endogenous ZBTB11 depletion. Cell migration (**b, c**) or invasion (**d, e**) of H1299-EV or H1299-MMP9-Flag stable cells upon ZBTB11 depletion (mean ± S.E.M., *n* = 3 biologically independent samples, two-sided *t*-test). **f** WB analysis of SET in H1299-EV or H1299-MMP9-Flag stable cells upon endogenous SET depletion. Cell migration (**g, h**) or invasion (**i, j**) of H1299-EV or H1299-MMP9-Flag stable cells upon SET depletion (mean ± S.E.M., *n* = 3 biologically independent samples, two-sided *t*-test). **k** ChIP–qPCR analysis of the enrichment of ZBTB11 and SET at *MMP9* loci in H1299 cells (mean ± S.E.M., *n* = 3 experimental replicates, two-sided *t*-test). **l** Re-ChIP–qPCR analysis of SET binding to *MMP9* loci following primary ChIP with anti-ZBTB11 antibody in H1299 cells (mean ± S.E.M., *n* = 3 experimental replicates, two-sided *t*-test). **m** ChIP–qPCR analysis of SET enrichment at *MMP9* loci in H1299-ZBTB11-iKO cells upon doxycycline (Doxy) treatment (mean ± S.E.M., *n* = 3 experimental replicates, two-sided *t*-test). **n** ChIP–qPCR analysis of ZBTB11 enrichment at *MMP9* loci in H1299-EV or H1299-FH-SET stable cells (mean ± S.E.M., *n* = 3

experimental replicates, two-sided *t*-test). **o** Luciferase assays of SET/ZBTB11-driven transcriptional of *MMP9*. The EV or FH-ZBTB11 construct was transfected into H1299-EV or H1299-FH-SET stable cells, together with luciferase reporter and Renilla control vector, for 24 h (mean ± S.E.M., *n* = 3 experimental replicates, two-sided *t*-test). **p** Schematic diagram (created with BioRender) of the workflow for analyzing tumor metastasis in vivo. **q** WB analysis of ZBTB11 and MMP9 in H1299-Luc2-tdT-2 cells stably transfected with the ZBTB11-expressing construct (ZBTB11-OE) and/or shRNA targeting MMP9 (MMP9-KD). **r** Bioluminescent image of lung metastasis from the primary tumors in a mouse model where H1299-Luc2-tdT-2 cells with ZBTB11-OE/MMP9-KD were subcutaneously inoculated into the flanks of immunodeficient B-NDG (NSG) mice. **s** Quantitative analysis of the metastasis of cancer cells in the lung based on (**r**) (mean ± S.E.M., *n* = 4 mice per group, two-sided *t*-test). **c, h** shared the same si-Ctr group for quantitative analysis; (**e**) and (**j**) shared the same si-Ctr group for quantitative analysis. Source data are provided as a Source Data file.

RNA-seq analysis. Thus, as expected, the suppression of cell migration and invasion caused by SET depletion in H1299 cells was not reversed by concomitant depletion of PRRG2 (Supplementary Fig. 7d–g). Similar to the findings in H1299 cells, depletion of SET in H1975 cells also resulted in no changes in *PRRG2* transcription (Supplementary Fig. 7h). Accordingly, neither cell migration nor invasion regulated by SET knockdown was rescued by simultaneous PPRG2 depletion (Supplementary Fig. 7i–l).

Since *PRRG2* was a putative direct target of ZBTB11 that showed a strong binding ability within the *PRRG2* promoter, as observed by ChIP-seq assay (Fig. 6k), we then validated this occupancy by ChIP–qPCR. Indeed, ZBTB11 exhibited a significant enrichment on the *PRRG2* promoter (Fig. 6k). To evaluate the functional consequence of the recruitment of ZBTB11 to the *PRRG2* promoter, we generated a luciferase reporter construct by cloning the ZBTB11-binding element of *PRRG2* into the pGL3 construct and measured whether ZBTB11 affects luciferase activity (Fig. 6l). Knockdown of endogenous ZBTB11 was sufficient to activate luciferase activity, whereas overexpression of ectopic ZBTB11 repressed luciferase activity (Fig. 6l), indicating that ZBTB11 transcriptionally represses *PRRG2* by acting as a master transcription factor for *PRRG2*.

To assess the role of PRRG2 in vivo, we established a xenograft tumor model to evaluate whether PRRG2 contributes to ZBTB11-mediated regulation of tumor metastasis (Fig. 6m). As expected, knockdown of ZBTB11 markedly elevated PRRG2 expression, which was abolished upon PRRG2 knockdown with shRNA (Fig. 6n). Again, depletion of endogenous ZBTB11 remarkably suppressed distal lung metastasis of the primary subcutaneous tumors (Fig. 6o–p). More importantly, this suppressive effect was largely reversed upon simultaneous knockdown of PRRG2 (Fig. 6o–p). Taken together, our data indicate PRRG2 as a direct target of ZBTB11, which contributes to ZBTB11-, but not SET-, mediated regulation of tumor metastasis.

## ZBTB11-PRRG2 axis links YAP1 for metastatic regulation

We next attempted to investigate the potential mechanisms by which PRRG2 represses cancer cell metastatic behaviors. To date, functional studies of PRRG2 are almost vacant. Previous analyses of PRRG2-binding proteins showed that PRRG2 may directly interact with YAP1[51,52]. Since accumulating evidence indicates that dysfunction of YAP/TAZ, such as amplification or aberrant activation, has emerged as a key event during cancer cell metastasis[53–56], we were then prompted to hypothesize whether PRRG2 participates in regulating cancer cell migration and invasion through a YAP1-involved mechanism. To this end, we first confirmed the interaction between PRRG2 and YAP1 in lung cancer cells (Fig. 7a). Next, we depleted cellular YAP1 in H1299-EV or H1299-PRRG2-Flag cells to measure potential changes in cancer cell metastatic behaviors (Fig. 7b). Notably, depletion of YAP1 had no effect on SET or ZBTB11 expression (Supplementary Fig. 8a). However,

consistent with a previous report[57], we observed that depletion of YAP1 downregulated MMP9 (Supplementary Fig. 8a), suggesting that YAP1 might participate in ECM remodeling during lung tumor progression. As expected, knockdown of YAP1 reduced both cell migration and invasion (Fig. 7c–f). More importantly, the PRRG2 overexpression-mediated reduction in cancer cell migration and invasion was profoundly abolished upon concomitant silencing of YAP1 (Fig. 7c–f), supporting a critical role of YAP1 in PRRG2-mediated metastatic regulation.

Aberrant YAP1 activation is sufficient to induce tumor progression[56,58], including in LUAD[59]. The phosphorylation of YAP1 mediated by upstream kinases plays a critical role in modulating YAP1 activity. For example, the Hippo pathway component LATS1/2-mediated phosphorylation of YAP1 at the serine 127 residue (p-YAP1-S127) may functionally inactivate YAP1 and sequester YAP1 within the cytoplasm[60,61]. Thus, p-YAP1-S127 may serve as one of reliable markers to dictate YAP1 activity. Since YAP1 interacted with PRRG2 and was largely required for PRRG2-mediated repression of cancer cell migration and invasion (Fig. 7a–f), we mechanistically speculated that PRRG2 likely exerts its metastatic suppressive actions by inhibiting YAP1 activity. To this end, we monitored YAP1 phosphorylation in control or PRRG2 stably expressing lung cancer cells. As shown in Fig. 7g, PRRG2 overexpression markedly elevated p-YAP1-S127 without changing overall YAP1 levels. Interestingly, p-YAP1-S397, a key phosphorylation that primes YAP1 for subsequent phosphorylation and degradation[62], displayed no alteration upon PRRG2 overexpression (Fig. 7g), consistent with the observation that PRRG2 had no effect on total YAP1 protein levels. In addition, ZBTB11 depletion induced an elevation of p-YAP1-S127, and this regulation was remarkably abrogated upon concomitant knockdown of PRRG2 (Fig. 7h). Along with these findings, we observed that a subgroup of metastasis-related YAP1 target genes, such as *CTGF*, *FOXM1* and *THBS1*, were downregulated upon ZBTB11 knockdown, and this effect could be reversed by depleting PRRG2 and ZBTB11 simultaneously (Supplementary Fig. 8b). This evidence suggests that the phosphorylation of YAP1 at S127 may act as a key event during ZBTB11-PRRG2 axis-mediated metastatic regulation.

To further evaluate the role of p-YAP1-S127 in ZBTB11-PRRG2 axis-mediated tumor metastatic regulation, we introduced a YAP1-S127A mutant that represents a constitutively activated form of YAP1[60,61,63]. Overexpression of the YAP1-S127A mutant readily reversed the suppressive effect on cancer cell migration induced by PRRG2 overexpression or ZBTB11 knockdown (Supplementary Fig. 8c–h). Moreover, we generated xenograft tumor models to measure the role of YAP1 and its phosphorylation in the regulation of metastasis in vivo (Fig. 7i). To this end, we generated H1299 cell lines that stably express PRRG2 and/or YAP1-S127A (Fig. 7j). We observed that PRRG2 overexpression indeed suppressed tumor cell metastasis (Fig. 7k, l). In addition, overexpression of the YAP1-S127A mutant significantly

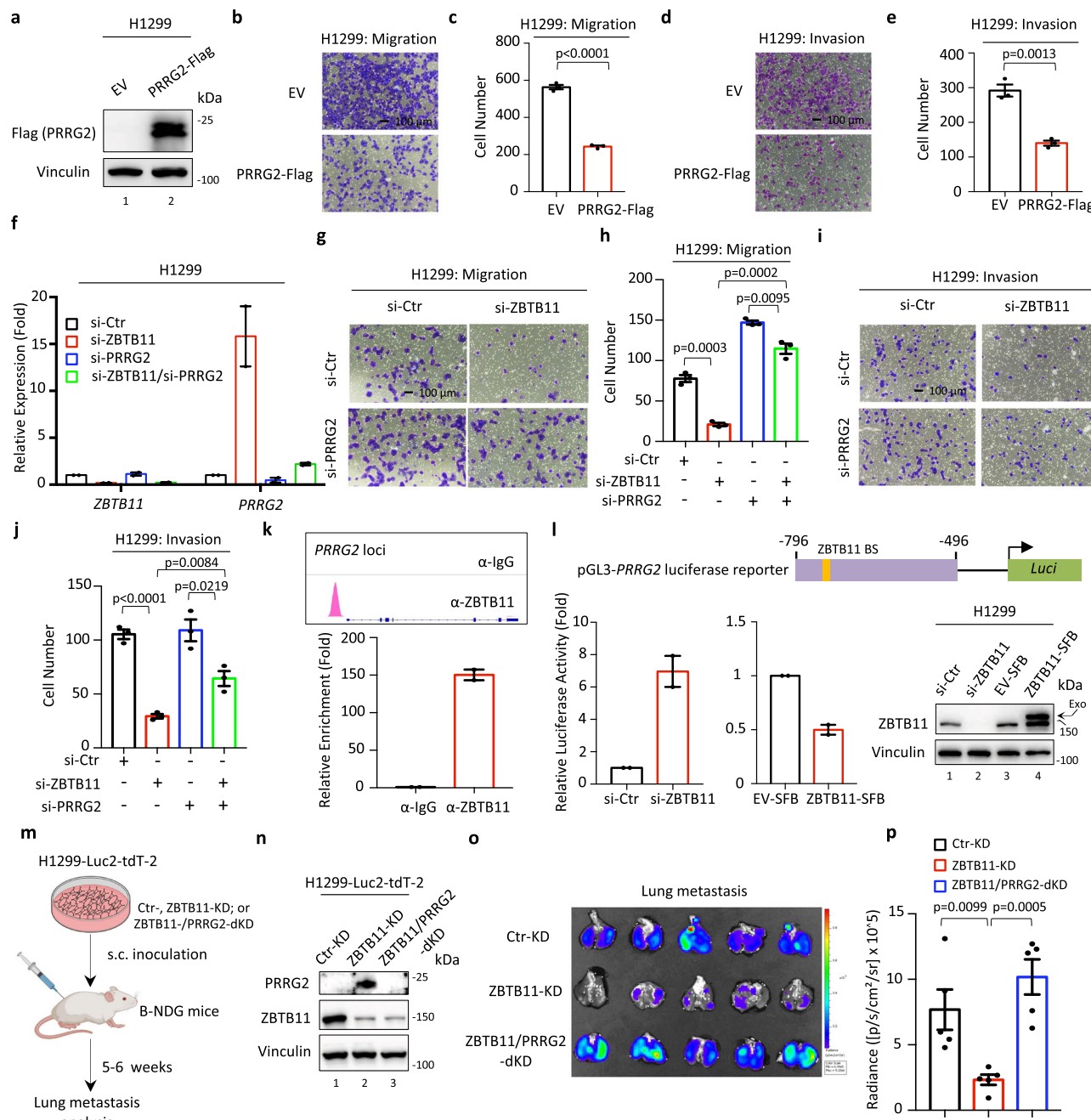

**Fig. 6 | Transcriptional repression of PRRG2 contributes to ZBTB11-induced metastasis independently of the SET-ZBTB11 complex. a** WB analysis of ectopic PRRG2 in H1299-EV or H1299-PRRG2-Flag stable cell lines. Cell migration (**b**, **c**) or invasion (**d**, **e**) of H1299-EV or H1299-PRRG2-Flag stable cells (mean ± S.E.M., *n* = 3 biologically independent samples, two-sided *t*-test). **f** RT–qPCR analysis of *ZBTB11* or *PRRG2* expression in H1299 cells depleted with or without ZBTB11 or PRRG2 for 96 h (mean ± S.E.M., *n* = 2 experimental replicates). Cell migration (**g**, **h**) or invasion (**i**, **j**) assays of H1299 cells with ZBTB11 or PRRG2 depletion, individually or together (mean ± S.E.M., *n* = 3 biologically independent samples, two-sided *t*-test). **k** ChIP-seq and ChIP–qPCR analysis of ZBTB11 enrichment on the *PRRG2* promoter in H1299 cells (mean ± S.E.M., *n* = 2 experimental replicates). **l** Luciferase assays of ZBTB11-driven transcriptional regulation of *PRRG2* (mean ± S.E.M., *n* = 2 experimental replicates). The luciferase reporter containing the ZBTB11-binding element

of the *PRRG2* promoter and Renilla control were transfected into the cells as indicated for 24 h. For ZBTB11 depletion, siRNA against ZBTB11 was used. For overexpression of ZBTB11, the H1299-ZBTB11-SFB stable cell line was used. The ZBTB11 knockdown efficiency or the expression of ectopic ZBTB11 was validated by WB assay. **m** Schematic diagram (created with BioRender) of the workflow for analyzing tumor metastasis in vivo. **n** WB analysis of ZBTB11 and PRRG2 in H1299-Luc2-tdT-2 cells stably knocked down with ZBTB11 (ZBTB11-KD) and/or PRRG2 (PRRG2-KD). **o** Bioluminescent image of lung metastasis from the primary tumors in a mouse model where H1299-Luc2-tdT-2 cells with ZBTB11-KD and/or PRRG2-KD were subcutaneously inoculated into the flanks of immunodeficient B-NDG (NSG) mice. **p** Quantitative analysis of the metastasis of cancer cells in the lung based on (**o**) (mean ± S.E.M., *n* = 5 mice per group, two-sided *t*-test). Source data are provided as a Source Data file.

abolished PRRG2-mediated inhibition of tumor metastasis (Fig. 7k, l). These data indicate a metastasis-suppressive role of PRRG2 by functionally regulating YAP1. Furthermore, we investigated whether PRRG2-mediated YAP1 regulation contributes to ZBTB11-controlled

metastasis in vivo. As expected, ZBTB11 overexpression promoted distal metastasis of the primary tumors (Fig. 7m−o). Notably, overexpression of PRRG2 or depletion of YAP1 markedly attenuated ectopic ZBTB11-induced tumor metastasis (Fig. 7m−o). Taken together, our

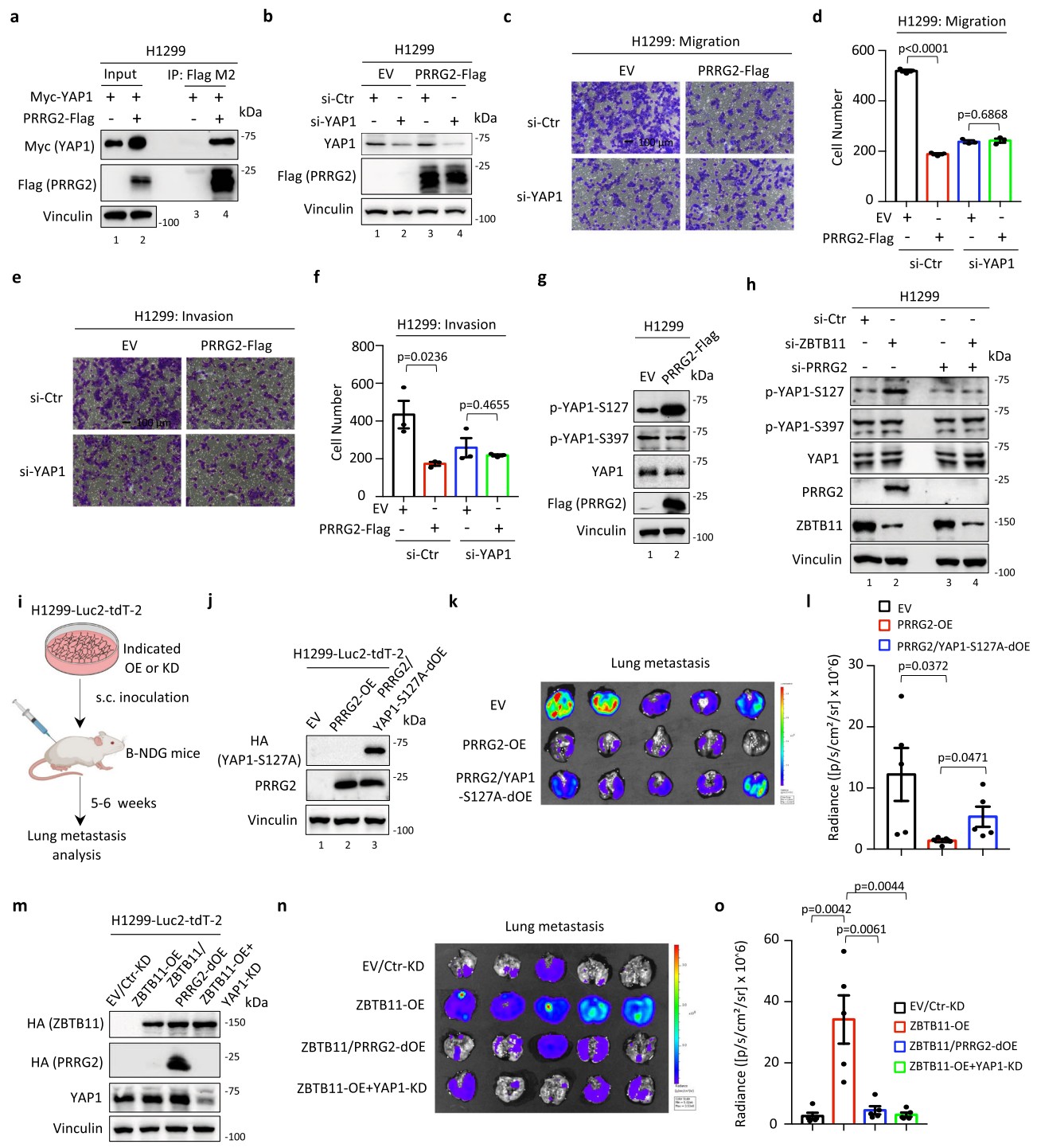

**Fig. 7 | ZBTB11-PRRG2 axis links YAP1 for metastatic regulation. a** Co-IP-WB assay of the interaction between YAP1 and PRRG2 in H1299 cells transfected with or without Myc-YAP1 and PRRG2-Flag, as indicated. **b** WB analysis of YAP1 knockdown in H1299-EV or H1299-PRRG2-Flag cells transfected with control siRNA (si-Ctr) or siRNA against YAP1 (si-YAP1) for 96 h. Cell migration (**c**, **d**) or invasion (**e**, **f**) of H1299-EV or H1299-PRRG2-Flag cells with or without YAP1 depletion (mean ± S.E.M., *n* = 3 biologically independent samples, two-sided *t*-test). **g** WB analysis of YAP1 phosphorylation in H1299-EV and H1299-PRRG2-Flag stable cells with the indicated antibodies. **h** WB analysis of YAP1 phosphorylation in H1299 cells with or without ZBTB11 and/or PRRG2 depletion, as indicated, for 48 h. **i** Schematic diagram of the workflow for analyzing tumor metastasis in vivo. **j** WB analysis of PRRG2 and YAP1 in H1299-Luc2-tdT-2 cells stably expressing PRRG2 (PRRG2-OE) and/or the YAP1-S127A construct (YAP1-S127A-OE). **k** Bioluminescent image of lung

metastasis from the primary tumors in a mouse model where H1299-Luc2-tdT-2 cells with PRRG2-OE and/or YAP1-S127A-OE were subcutaneously inoculated into the flanks of immunodeficient B-NDG (NSG) mice. **l** Quantitative analysis of the metastasis of cancer cells in the lung based on (**k**) (mean ± S.E.M., *n* = 5 mice per group, two-sided *t*-test). **m** WB analysis of ZBTB11, PRRG2 and YAP1 in H1299-Luc2-tdT-2 cells stably expressing the ZBTB11 (ZBTB11-OE) and/or PRRG2 (PRRG2-OE) construct and/or depleted of YAP1 (YAP-KD), as indicated. **n** Bioluminescent image of lung metastasis from the primary tumors in a mouse model where H1299-Luc2-tdT-2 cells with ZBTB11-OE, PRRG2-OE and/or YAP1-KD were subcutaneously inoculated into the flanks of immunodeficient B-NDG (NSG) mice. **o** Quantitative analysis of the metastasis of cancer cells in the lung based on (**n**) (mean ± S.E.M., *n* = 5 mice per group, two-sided *t*-test). Source data are provided as a Source Data file.

data reveal that YAP1 acts as a downstream effector of the ZBTB11-PRRG2 regulatory axis to modulate tumor metastasis.

## Loss of ZBTB11 suppresses genetic aberration-induced lung tumor metastasis

To further investigate the role of ZBTB11 in tumor behaviors in vivo, we generated a *Zbtb11* conditional knockout (cKO) mouse model by the CrispR/Cas9 technique (Supplementary Fig. 9a). The floxed allele of *Zbtb11*, as well as its germline transmission, was confirmed by Southern blot analysis of genomic DNA isolated from F1 mice (Supplementary Fig. 9b). We further obtained F2 *Zbtb11^Fl/Fl* homozygous offspring by intercrossing *Zbtb11^+/Fl* F1 mice (Supplementary Fig. 9c) and validated the complete abrogation of *Zbtb11* expression in *Zbtb11^Fl/Fl* mouse embryonic fibroblasts (MEFs) infected with adenovirus expressing Cre recombinase (Supplementary Fig. 9d). *Zbtb11^+/-* offspring obtained by intercrossing *Zbtb11^Fl/Fl* mice with *CMV-Cre* transgenic mice were viable and fertile. However, the intercrossing of *Zbtb11^+/-* mice failed to obtain *Zbtb11^-/-* offspring despite a normal sexual ratio (Supplementary Fig. 9e–f), which demonstrates an embryonically lethal phenotype of the *Zbtb11* KO mice. Moreover, both the repression of *Mmp9* and the activation of *Prrg2* were easily recapitulated in *Zbtb11* KO MEFs (Supplementary Fig. 9g), further confirming our previous results in cancer cells where MMP9 and PRRG2 are critical downstream targets of ZBTB11.

Since somatic activation of oncogenic *Kras* and inactivation of *Lkb1* have been well-established strategies to induce sporadic lung tumors with distal metastasis[64–66], we then introduced a *Kras^+/LSL-G12D*, *Lkb1^Fl/Fl*, *Rosa26-e(CAG-LSL-Luc-EGPF)* mouse model, in which lung tumor formation and metastasis can be induced by inhalation of adenovirus-based Cre recombinase (Ad-Cre) and observed by bioluminescence through an in vivo imaging system (IVIS) upon D-luciferin administration. After intercrossing with *Zbtb11^Fl/Fl* mice, we successfully obtained *Zbtb11^+/+*, *Kras^+/LSL-G12D*, *Lkb1^Fl/Fl*, *Rosa26-e(CAG-LSL-Luc-EGFP)* and *Zbtb11^Fl/Fl*, *Kras^+/LSL-G12D*, *Lkb1^Fl/Fl*, *Rosa26-e(CAG-LSL-Luc-EGFP)* mouse models (hereafter named KLLE and KLLE-Zbtb11^Fl/Fl, respectively) (Fig. 8a). Approximately 9 weeks after Ad-Cre inhalation, the bioluminescent signals within the chest could be clearly monitored by in vivo imaging system (IVIS), suggesting the formation of primary lung tumors (Fig. 8b). In addition, the isolated lung tissues exhibited striking bioluminescent signals that corresponded to the local sporadic tumors observed macroscopically on the lung surface (Fig. 8c). Notably, despite no statistical significance, loss of Zbtb11 still showed a tendency of reduced or slowed formation of the primary lung tumors induced by aberrant Kras and Lkb1 (Fig. 8d), supporting ZBTB11 as an oncoprotein that may contribute to tumor initiation, to some extent, in vivo. We also evaluated whether ZBTB11-mediated regulation of ECM remodeling, MMP9 expression and the PRRG2-YAP1 axis were recapitulated in vivo. As shown in Supplementary Fig. 10a, loss of Zbtb11 increased the collagen IV levels surrounding tumor cells. This phenomenon was consistent with the downregulation of Mmp9 upon Zbtb11 loss (Supplementary Fig. 10b), since collagen IV has long been recognized as a substrate of MMP9 for degradation[43]. This evidence suggests that ZBTB11 is physiologically involved in ECM remodeling during lung tumor progression. In contrast to Mmp9 expression, Prrg2 displayed upregulation upon Zbtb11 loss (Supplementary Fig. 10c). Although the overall levels of Yap1 were comparable in lung tumors between the KLLE and KLLE-Zbtb11^Fl/Fl groups (Supplementary Fig. 10d), loss of Zbtb11 dramatically increased p-Yap1-S127 (Supplementary Fig. 10e), further validating the existence of the ZBTB11-PRRG2-YAP1 regulatory axis in vivo.

Next, we evaluated the distal metastasis of primary lung tumors in the KLLE and KLLE-Zbtb11^Fl/Fl mouse models. Ex vivo bioluminescence assays of the dissected tissues confirmed that multiple organs, including the liver, kidney, spleen and intestine, were targeted for metastasis by primary lung tumors (Fig. 8e). More importantly, the metastasis of the primary lung tumors in KLLE mice was much more

severe than that in KLLE-Zbtb11^Fl/Fl mice (Fig. 8e). As expected, the heterozygosity of metastatic tumors among different organs was observed, and metastasis to the liver was more striking than that to the kidney, spleen and intestine (Fig. 8e). Indeed, quantitative analyses validated a statistically significant reduction in overall metastasis in the liver (Fig. 8f) but not in the kidney, spleen or intestine in KLLE-Zbtb11^Fl/Fl mice (Fig. 8g–i). Furthermore, we observed that the overall survival of KLLE-Zbtb11^Fl/Fl mice was significantly extended compared with that of KLLE mice (Fig. 8j). Taken together, our data demonstrate that loss of ZBTB11 suppresses genetic aberration-induced lung tumor metastasis.

## Dysfunction of ZBTB11 and SET correlates with a poor prognosis of lung cancer

To evaluate the clinical relevance of ZBTB11 and SET in lung cancer progression, we analyzed the TCGA database (https://ualcan.path.uab.edu/analysis.html)[67] and found that the high expression of both ZBTB11 and SET was positively correlated with LUAD or LUSC (Fig. 9a and Supplementary Fig. 10f). Specifically, LUAD stages 1, 3 and 4, but not stage 2, displayed higher expression of ZBTB11 than normal lung tissues, while SET overexpression was exhibited in all stages of LUAD (Fig. 9b). In addition, the primary lung tumors of the patients without adjacent lymph node metastasis (N0) showed the highest levels of ZBTB11 expression in statistics in comparison with those of patients with affected lymph nodes (Fig. 9c, left panel). In contrast, the expression of SET in primary lung tumors showed an increasing tendency with the progression of lymph node metastasis (Fig. 9c, right panel). Moreover, the expression of ZBTB11 and SET in lung tissues exhibited a strong correlation (Fig. 9d), supporting the notion of cooperation between ZBTB11 and SET in promoting lung cancer progression.

In addition, we evaluated the protein levels of ZBTB11 and SET in tissue arrays containing paired adjacent normal lung tissues and primary LUAD and distal metastatic tumors. As shown in Fig. 9e, the primary tumor samples displayed higher expression of both ZBTB11 and SET than the adjacent tissues. More importantly, both ZBTB11 and SET showed a further increase in their expression in metastatic tumors (Fig. 9e), strongly supporting critical roles of aberrant ZBTB11 and SET in modulating distal metastasis of lung tumors. Consistent with the analysis of the TCGA dataset[67], the protein levels of ZBTB11 and SET also displayed a positive correlation in lung tissues (Fig. 9f).

Furthermore, based on analyses of a lung cancer dataset (GSE37745)[68], we observed a positive correlation between *MMP9* and *SET* expression (Supplementary Fig. 10g). In addition, the expression of *ZBTB11* and *MMP9* displayed a tendency of positive correlation, although the statistical significance was not sound (Supplementary Fig. 10g). Since the expression of *MMP9* can be transcriptionally regulated by SET in a ZBTB11-dependent manner, these correlations might reflect that the overexpression of SET is sufficient to activate ZBTB11-mediated *MMP9* transcriptional upregulation by acting as a cofactor. For *PRRG2*, we observed that the expression of ZBTB11, but not SET, exhibited a significant negative correlation with *PRRG2* expression (Supplementary Fig. 10h), consistent with our data that *PRRG2* was transcriptionally repressed by ZBTB11 but not SET.

Finally, we investigated whether dysfunction of ZBTB11 and/or SET contributed to the prognosis of lung cancer. To this end, we mined a GEO dataset (GSE30219)[69] and observed that high expression levels of both *ZBTB11* and *SET* correlated with a poor prognosis, as evidenced by markedly reduced overall survival (OS) (Fig. 9g). Since *MMP9* and *PRRG2* are key downstream targets of ZBTB11 and/or SET in regulating the metastasis of lung cancer cells, we also considered them in the clinical survival analyses. As shown in Fig. 9g, the high expression of *MMP9* was associated with reduced OS, whereas the high expression of *PRRG2* was relevant to extended OS. In addition to the correlation with OS, we found that high expression of *SET* or low expression of *PRRG2* was also correlated very well with reduced disease-free survival (DFS)

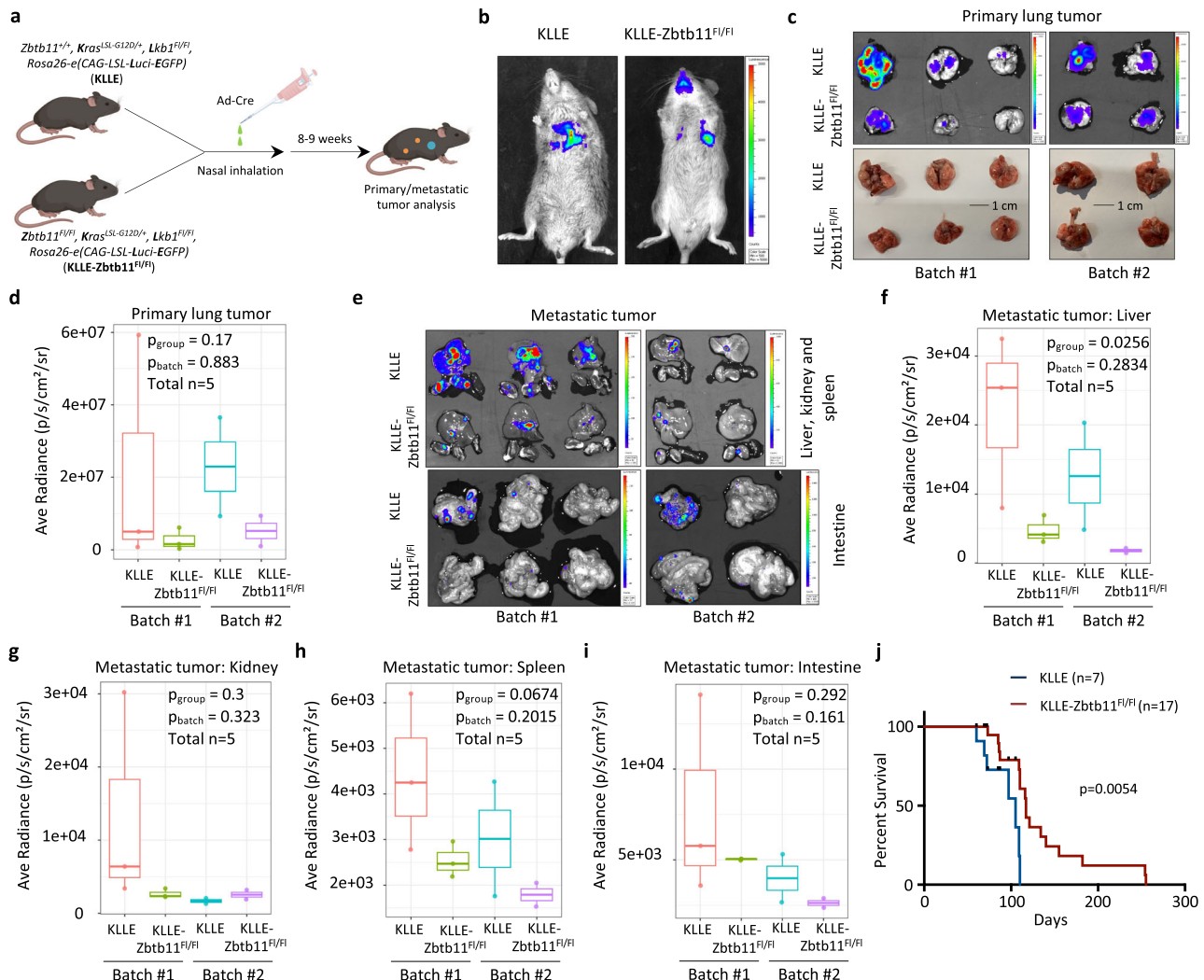

**Fig. 8 | Loss of ZBTB11 suppresses genetic aberration-induced lung tumor metastasis. a** Schematic diagram (created with BioRender) of the strategy to establish a metastatic lung tumor mouse model. **b** IVIS of the primary lung tumors in KLLE and KLLE-Zbtb11^Fl/Fl mice at approximately 9 weeks after Ad-Cre inhalation. **c** Ex vivo bioluminescent assays of biopsied lung tissues derived from KLLE or KLLE-Zbtb11^Fl/Fl mice showing primary lung tumor formation. **d** Quantitative analysis of primary lung tumor formation in (**c**). Data were shown as boxplots with medians, interquartile ranges and lower/upper whiskers, $n = 5$ mice per group. The $p$-values were determined by two-way ANOVA. **e** Ex vivo bioluminescent assays of biopsied

metastatic tumors in the liver, kidney, spleen and intestine derived from KLLE or KLLE-Zbtb11^Fl/Fl mice. Quantitative analysis of metastatic tumors in the liver (**f**), kidney (**g**), spleen (**h**) and intestine (**i**) based on (**e**). Data were shown as boxplots with medians, interquartile ranges and lower/upper whiskers, $n = 5$ mice per group. The $p$-values were determined by two-way ANOVA. **j** Overall survival (OS) analysis of KLLE or KLLE-Zbtb11^Fl/Fl mice with lung tumor onset induced by Ad-Cre virus inhalation. The $p$-value was determined by log-rank test. Source data are provided as a Source Data file.

of lung cancer (Supplementary Fig. 10i). Despite not being statistically significant, the high expression of both *ZBTB11* and *MMP9* exhibited a tendency to be associated with reduced DFS (Supplementary Fig. 10i). Taken together, our data provide clinical evidence that the high expression of both ZBTB11 and SET is associated with distal metastasis of lung tumors and correlates with a poor prognosis.

## Discussion

Metastasis is a complex process in which cancer cells migrate from primary tumors and colonize distal organs[70]. The migration and invasion of cancer cells into surrounding tissues or lymphatic/blood vessels represent key steps during the development of distant metastasis, which are mechanistically associated with multiple alterations, such as dysfunction of oncogenes or tumor suppressor genes, hyperactivation of metastasis-related signaling transduction, and disorder of extracellular matrix organization[71,72]. In this study, we uncovered a SET-associated transcription factor, ZBTB11, and demonstrated that

ZBTB11 is a prometastatic factor in lung cancer cells in vitro and in vivo. SET and ZBTB11 jointly promoted cancer cell metastasis, but the mechanisms could be diversified. On the one hand, SET and ZBTB11 form a stable protein complex in the nucleus, where they cooperate and reinforce each other to transcriptionally regulate a certain group of genes involved in extracellular matrix organization, a biological process closely related to cell migration and invasion. To this end, we speculated that SET may act as a transcription cofactor whose chromatin recruitment to specific genes was dependent on its interactions with ZBTB11. This notion was confirmed by investigating SET-ZBTB11 complex-mediated transcriptional activation of MMP9 (Fig. 5k–o), a key matrix metallopeptidase for extracellular matrix degradation that facilitates cancer cell migration and invasion. On the other hand, ZBTB11 may also participate in metastatic regulation in a SET-ZBTB11 complex-independent manner. To this end, ZBTB11 may link the activation of YAP1, an important trigger of metastasis[53–56], by transcriptional repression of PRRG2 (Fig. 6k–l), a transmembrane protein that

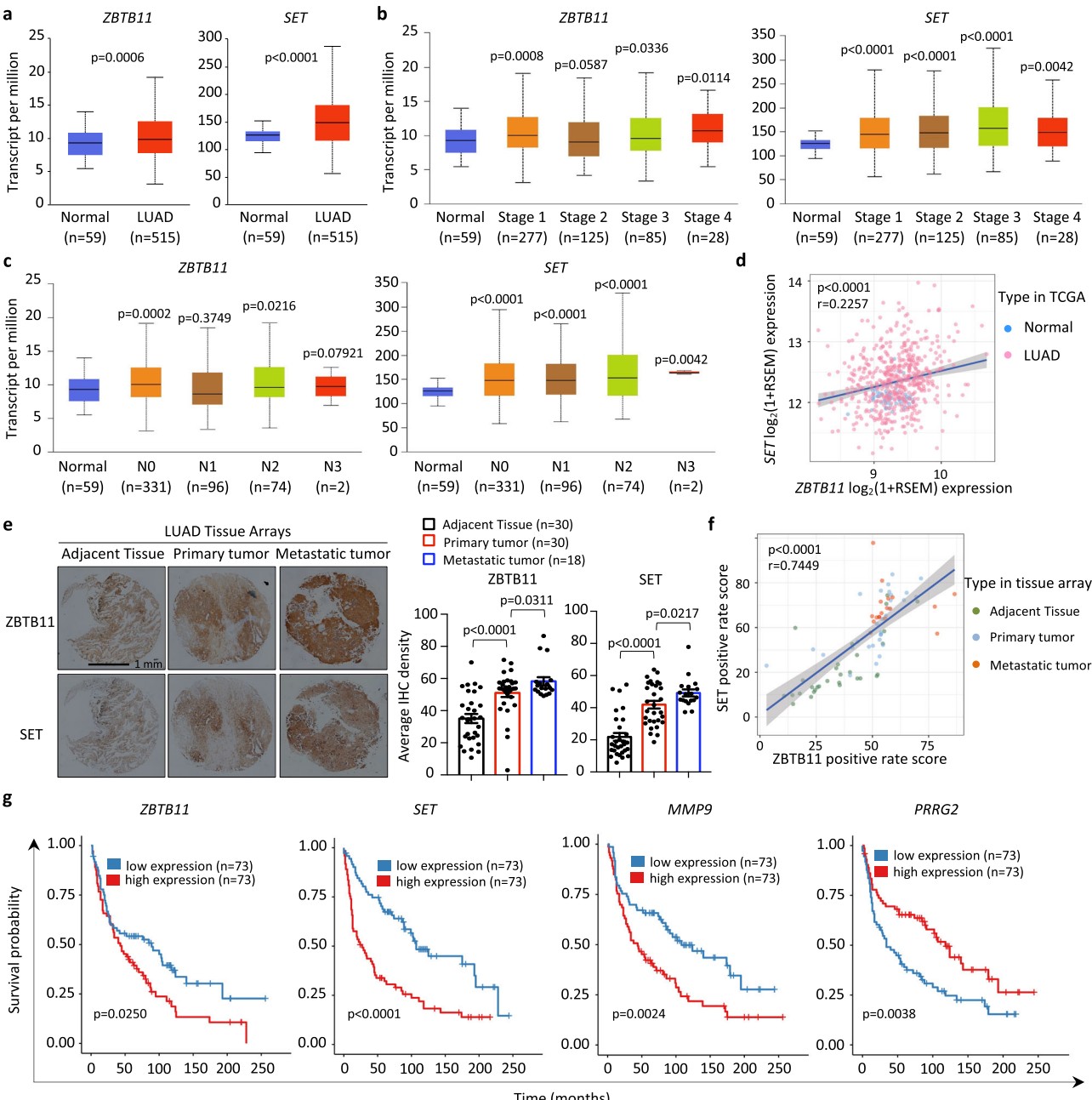

**Fig. 9 | Dysfunction of ZBTB11 and SET correlates with a poor prognosis of lung cancer. a** The positive correlation of high expression of both *ZBTB11* and *SET* with LUAD based on the TCGA database[67]. **b** The relationship of *ZBTB11* and *SET* expression with the stages of LUAD based on the TCGA database[67]. **c** The relationship of *ZBTB11* and *SET* expression with the lymph node metastatic status of LUAD based on the TCGA database[67]. N0: no regional lymph node metastasis; N1: 1–3 axillary lymph node metastases; N2: 4–9 axillary lymph node metastases; N3: ≥10 axillary lymph node metastases. Data were shown as boxplots with medians, interquartile ranges and lower/upper whiskers in (**a**–**c**). The *p*-values were determined by two-sided *t*-test. **d** The positive correlation of the expression between *ZBTB11* and *SET* in lung tissues based on TCGA database[67]. Pearson's correlation analysis was performed to determine correlation coefficients and *p*-values.

The gray band represents the 95% confidence interval band. **e** Representative and quantitative IHC of ZBTB11 and SET in LUAD tissue arrays containing primary lung tumors with paired adjacent normal lung tissues and long-distance metastatic tumors from primary LUAD. Data were shown as mean±S.E.M. The *p*-values were determined by one-sided *t*-test. **f** The positive correlation of the expression between ZBTB11 and SET in lung tissue arrays. Pearson's correlation analysis was performed to determined correlation coefficients and *p*-values. The gray band represents the 95% confidence interval band. **g** Kaplan–Meier plots of lung cancer patients stratified by *ZBTB11*, *SET*, *MMP9*, and *PRRG2* expression levels, based on GEO dataset (GSE30219)[69]. The *p*-values were determined by log-rank test. Source data are provided as a Source Data file.

physically interacts with YAP1[51,52] and profoundly influenced p-YAP1-S127 (Fig. 7g–h). Interestingly, depletion of YAP1 markedly reduced MMP9 levels (Supplementary Fig. 8a), suggesting potential mechanistic crosstalk between the SET/ZBTB11-MMP9 axis and the ZBTB11-PRRG2-YAP1 axis in ZBTB11-mediated metastatic regulation. Therefore, our study highlights *ZBTB11* as an oncogene whose dysfunction plays a profound role in promoting cancer cell metastasis through multiple mechanisms.

Dysfunction of the oncoprotein SET in lung tumors correlates with poor prognosis[13]. Targeting aberrant SET is associated with reduced migration and/or invasion in multiple cancer cells[14,16], but the mechanisms are less clear. An early study showed that cytosolic

translocation of SET induced by phosphorylation was a prerequisite for SET interaction with Rac1, a critical Rho GTPase involved in stimulating kinase-mediated signaling of cell migration[73]. Cytosolic SET is further recruited by activated Rac1 to the plasma membrane, where SET amplifies the Rac1 signaling pathway by antagonizing PP2A activity, consequently facilitating Rac1-dependent cell migration and invasion[73–75]. Complementary to the mechanisms that SET employs in metastatic regulation in the cytoplasm, our study provides an alternative explanation that SET directly controls a series of metastasis-related genes by acting as a transcriptional cofactor of ZBTB11 in the nucleus (Fig. 3f–g). This observation was reasonable since SET is primarily localized in the nucleus, where SET has been reported to bind TFs or histones for transcriptional regulation[6,17,18,76,77]. Apparently, as a cofactor, the biological functions of nuclear SET were largely dependent on which transcription factor SET was associated with. Indeed, although overexpression of both ZBTB11 and SET displayed a promotive effect on cancer cell migration and invasion (Fig. 4a–j), only full-length SET, but not acidic domain-truncated SET that failed to bind with ZBTB11, enhanced ZBTB11-dependent metastatic regulation (Supplementary Fig. 3a–c). In addition, the reduced migration and invasion by knockdown of SET in lung cancer cells was almost completely abrogated upon concomitant ZBTB11 depletion, suggesting that ZBTB11 was one of the major TFs binding by SET to exert SET-involved metastatic regulation in the nucleus. Taken together, our study underlines a critical role of nuclear SET in regulating cancer cell metastasis.

Of note, the outcomes of ZBTB11-mediated transcription may vary among cells of different origins. A study conducted in zebrafish suggested ZBTB11 as a transcriptional repressor in regulating neutrophil development, whereas another report on mESCs proposed ZBTB11 as a primary transcriptional activator in controlling mitochondrial functions[24,26]. In contrast, a recent study revealed that ZBTB11 may serve as either a transcriptional activator or repressor in fibroblasts isolated from ID patients[25]. Similarly, our data also supported the notion that ZBTB11 maintained either positive or negative transactivity toward its target genes in lung cancer cells (Fig. 2g–h). The mechanisms by which the activator or repressor activity of ZBTB11 to a given target gene is determined are unknown. In general, the regulatory consequences of target genes by a given TF are context-dependent in eukaryotic cells, which largely rely on TF-mediated recruitment of cofactors to specific target gene loci. ZBTB family proteins share a conserved BTB domain that is primarily responsible for the recruitment of transcriptional cofactors[78]. Accumulating evidence indicates that ZBTB family proteins may not only act as transcriptional repressors by recruiting corepressors such as nuclear receptor corepressor (N-CoR) or silencing mediator for retinoid and thyroid hormone receptor (SMRT)[79] but also may serve as transcriptional activators by recruiting coactivators such as p300[80]. For example, ZBTB17 (also known as MIZ1) functions as a transcriptional activator or repressor, depending on ZBTB17-mediated recruitment of its binding partners to specific promoters[80,81]. Thus, based on the structural similarity of the BTB domain among ZBTB family proteins, we speculated that target-specific transcriptional activation or repression by ZBTB11 is likely attributed to differential binding and recruitment of cofactors of ZBTB11 to its target loci. Interestingly, the genes coregulated by ZBTB11 and SET can be clearly classified into two groups (Fig. 3e), suggesting that SET always promoted, but did not antagonize, ZBTB11-mediated transcription. Consistent with this notion, we observed that SET mechanistically reinforced ZBTB11-dependent transcription of their cotarget gene MMP9 by promoting ZBTB11 recruitment to MMP9 loci (Fig. 5n–o). These findings further support the notion that the cooperative regulation of gene expression by SET and ZBTB11 could be functionally synergistic and important.

It is unclear which upstream signaling pathways may influence the interaction between SET and ZBTB11. Since the expression of SET and ZBTB11 was elevated in lung tumors, especially in distal metastasized lung tumors where SET and ZBTB11 exhibited a strong positive correlation in terms of their expression and contributed to a poor prognosis (Fig. 9a–f), we speculated that tumor progression may represent one of the selective pressures to promote SET-ZBTB11 complex formation. In addition, since the "charge effect" is critical to mediate SET-ZBTB11 interaction (Supplementary Fig. 1g), the signalings or stresses inducing posttranslational modifications (PTMs) that alter local charge status of ZBTB11 or SET (e.g., neutralization of the positive charge by acetylation of lysine residues on K/R-rich region of ZBTB11) is likely to have an impact on the interaction between SET and ZBTB11.

ZBTB11 was thought to be an essential gene in human cells[82,83], and abolishment of ZBTB11 in mESCs severely impaired cell proliferation and finally resulted in cell death[24]. In this study, we could not obtain a complete ZBTB11 knockout cancer cell line, although clones with one allele disruption were always successfully maintained after CrispR/Cas9-mediated gene editing. More importantly, the *Zbtb11*[+/−] mice were normally developed and fertile, whereas the *Zbtb11*[−/−] littermates exhibited an embryonic lethality phenotype (Supplementary Fig. 9e), demonstrating a pivotal role of ZBTB11 in maintaining cellular homeostasis in vivo in a haplosufficient manner. Notably, the extent of ZBTB11 depletion by siRNA/shRNA in cancer cells was adequate to markedly reduce cell migration and invasion but not cell proliferation (Supplementary Figs. 4a and 5g–i), supporting the capability of ZBTB11-mediated metastatic regulation of cancer cells prior to or independent of ZBTB11-mediated changes in cell proliferation. Consistent with the notions achieved from in vitro assays, loss of *Zbtb11* in a sporadic lung cancer mouse model induced by dysfunctions of the oncogene *Kras* and tumor suppressor gene *Lkb1* demonstrated a remarkable reduction in distal metastasis of lung tumors (Fig. 8e–i). Interestingly, complete abrogation of Zbtb11 also showed a tendency to limit primary lung tumor growth (Fig. 8c–d), reflecting the mechanistic diversity and complexity of ZBTB11 in regulating cancer cell metastasis in vivo. Taken together, although the characterization of ZBTB11 in cancer biology is just emerging and far from complete, our current study sheds light on a critical role of ZBTB11 as a prometastatic regulator.

## Methods

### Ethical statement

The maintenance and experimental procedures of animal studies were performed under the protocol (ACUC-A01-2019-014) approved by Institutional Animal Care and Use Committee (IACUC) of the Chinese Academy of Medical Sciences & Peking Union Medical College (CAMS & PUMC).

### Cell cultures, constructs, stable cell lines and reagents

H1299 (CRL-5803) cell line was originally purchased from ATCC; H1299-Luc2-tdT-2 (1101HUM-PUMC000645), H1975 (1101HUM-PUMC000252) and HEK293T (1101HUM-PUMC000091) cell lines were originally purchased from Cell Resource Center of IBMS-CAMS. All these cells were cultured in DMEM (Corning, 10-013-CVR) supplemented with 10% (vol/vol) FBS (Gibco, 10099141). MEFs derived from *Zbtb11*[Fl/Fl] mice were cultured in DMEM supplemented with 10% heat-inactivated FBS. The cell lines used in this project were freshly thawed from our stock and cultured for no longer than 2 months. All cell lines were negative for mycoplasma contamination.

Mammalian expression constructs, including SET, ZBTB11, MMP9, and PRRG2, were generated by cloning each cDNA into a modified pIRESneo2 vector (Clontech, 6938-1), a pCMV-myc vector (Clontech, 635689), an SFB-NT or -CT destination vector (Gifted by Dr. Wenqi Wang, University of California Irvine), a pBabe retrovirus vector (Addgene, 1764) or a pMSCV retroviral vector (Addgene, 75085). The YAP1 construct was a gift from Dr. Aifu Lin from Zhejiang University and subcloned into the pCMV-myc vector or pMSCV retroviral vector. The bacterial expression constructs were generated by cloning or

subcloning each cDNA into a PGEX-2TL vector. The luciferase constructs were generated by cloning the binding elements of ZBTB11 into the pGL3-firefly luciferase vector (Promega, E1761). The shRNA was constructed by using the pLKO.1-puro (Addgene, 8453) or pLKO.1-blast (Addgene, 26655) vector. The ZBTB11-iKO construct was generated by cloning sgRNA targeting ZBTB11 into a modified LentiCRISPR V2 construct (TLCV2, Addgene, 87360).

The stable cell lines were generated by transfecting each expressing construct into H1299 cells or by infecting H1299 cells with the indicated retrovirus, followed by selection with G418 (500 µg/ml, Sigma, H1720), puromycin (1 µg/ml, Invitrogen, A1113803) or blasticidin (5 µg/ml, Beyotime, ST018). For stable knockdown cell lines, pLKO.1-based shRNA was packaged as lentiviruses, which were then used to infect H1299 cells, followed by selection with puromycin. The corresponding empty vectors (EV) were used to generate control stable cell lines as indicated.

All transfections, including the expression construct, siRNA and shRNA, were performed by using Lipofectamine 3000 (Invitrogen, L3000015) according to the manufacturer's protocol. All siRNAs were purchased from GenePharma Co., Ltd. Doxycycline (Doxy, Sigma, D9891) was used at 1 µg/ml.

Antibodies: ZBTB11 (Bethyl, A303-240A; 2 µg for ChIP; 5 µg for ChIP-seq; 2 µg for Co-IP; 1:1000 for WB; 1:100 for IF; 1:100 for IHC); Rabbit IgG (Invitrogen, TF272445A; 2 µg for Co-IP; 5 µg for ChIP-seq); SET (Homemade; 2 µl for Co-IP; 10 µl for ChIP); SET (Bethyl, A302-262A; 1:1000 for WB; 1:100 for IHC); SET (Santa Cruz, sc-133138; 1:1000 for WB); SET (Sigma, WH0006418M1-100UG; 1:100 for IF); Flag (MBL, PM020; 1:5000 for WB); Myc (Santa Cruz, sc-40; 1:1000 for WB); HA (Roche, 11867423001; 1:2000 for WB); Vinculin (Sigma, V9131; 1:10000 for WB); YAP1 (Santa Cruz, sc-376830; 1:1000 for WB; 1:100 for IHC); MMP9 (CST, 13667 S; 1:1000 for WB); MMP9 (ABclonal, A11521; 1:100 for IHC); PRRG2 (Abcam, ab228870; 1:1000 for WB; 1:100 for IHC); p-YAP1-S127 (ABclonal, AP0489; 1:1000 for WB; 1:100 for IHC); p-YAP1-S397 (ABclonal, AP0922; 1:1000 for WB); β-actin (Proteintech, 60008-1-Ig; 1:1000 for WB); HDAC1 (Santa Cruz, sc-81598; 1:1000 for WB); Collagen IV (NOVUS, NB120-6586S; 1:100 for IHC).

The detailed information of the antibodies and the sequences of oligonucleotides are provided in Supplementary Data 2.

## Protein complex purification and mass spectrometry (MS) analysis

The nuclear fraction was prepared by sequentially lysing the cells with Buffer A (10 mM HEPES pH 7.9, 10 mM KCl, 0.1 mM EDTA, 0.1 mM EGTA, 1 mM DTT, 0.15% NP40, 1× protease inhibitor (Sigma, P8340)) and Buffer B (20 mM HEPES pH 7.9, 400 mM NaCl, 1 mM EDTA, 1 mM EGTA, 1 mM DTT, 0.5% NP40, 1× protease inhibitor). After adjusting the salt concentration to 100 mM with Buffer C (20 mM HEPES, 1× protease inhibitor), the SET-containing protein complex was tandemly precipitated with Flag M2 beads (Sigma, A2220) and HA beads (Sigma, A2095), followed by elution with 0.1% trifluoroacetic acid (TFA, Sigma, T6508-1AMP). After lyophilization, the eluents were redissolved in 1× Laemmli buffer at 95 °C for 5 min, separated by SDS−PAGE and stained with GelCode Blue reagent (Pierce, 24592). The visible bands were isolated and digested with trypsin, followed by liquid chromatography (LC) MS/MS analysis.

## Co-IP assay

Cells were lysed with NP40 buffer (50 mM Tris-HCl pH 8.0, 150 mM NaCl, 1% NP40, 1× protease inhibitor) or modified NP40 buffer (50 mM Tris-HCl pH 8.0, 150 mM NaCl, 1% NP40, 1 mM MgCl₂, 250 U/ml benzonase (Sigma, E1014), 1× protease inhibitor; aims to degrade DNA) for 30 min on ice. After centrifugation at 13,400 × $g$ for 15 min at 4 °C, the supernatant was collected for precipitation with the indicated antibodies overnight at 4 °C, followed by incubation with Protein A/G beads (Santa Cruz, sc-2003) for 2 h at 4 °C (for immobilized antibodies,

precipitate for 1 h at 4 °C and without further incubation with Protein A/G beads). The beads were then washed with BC100 buffer (20 mM Tris-HCl pH 7.3, 100 mM NaCl, 10% glycerol, 2 mM EDTA, 0.1% Triton X-100) three times, and the protein complexes were then eluted with competitive peptide, 2× Laemmli buffer or 0.1% TFA.

## Cellular fractionation

The cells were sequentially lysed on ice by Buffer A (10 mM HEPES pH 7.9, 10 mM KCl, 0.1 mM EDTA, 0.1 mM EGTA, 1 mM DTT, 0.15% NP40, 1× protease inhibitor) for 10 min and Buffer B (20 mM HEPES pH 7.9, 400 mM NaCl, 1 mM EDTA, 1 mM EGTA, 1 mM DTT, 0.5% NP40, 1× protease inhibitor) for 15 min to extract the cytoplasmic and nuclear fractions, respectively. β-Actin serves as a marker of the cytoplasmic fraction, while HDAC1 acts as a marker of the nuclear fraction.

## IF assay

The cells were fixed with 4% paraformaldehyde (PFA), followed by permeabilization with 0.1% Triton X-100. After blocking with 1% BSA, the cells were incubated sequentially with the primary and fluorophore-conjugated secondary antibodies, as indicated. The localization of the proteins of interest was visualized by confocal microscopy (Zeiss LSM780). DAPI dye (Solarbio, C0050) was used for nuclear counterstaining.

## GST pull-down assay

The *E. coli* (Tiangen, CB108) transformed with the indicated PGEX-based constructs were grown in lysogeny broth (LB) medium at 37 °C with shaking at 225 rpm. Isopropyl β-D-1-thiogalactopyranoside (IPTG, 0.1 mM) was administered for further incubation at 25 °C with shaking at 200 rpm for 4 h to induce GST or GST-fusion protein expression. After purification by GST•Bind™ Resin (Novagen, 70541), equal amounts of immobilized GST or GST-fusion proteins were incubated with purified SET or ZBTB11, as indicated, for 1 h at 4 °C. The beads were washed with BC100 buffer three times, and the binding components were eluted with 2× Laemmli buffer.

## RNA extraction, reverse transcription and quantitative PCR (qPCR)

In brief, the cells were lysed by TRIzol® reagent (Invitrogen, 15596018), followed by sequential RNA precipitation by isopropanol and ethanol. After air drying, the RNA pellet was redissolved in an appropriate volume of DNase/RNase-free H₂O. To obtain complementary DNA (cDNA), 1 µg of total RNA was reverse transcribed by using iScript™ Reverse Transcription Supermix (Bio-Rad, 1708841) according to the manufacturer's protocol. The relative expression of each gene was measured in a Bio-Rad CFX Connect™ Real-Time PCR System by using the SYBR Green method (Tiangen, FP205-02). The expression of human or mouse *β-Actin* was used as an internal control.

## RNA-seq assay

H1299 cells were transfected with control siRNA, ZBTB11-specific siRNA and/or SET-specific siRNA twice for a total of 4 days. Each sample group had two biological replicates. Total RNA was extracted using TRIzol® reagent. Before performing RNA-seq analysis, a small aliquot of each sample was analyzed by RT−qPCR to confirm ZBTB11 and SET knockdown efficiency. The RNA quality was assessed by an Agilent 2100 bioanalyzer. The sequencing libraries were generated using the NEBNext® UltraTM RNA Library Prep Kit for Illumina® (NEB, E7530L) following the manufacturer's protocol. Libraries were then sequenced using the Illumina HiSeq platform, and 125 bp/150 bp paired-end reads were generated. The index of the reference genome (hg38) was built using HISAT2 v2.0.5, and paired-end clean reads were aligned to the reference genome using HISAT2 v2.0.5. Then, featureCounts v1.5.0-p3 was used to count the read numbers mapped to each gene. Differential expression analysis of the two groups was performed

using the DESeq2 R package (1.26.0). DESeq2 provides statistical routines for determining differential expression in digital gene expression data using a model based on the negative binomial distribution. Genes with a $p < 0.05$ and an absolute value of log2 (fold change)>1 found by DESeq2 were considered differentially expressed. The GO biological process analysis was conducted using the ClusterProfiler R package (4.0.5), and a $p < 0.05$ was used as the cutoff of statistical significance.

## ChIP assay

In brief, the cells were fixed with 1% formaldehyde, followed by lysis with ChIP lysis buffer (50 mM Tris-HCl pH 8.0, 5 mM EDTA, 1% SDS, 1× protease inhibitor). After sonication, the lysates were centrifuged, and the supernatants were collected and diluted with dilution buffer (20 mM Tris-HCl pH 8.0, 2 mM EDTA, 150 mM NaCl, 1% Triton X-100, 1× protease inhibitor) at a 1:9 ratio. Precleaning of the diluted lysates was performed by salmon sperm DNA saturated protein A agarose (Millipore, 16-157). The precleaned lysates were incubated with the indicated antibodies overnight, followed by the addition of saturated Protein A agarose for another 2 h of incubation. The agarose was sequentially washed with TSE I (20 mM Tris-HCl pH 8.0, 2 mM EDTA, 150 mM NaCl, 0.1% SDS, 1% Triton X-100), TSE II (20 mM Tris-HCl pH 8.0, 2 mM EDTA, 500 mM NaCl, 0.1% SDS, 1% Triton X-100), Buffer III (10 mM Tris-HCl pH 8.0, 1 mM EDTA, 0.25 M LiCl, 1% DOC, 1% NP40), and Buffer TE (10 mM Tris-HCl pH 8.0, 1 mM EDTA). The agarose-attached protein−DNA complex was eluted with elution buffer (1% SDS, 0.1 M NaHCO3) and subjected to reverse crosslinking at 65 °C for at least 6 h. DNA was extracted using a PCR purification kit (Qiagen, 28106). Real-time PCR was performed to detect the relative enrichment of each protein to the indicated genes.

## ChIP-seq assay

The DNA samples were prepared by using a SimpleChIP® Plus Sonication Chromatin IP kit (CST, 56383) according to the manufacturer's protocol. DNA contamination and degradation were checked on agarose gels. The purity of DNA was evaluated by a NanoPhotometer® spectrophotometer (Implen), and the DNA concentration was measured by a Qubit® dsDNA HS Assay Kit (Thermo, Q32851) in Qubit® 3.0 Fluorometer (Thermo). The library was prepared by using the NEBNext Ultra II DNA Library Prep Kit for Illumina (NEB, E7645) according to the manufacturer's protocol. The quality of the library was assessed by an Agilent Bioanalyzer 2100. Pair-end sequencing of each sample was conducted on an Illumina NovaSeq 6000 platform. The raw reads were aligned to the human reference genome hg38 using Burrows Wheeler Aligner (BWA, v0.7.12). After mapping reads to the reference genome, the model-based analysis of ChIP-seq (MACS, v1.4.2) tools were used for peak calling of IgG- and ZBTB11-binding DNA elements independently under a $p$-value cutoff <0.05 and all other parameters default. Peak annotation was performed using ChIPseeker packages (1.22.1) of R with default parameters, considering the promoter region as 3 kb upstream and 3 kb downstream of the TSS. Finally, we obtained the ZBTB11-specific binding loci by comparing two peak annotation files.

## Luciferase assay

A firefly reporter containing the ZBTB11-binding element of *MMP9* or *PRRG2* loci and a Renilla control reporter were cotransfected with other expression constructs, as indicated, into H1299 cells for 48 h. The relative luciferase activity was measured by the Dual-Luciferase® Reporter Assay System according to the manufacturer's protocol (Promega, E1910).

## Proliferation assay

A total of $1 \times 10^5$ living cells were seeded into 6-well plates with a total of 3 replicates for each sample. Cell growth was monitored for 3 days by crystal violet staining. Generally, the cells were fixed with 4% paraformaldehyde (PFA) and stained with 0.1% crystal violet. The cell-

containing crystal violet was extracted by 10% acetic acid, and the relative cell number was calculated by measuring the absorption of the extracted crystal violet at OD590.

## Cell viability analysis

A total of $1 \times 10^5$ cells were seeded into 12-well plates and cultured in complete DMEM at 37 °C overnight. After washing the cells with PBS, the cells were treated with the indicated reagents (1 µM camptothecin (Cpt)) in FBS-free DMEM at 37 °C for another 24 h. CCK-8 reagent (Dojindo, CK04-500T) was then added to each well and incubated at 37 °C for 2 h according to the manufacturer's instructions. The relative cell viability was calculated by measuring OD450 (cell viability (%) = [A (treatment group) − A (blank group)]/[A (control group) − A (blank group)]×100).

## Cell cycle analysis

The cells were harvested by trypsin digestion and briefly centrifuged. $2 \times 10^5$ cells were fixed with 70% ethanol for 30 min. After removal of the fix solution and gentle washing of the cells with PBS three times, the cells were stained with 0.5 ml PBS containing 50 µg/ml PI and 200 µg/ml RNase for 30 min at 37 °C. The cell cycle was analyzed with the Beckman Coulter CytoPlus C6 platform.

## Colony formation

$10^4$ cells were seeded into a 10-cm cell culture dish and colony formation was taken for 14 days, followed by staining with crystal violet. For details, the cells were fixed with 4% paraformaldehyde (PFA) for 20 min at RT and stained with 0.1% crystal violet for 30 min at RT. The cells were gently rinsed with ddH2O 3 times, air-dried and imaged with inverted microscope (Nikon, ECLIPSE Ts2R).

## Transwell assay

Transwell migration and invasion assays were performed in transwell inserts with an 8.0-micron PET membrane (Corning, 354234). For the migration assay, cells were suspended in serum-free medium and placed in the upper chamber ($2 \times 10^4$ cells per chamber). The lower chamber contained complete growth medium. Cells were incubated for 24 h, and then the media and remaining cells on the inside of the membrane were carefully removed with a cotton-tipped applicator, while migrating cells on the outside of the membrane were fixed with 4% fixative solution (Solarbio, P1110), stained with 0.1% crystal violet and imaged using an inverted microscope (Nikon, ECLIPSE Ts2R). For the invasion assay, the protocol was modified from the migration assay in which the upper surface of the membrane was coated with 40 µl Matrigel (Corning, 354234) before placing the cells.

## Xenograft tumor growth and metastasis assay

A total of $2 \times 10^6$ living cells were mixed with Matrigel (Corning, 354248) at a 3:1 ratio for a total volume of 200 µl. The cell-Matrigel mixture was then subcutaneously injected into B-NDG (NSG) mice (6 weeks old; female; Biocytogen). After ~5 weeks, the mice were sacrificed, and the final tumor weight was measured. Whole lung tissue was isolated and incubated in 15 µg/ml D-luciferin (PerkinElmer, PN122799) in PBS and then photographed by a Xenogen IVIS Spectrum (PerkinElmer, 1400228 S). Luminescence photon flux was calculated for each lung by using the same size of the circular region of the whole lung tissue. The maintenance and experimental procedures of mice were approved by the IACUC of CAMS & PUMC. The maximum xenograft tumor size (for either length or width) permitted by IACUC of CAMS & PUMC is 2 cm. The maximal tumor size was not exceeded during the experiments.

## Genetically engineered mouse (GEM) models

*Zbtb11^Fl/Fl* mice were generated using the CRISPR/Cas9-based approach by Biocytogen Inc., Beijing. In brief, sgRNAs were designed using the

Article

CRISPR design tool (http://crispr.mit.edu) to target regions upstream or downstream of exon 2 of *Zbtb11*. A gene targeting vector containing a 5′ homologous arm, a target fragment (exon 2) and a 3′ homologous arm was used as a template to repair the double-stranded breaks (DSBs) generated by Cas9/sgRNA. The two loxP sites were precisely inserted at both sides of the target fragment of the *Zbtb11* gene. Cas9 mRNA, targeting vector and sgRNAs were coinjected into the cytoplasm of one-cell-stage fertilized C57BL/6 N eggs. The injected zygotes were transferred into the oviducts of Kunming pseudopregnant females to generate F0 mice. F0 mice with the expected genotype were mated with C57BL/6 N mice to establish germline-transmitted F1 heterozygous mice. Heterozygous loxP-flanked mice were crossed to generate homozygous loxP-flanked mice. *Kras*$^{LSL-G12D/+}$, *Lkb1*$^{Fl/Fl}$, *and Rosa26-e(CAG-LSL-Luci-EGFP)* (129:C57BL/6 J mixed background) mice were purchased from Shanghai Model Organisms Center, Inc. Lung tumorigenesis was induced using Ad-Cre ($4 \times 10^7$ PFU) via intranasal inhalation by the indicated GEMs (6 ~ 8 weeks old, male/female). The bioluminescent signals of both the primary and distant metastatic tumors were detected by Xenogen IVIS spectrum 8–9 weeks post-infection. Maintenance and experimental procedures of mice were approved by the IACUC of CAMS & PUMC. The maximum of 20% body-weight loss was considered the humane tumor endpoint and the endpoint criterion was not exceeded during the experiments.

### Tissue array

A tissue array of human lung adenocarcinoma (HLugA060PG02, T21-1815 TMA) was purchased from Shanghai Outdo Biotech Co., Ltd. Immunohistochemistry (IHC) was performed using the indicated antibodies. The tissue array was photographed by upright microscopy (Leica, DM6 B, Germany), and positive staining was calculated by the IHC profiler plugin of ImageJ (v1.8.0). The whole pictures and the detailed information of the tissue array were shown in Supplementary Fig. 11a, b.

### Immunohistochemistry (IHC)

The primary lung tumors from KLLE or KLLE-Zbtb11$^{Fl/Fl}$ mice were fixed in 4% PFA overnight, after which the tissues were embedded in paraffin and cut into serial 5 μm sections. The sections were stained with the indicated antibodies and visualized by DBA exposure. The sections were photographed by using a ZEISS Axio Scope A1 microscope.

### Survival analysis

For the public datasets, survival analysis was conducted by the survival (3.2.7) package of R using lung cancer data from GSE30219[69]. According to the quartile of the indicated gene expression level, we divided patients into a high expression group (upper quantile) and a low expression group (lower quantile). Kaplan–Meier survival curves were generated, and the survival impacts of different expression groups were compared by the log-rank test. For the survival analysis of the GEMM, lung tumorigenesis was induced using Ad-Cre ($4 \times 10^7$ PFU) via intranasal inhalation in KLLE and KLLE-Zbtb11$^{Fl/Fl}$ mice (6 ~ 8 weeks old, male/female). The overall survival (OS) of infected mice was collected and used to plot survival curves with Graphpad Prism (version 9). The difference in OS between the KLLE and KLLE-Zbtb11$^{Fl/Fl}$ groups was determined by the log-rank test.

### Statistics and reproducibility

The results were presented as the mean ± S.E.M. for bar plot, or as median, interquartile ranges, whiskers for box plot. The statistical significance was determined by using one-sided or two-sided unpaired Student's *t* test, or two-way ANOVA with Bonferroni's post-hoc test. The correlation test was conducted by the Pearson's correlation method. All statistical analyses were performed using R statistical programming. $p < 0.05$ was considered statistically significant. The representative images for immunoblot, immunofluorescence and transwell staining were shown, and each of these experiments was repeated independently for three times with similar results. The uncropped scans of all blots were provided in Source Data file.

### Reporting summary

Further information on research design is available in the Nature Portfolio Reporting Summary linked to this article.

## Data availability

RNA-seq and ChIP-seq data generated in this study has been deposited in the GEO database under accession code GSE206957 and GSE206958. The publicly available lung cancer clinical data and RNA-seq data used in this study are available in the GEO database under accession code GSE30219[69] and GSE37745[68]. The TCGA-LUAD data was obtained from UALCAN database (https://ualcan.path.uab.edu/analysis.html)[67]. The remaining data are available within the Article, Supplementary Information or Source Data file. Source data are provided with this paper.

## Code availability

The R code used for RNA-seq and ChIP-seq analysis can be accessed at https://github.com/SEO-DataInspire/ZBTB11-SET-project.git.

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

## Acknowledgements

This research was supported by the National Natural Science Foundation of China (Grant Nos. 82073132, 82122054 and 81872311 to D.W.; Grant No. 82203340 to M.Z.), the CAMS Innovation Fund for Medical Sciences (Grant Nos. 2021-I2M-1-016 and 2022-I2M-JB-003 to D.W.), the Fundamental Research Funds for the Central Universities (Grant No. 3332023147 to H.Y.), the Overseas Expertise Introduction Center for Discipline Innovation ("111 Center") (Grant No. BP0820029 to D.W.) and the State Key Laboratory Special Fund (Grant No. 2060204 to D.W.).

## Author contributions

W.X., H.Y. and D.W. conceived and designed the research; W.X. and H.Y. performed the experiments; Z.W., X.Y., Z.J., Y.L. and M.Z. contributed new reagents/analytic tools; W.X., H.Y. and D.W. analyzed and interpreted the data; W.X., H.Y. and D.W. wrote the manuscript.

## Competing interests

The authors declare no competing interests.
