## [Peer Review File · Nature Communications]

Reviewers' comments:

Reviewer #1 - Lung metastasis, mouse models, sequencing - (Remarks to the Author):

By integrating in vitro assays, tumor implantation experiments, a GEMM model, and analysis of patient samples, the authors provide evidence that transcription factor ZBTB11, a SET-interacting protein, contributes to lung adenocarcinoma (LUAD) metastasis by regulating genes that are associated with metastasis. The authors investigated genes regulated by ZBTB11.

Major concerns:

- the work does not clarify how SET and ZBTB11 interact and how these factors regulate gene transcription,
- the work does not define the effect of the main downstream target of SET-ZBTB11 on LUAD metastasis, and
- the work on “metastasis” largely relied on in vitro cell migration assays.

Specific concerns:

1. The authors must clearly describe the reason they choose to interrogate ZBTB11 among all SET-interacting proteins.
2. The authors define the domains responsible for the interaction between ZBTB11 and SET (Fig. 1h), but did not show the functional consequence of disrupting this interaction. The data in Fig. 4a does not convincingly show disruption of the interaction. The authors could use the deletion mutant constructs (Fig. 1h,i) in rescue experiments to see if this interaction mediates cell migration, metastasis, and MMP9 expression. In Fig. 1i, the linker between BTB and ZF of ZBTB11 serves as an interacting region for SET. How does this region function?
3. Fig. 5p,g showed co-occurrence of ZBTB11 and SET at the ZBTB11 binding sites of MMP9. Does the interaction between ZBTB11 and SET occur before they bind to DNA or only on the DNA?
4. There is a big discrepancy between ZBTB11 ChIP-seq peaks and ZBTB11-regulated genes (Fig. 2g). Less than 1% of ZBTB11 binding loci show changes in gene transcription. Why does ZBTB11 bind to a large number of loci without changing transcription? Fig. 2 shows that ZBTB11 is either a transcriptional activator or a repressor. How does ZBTB11 repress the transcription of its target genes?
5. Authors claim that SET and ZBTB11 activate genes implicated in ECM remodeling, yet they did not characterize the tumor stroma in their tumor implantation model (Fig. 4) or GEMM model (Fig. 7).
6. Authors claimed “this effect on cancer cell migration and invasion by SET deletion was abolished upon concomitant knockdown of ZBTB11 (Fig. 4b-e)” in p.10. Since ZBTB11 knockdown dramatically reduced migration, it looks like ZBTB11 knockdown cells nearly reached baseline and it would be difficult to see the synergistic effect of double knockdown. If SET or ZBTB11 overexpression increases cell migration and invasion, authors can try to test the effect of ZBTB11 knockdown in cells overexpressing SET, and the effect of SET knockdown in cells overexpressing ZBTB11.

7. The authors mix siRNA-mediated transient knockdown (Fig. 1-6), shRNA-mediated stable knockdown (Fig. 4h), and inducible knockout (Fig. 5r). There is no apparent or given reason for the use of these different methods.

8. The effect of stable knockdown of SET or ZBTB11 on cell proliferation and viability are not shown.

9. The paper seems to be biased for MMP among SET or ZBTB11 common targets that mediate. However, MMP9 has been long implicated in metastasis and lacks novelty. Among other SET and ZBTB11 targets, SPARC and MMP19 showed very limited effects (Fig. 5 and Extended Data Fig. 2) and the authors imply, without validation, that this is because these genes are lowly expressed. The authors also out PECAM1 without giving a reason.

10. An unclear rationale, or a clear bias, is also noticeable in the authors' decision to interrogate the SET-independent effect of ZBTB11 due to relatively strong effect of ZBTB11 knockdown on in vitro migration compared to the SET knockdown. ZBTB11 is a transcription factor that can either activate or repress gene expression. Why did the authors focus on ZBTB11-repressed genes to test the SET-independent role of ZBTB11? There are 43 ZBTB11-repressed genes and 17 ZBTB11/SET-repressed genes. How did the authors pick PRRG2 among the 26 SET-independent repressed genes?

11. Does knockdown of YAP have any effect on the expression of ZBTB11, SET, or MMP9?

12. The authors show that SET and ZBTB11 expression is associated with poor prognosis in LUAD patients (Fig. 8). Was there also a correlation with metastasis-free survival or progression-free survival in these patients?

13. Does the expression of MMP9 correlate with expression of SET or ZBTB11 in tumor samples?

Minor

14. Various statements are not properly supported by data: "SET and ZBTB11 showed the same tendency in transcriptional regulation for a given co-target (18 coactivated targets vs. 17 corepressed targets; Fig. 3e), suggesting that SET may functionally synergize with ZBTB11 for transcriptional regulation (p.9)", "knockdown of MMP19 or SPARC had no obvious effect on cell migration and invasion, most likely due to the low expression of these genes in H1299 cells (p.11)", and "More importantly, the endogenous interaction between SET and ZBTB11 was easily observed in native H1299 cells, indicating their physiological interplay in cancer cells (p. 7)"

15. Ref No.35 is about the role of MMP9 derived from endothelial cells in the premetastatic lung in lung metastasis, not the role of tumor cell-derived MMP9 in metastasis.

Reviewer #2 - SET signalling, lung cancer - (Remarks to the Author):

In this manuscript entitled "ZBTB11, a SET-associated Transcription Factor, Triggers Lung Cancer Metastasis" the authors identify and validate through a series of cell based and in vivo experiments the role of ZBTB11 and SET in regulating lung cancer development and progression. There are a number of

additional experiments and edits that in my opinion would strengthen the manuscript and make it suitable for publication. Specific comments are outlined below

1. The authors perform IP-MS to identify proteins that interact with SET and then focus specifically on one protein ZBTB11 - how many other proteins were identified in their interactive studies and what was the rationale for specifically focusing on that one protein for all subsequent studies
2. The authors should provide validation data for the specificity of the ZBTB11 antibody using either KD or KO
3. For the immunocytochemistry data presented in Figure 1 in terms of sub cellular localization of SET And ZBTB11 the authors should further validate the observation that both proteins are nuclear by performing nuclear cytoplasmic fractionation
4. Is there any structural or modeling data that would further support that the postulated sites of interaction between SET and ZBTB11 that are being proposed in this manuscript
5. For Figure 2a what time point was chosen for the RNA seq analysis as given the nature of the assay it would seem that many of the identified transcriptional changes are indirect and not direct targets of the ZBTB11 or SET complexes
6. For the CHIP-SEQ data it is not clear how the target genes/promoters identified were narrowed down to the ones that are presented and there is overlap between the putative direct transcriptional targets of ZBTB11 and changes in their transcription from the RNA seq data that they performed in Figure 2A
7. Further validation of at least a subset of the identified gene targets from their CHIP-SEQ experiments should be explored by either CHIP-qRT-PCR and promoter luciferase assays
8. For Figure 3 - the same comment as for Figure 2 - rationale for time point chosen and again many of the identified transcriptional changes by RNA seq are most likely indirect changes based upon effects on cell phenotypes rather than direct transcriptional targets
9. Does SET-ZBTB11 complex together directly on any of the target promoters that the authors suggest that they regulate and do they synergize to activate transcription and expression of target genes
10. In Figure 4A the authors state there is no effect on cell proliferation what about on the cell cycle, response to chemotherapeutic drugs and colony formation?
11. Did the authors consider doing the converse experiment instead of knocking down the target gene - did they overexposes SET or ZBTB11 or both and study the migration and invasion phenotypes
12. Did the authors attempt to CRIPSR the targets instead of just performing KD experiments?
13. The rationale for exploring PRRG2 is not well justified or explained
14. The transition to the study of the YAP/TAZ pathways is not well explained or justified
15. In figure 8 what does the expression of SET and In this manuscript entitled "ZBTB11, a SET-associated Transcription Factor, Triggers Lung Cancer Metastasis" the authors identify and validate through a series of cell based and in vivo experiments the role of ZBTB11 and SET in regulating lung cancer development

and progression. There are a number of additional experiments and edits that in my opinion would strengthen the manuscript and make it suitable for publication. Specific comments are outlined below

1. The authors perform IP-MS to identify proteins that interact with SET and then focus specifically on one protein ZBTB11 - how many other proteins were identified in their interactive studies and what was the rationale for specifically focusing on that one protein for all subsequent studies
2. The authors should provide validation data for the specificity of the ZBTB11 antibody using either KD or KO
3. For the immunocytochemistry data presented in Figure 1 in terms of sub cellular localization of SET And ZBTB11 the authors should further validate the observation that both proteins are nuclear by performing nuclear cytoplasmic fractionation
4. Is there any structural or modeling data that would further support that the postulated sites of interaction between SET and ZBTB11 that are being proposed in this manuscript
5. For Figure 2a what time point was chosen for the RNA seq analysis as given the nature of the assay it would seem that many of the identified transcriptional changes are indirect and not direct targets of the ZBTB11 or SET complexes
6. For the CHIP-SEQ data it is not clear how the target genes/promoters identified were narrowed down to the ones that are presented and there is overlap between the putative direct transcriptional targets of ZBTB11 and changes in their transcription from the RNA seq data that they performed in Figure 2A
7. Further validation of at least a subset of the identified gene targets from their CHIP-SEQ experiments should be explored by either CHIP-qRT-PCR and promoter luciferase assays
8. For Figure 3 - the same comment as for Figure 2 - rationale for time point chosen and again many of the identified transcriptional changes by RNA seq are most likely indirect changes based upon effects on cell phenotypes rather than direct transcriptional targets
9. Does SET-ZBTB11 complex together directly on any of the target promoters that the authors suggest that they regulate and do they synergize to activate transcription and expression of target genes
10. In Figure 4A the authors state there is no effect on cell proliferation what about on the cell cycle, response to chemotherapeutic drugs and colony formation?
11. Did the authors consider doing the converse experiment instead of knocking down the target gene - did they overexposes SET or ZBTB11 or both and study the migration and invasion phenotypes
12. Did the authors attempt to CRIPSR the targets instead of just performing KD experiments?
13. The rationale for exploring PRRG2 is not well justified or explained
14. In Figure 8 what did the expression of SET and ZBTB11 look like in other lung cancer subtypes like lung adenocarcinoma or small cell lung cancers

15. The clinical data and expression pattern of SET and ZBTB11 are at times inconsistent with observed phenotypes proposed by authors - how do they explain these discrepancies
16. For the survival data have the authors accounted for differences in stage, grade and treatment status when performing this analysis
17. In the patient cohorts is their co-occurrence data with over expression of SET and ZBTB11 and some of the putative targets of these genes like MMP9 and others

Reviewer #3 - HIPPO/YAP - (Remarks to the Author):

In this manuscript, the authors studied the function of SET and ZBTB11 in regulating lung cancer metastasis. Through a tandem affinity purification and mass spectrometry, they identified ZBTB11 as a novel SET binding protein in lung cancer cells. They further showed that SET cooperated with ZBTB11 in transcriptional regulation of a set of genes involved in extracellular matrix organization. Through regulating these genes, such as MMP9, SET and ZBTB11 promote cancer cell migration and invasion. Interestingly, the study also found that ZBTB11 may also regulate the YAP signaling pathway through a SET-independent manner in regulating metastasis. Overall, the study revealed a previously unknown function of ZBTB11 as a metastasis promoter in lung cancer and could contribute to our understanding of the underlying mechanisms of lung cancer malignant progression. The manuscript was well written with clear presentation of the results; however, it should be further improved by addressing the following concerns.

Major concerns:

1. Although two mouse models have been employed to demonstrate the essential role of ZBTB11 in lung cancer metastasis, these models have not been utilized to confirm if the two main mechanistic effectors of ZBTB11, including MMP9 and PRRG2-YAP, are involved in metastasis. Therefore, if these effectors found in vitro also exist in vivo is still unclear.
2. The study indicated that both SET and ZBTB11 are metastasis promoters. However, all major functional tests have focused on loss of function studies through depletion of these genes. While these tests showed the requirement of them, they cannot be the sole evidence to support their promotive roles.
3. The relationship between SET and ZBTB11 is confusing. The authors used the results shown in Figure 4b-e to support the notion that "SET hierarchically resides upstream of ZBTB11". However, this evidence is vague. To support this point, the study needs to examine if SET and/or ZBTB11 overexpression can promote cell migration, invasion, and metastasis (as indicated in the above concern), followed by examining if the effects by SET overexpression can be diminished by ZBTB11 depletion, and vice versa.
4. The rationale for studying PRRG2 as a ZBTB11 target metastasis gene is unclear. How was PRRG2 noticed among the 119 putative ZBTB11 direct target genes (Figure 2g)? PRRG4 is a metastasis-promoting gene, why PRRG2 has a metastasis inhibitory role? Another major concern regarding PRRG2 is its link to YAP. Although the study confirmed the interaction between PRRG2 and YAP as reported previously, there were no further mechanistic study to examine the effect of their interaction on YAP

function in regulating metastasis. Therefore, it is quite premature to conclude that the PRRG2-YAP signaling is an effector of ZBTB11.

Minor concerns:

1. In Figure 1i, the author stated, “we also mapped that the regions of ZBTB11 containing amino acids 313-568 and amino acids 569-937 were critical for interacting with SET (Fig. 1i)”. However, the result showed that aa 313-568 is sufficient, but aa 569-937 is not required for the interaction. Please clarify this part.
2. In Figure 2d, it was unknown how the TSS were defined in this study. The method for defining TSS should be provided.
3. The effect shown in Extended Data Fig. 3d-g should be confirmed in another cell line. In Figure 3g, these results should be validated using additional cell lines.
4. In Figure 5g-j, and 5i-o, when overexpressing MMP9, why ZBTB11 KD or SET KD can still inhibit migration and invasion? Does this indicate that other factors are also involved in the effects?
5. Based on the results and model shown in Figure 5s-t, SET can promote ZBTB11 function. Can the authors provide certain mechanism for this effect?
6. In Figure 7c-f, how were the batch 1 and 2 defined? Why not put them together for analysis? What were the statistics test used in Figure 7d and 7f? It appeared that the numbers of individuals in each genotype group are too small for a statistic study. Can the authors examine liver, kidney, and spleen individually other than grouping them together?
7. Why the organs with metastatic tumors in human (Extended Data Fig. 5b) have no overlap with those in the mouse model (Figure 7e)? Does this raise a concern regarding feasibility of the mouse model?
8. Molecular weight markers should be indicated with all western blotting results to indicate the relative size of each detected protein.

Point-by-Point Response

Ref: ZBTB11, a SET-associated Transcription Factor, Triggers Lung Cancer Metastasis (Manuscript #: NCOMMS-22-29298A-Z)

Reviewer #1: Lung metastasis, mouse models, sequencing

By integrating in vitro assays, tumor implantation experiments, a GEMM model, and analysis of patient samples, the authors provide evidence that transcription factor ZBTB11, a SET-interacting protein, contributes to lung adenocarcinoma (LUAD) metastasis by regulating genes that are associated with metastasis. The authors investigated genes regulated by ZBTB11.

Major points: 1) the work does not clarify how SET and ZBTB11 interact and how these factors regulate gene transcription; 2) the work does not define the effect of the main downstream target of SET-ZBTB11 on LUAD metastasis, 3) and the work on “metastasis” largely relied on in vitro cell migration assays.

Response: We sincerely appreciate the reviewer for his/her efforts to carefully evaluate our manuscript and take very seriously on the reviewer’s comments, which are very insightful and constructive for us to improve our study. To address the reviewer’s concerns and strengthen our manuscript, we have performed a large number of experiments and provided more detailed explanations and discussions to support our conclusions. Particularly, for the three major concerns raised by the reviewer, in this revised manuscript, we **1)** provide a mechanistic explanation of how SET interacts with ZBTB11 and regulates ZBTB11-mediated transcription; **2)** evaluate the implications of SET/ZBTB11 downstream targets, including MMP9 and PRRG2, in LUAD by using GEMM-based tumor samples; and **3)** improve the analyses of GEMM-based lung tumor metastasis to further validate ZBTB11-mediated metastatic regulation *in vivo*. Detailed point-by-point responses to the reviewers’ comments are shown below.

Specific points:

1. The authors must clearly describe the reason they choose to interrogate ZBTB11 among all SET-interacting proteins.

Response: We appreciate the reviewer for this constructive comment. We identified 66 proteins from control purification (H1299-EV) and 225 proteins from H1299-F-HA-SET purification (**Revised Manuscript Extended Data Fig. 1a and Extended Data Tab. 1**). After excluding 36 proteins shared by these two purifications, we focused on the remaining 189 proteins that represent potential SET-specific binding partners. Of note, TRIM28, a known SET direct binding protein in the nucleus, was ranked No. 1 among the listed proteins (**Revised Manuscript Extended**

Data Fig. 1b), suggesting that our MS results can reflect the cellular profiles of SET-binding proteins. Since we particularly focused on SET-mediated transcriptional regulation, we noticed that ZBTB11, a putative transcription factor, was enriched as a major SET-binding protein with top rank No. 10 (**Revised Manuscript Extended Data Fig. 1b**) and even ranked higher than CBX3 (No. 30) and histone H1 (No. 35), two other known direct binding proteins of SET (**Revised Manuscript Extended Data Tab. 1**). Taken together, based on our TAP-MS analysis, we found that ZBTB11 is a major binding protein of SET in the nucleus, and we speculated that the interaction between SET and ZBTB11 could be implicated in cellular functional regulation.

2. The authors define the domains responsible for the interaction between ZBTB11 and SET (Fig. 1h), but did not show the functional consequence of disrupting this interaction. The data in Fig. 4a does not convincingly show disruption of the interaction. The authors could use the deletion mutant constructs (Fig. 1h,i) in rescue experiments to see if this interaction mediates cell migration, metastasis, and MMP9 expression. In Fig. 1i, the linker between BTB and ZF of ZBTB11 serves as an interacting region for SET. How does this region function?

Response: We greatly appreciate the reviewer for these constructive suggestions. ZBTB11 overexpression markedly promoted cancer cell migration and invasion, and these changes were largely blocked by concomitant depletion of SET (**Revised Manuscript Fig. 4f-j**). These data support the notion that SET is functionally involved in ZBTB11-mediated regulation of cancer cell metastatic behaviors. To investigate the role of the SET-ZBTB11 interaction in the metastatic regulation of cancer cells, we evaluated whether re-expression of RNAi-resistant full-length SET (re-FH-SET-FL) or acidic domain-deleted SET (re-FH-SET- Δ AD) is able to rescue the reduced cell migration and invasion caused by depletion of endogenous SET in stable H1299 ZBTB11-SFB cells (**Revised Manuscript Fig. 4k**). As shown in **Revised Manuscript Fig. 4l-m**, re-expression of SET-FL, but not SET- Δ AD, which does not bind with ZBTB11, obviously elevated cell migration and invasion, suggesting that the interaction between SET and ZBTB11 critically contributes to SET-enhanced, ZBTB11-mediated cancer cell metastatic regulation. Accordingly, we also found that ZBTB11-mediated transcriptional activation of *MMP9* can be enhanced by SET-FL but not SET- Δ AD (**Revised Manuscript Extended Data Fig. 5l**).

The linker region of ZBTB family proteins is unstructured, which is often targeted for posttranslational modifications (PTMs) and mediates protein-protein interactions (**Maeda et al. International Journal of Hematology 2016, PMID: 27250345**). For example, BCL6, a ZBTB family protein, has been reported to interact with Mi-2 and MTA3 by using its linker region (**Fujita et al. Cell 2004, PMID: 15454082**). Consistent with this notion, our data revealed that the linker region (313-568 aa) of ZBTB11 is involved in mediating its physical interaction with the acidic domain (AD) of SET (**Revised Manuscript Fig. 1i-j**). To explore why the linker region of ZBTB11

binds with SET, we further identified 522-561 aa within the ZBTB11 linker region as a K/R-rich region conserving a positive charge, which is particularly critical for binding with the SET acidic domain, which is enriched with a negative charge (**Revised Manuscript Extended Data Fig. 1d-e**). More importantly, loss of the positive charge of the linker region completely abrogated its ability to bind with SET (**Revised Manuscript Extended Data Fig. 1f**). Together, our data reveal the “charge effect” as a physical basis for mediating the SET-ZBTB11 interaction, where the negative charge within the acidic domain of SET attracts the positive charge within the linker region of ZBTB11 (**Revised Manuscript Extended Data Fig. 1g**).

3. Fig. 5p, g showed co-occurrence of ZBTB11 and SET at the ZBTB11 binding sites of MMP9. Does the interaction between ZBTB11 and SET occur before they bind to DNA or only on the DNA?

Response: We thank the reviewer for this insightful comment. After we identified ZBTB11 as a SET-binding partner from the TAP-MS assay, we further evaluated their endogenous interaction in the soluble nuclear fraction (without chromatin fraction) by a Co-IP assay in H1299 cells. **As shown in Revised Manuscript Fig. 1d-e**, our data indicated that there is an endogenous interaction between SET and ZBTB11 in a soluble nuclear fraction, suggesting that DNA is dispensable for SET-ZBTB11 complex formation. To further validate this point, we performed a Co-IP assay in H1299 cellular lysates treated with benzonase (250 U/ml) to completely degrade DNA. Under such conditions, we still observed the interaction between endogenous SET and ZBTB11 (**Revised Manuscript Extended Data Fig. 1c**). Taken together, our data support that DNA is not necessary for the occurrence of the SET-ZBTB11 interaction.

4. There is a big discrepancy between ZBTB11 ChIP-seq peaks and ZBTB11-regulated genes (Fig. 2g). Less than 1% of ZBTB11 binding loci show changes in gene transcription. Why does ZBTB11 bind to a large number of loci without changing transcription? Fig. 2 shows that ZBTB11 is either a transcriptional activator or a repressor. How does ZBTB11 repress the transcription of its target genes?

Response: We greatly appreciate the reviewer for these critical and insightful comments. The reviewer was right. We also noticed that in our study, only 0.59% (119 out of 20336) of ZBTB11-binding loci was linked to transcriptional regulation. Since, to our knowledge, our study by ChIP-seq assay was the first to attempt to identify ZBTB11 target genes in human cancer cells, we aimed to screen potential ZBTB11 target genes as much as possible for subsequent investigation. Thus, we employed a general but not extremely stringent threshold to perform peak-calling through the MACS tool (p value cutoff <0.05 with other parameters default). Although this modest criterion can yield, in relatively, more binding loci, the results were still highly reliable. For example, along the ChIP-seq analysis, we easily tested a group of

candidate genes by ChIP-qPCR and successfully validated ZBTB11 binding to their promoters in cancer cells (**Revised Manuscript Extended Data Fig. 2a-h**). More importantly, this strategy did not affect subsequent functional and mechanistic studies, as evidenced by the identification of *MMP9* and *PRRG2* as *bona fide* target genes of ZBTB11 by ChIP-qPCR and luciferase assays (**Revised Manuscript Fig. 5p-t and Fig. 6k-l**).

To further evaluate the profiles of ZBTB11-mediated transcriptional regulation, we applied exactly the same strategy and threshold to analyze the genome-wide distribution of ZBTB11 and ZBTB11-mediated transcriptional alteration in mouse embryonic stem cells (mESCs) based on published ChIP-seq and RNA-seq datasets (**Wilson et al. Nature Communications, 2020; PMID: 33122634**). We still found a relatively low percentage of ZBTB11-binding loci associated with transcriptional changes (1.27%, 87 out of 6825). Moreover, the binding motif of ZBTB11 revealed by our ChIP-seq analysis was almost completely identical to that uncovered in the published study mentioned above (**Rebuttal Fig. 1**), further supporting that our ChIP-seq results indeed reflect the DNA-binding profiles of ZBTB11. Although the detailed profiles of the genome-wide distribution of ZBTB11 as well as ZBTB11-mediated transcriptional regulation may vary depending on cell type, tissue of origin or even species, the analyses based on both our study and published datasets prompt us to speculate that such unexpected binding profiles of ZBTB11 (fewer binding loci show changes in gene transcription) might reflect intrinsic characteristics of ZBTB11 in terms of its molecular functions by acting as a chromatin/DNA-binding protein. For example, a recent study showed that the chromatin association of transcription factors (TFs) may serve as chromatin architectural regulators that are responsible for high-order chromatin structure organization (e.g., MyoD, **Wang et al. Nature Communications, 2022, PMID: 35017543**). This evidence implies that chromatin/DNA-binding of TFs could be functionally diversified and beyond traditionally recognized transcriptional regulation. Although it is currently unclear whether ZBTB11 works similarly, the reviewer's comments inspire us and open an interesting direction for further biological studies on ZBTB11 in high-order chromatin structural regulation.

Rebuttal Fig. 1 DNA-binding motif of ZBTB11. Left: DNA-binding motif of ZBTB11 identified in the current study. Right: DNA-binding motif of ZBTB11 identified by Wilson et al. published in

For the reviewer's second comment, we indeed found that ZBTB11 may serve as either transcriptional activator (e.g., for transcriptionally activating *MMP9*) or repressor (e.g., for transcriptionally repressing *PRRG2*) in cancer cells. Generally, the regulatory consequences of specific target genes by a given TF are context-dependent in eukaryotic cells, which largely rely on TF-mediated differential recruitment of cofactors to target gene promoters or other regulatory elements. ZBTB family proteins share a conserved BTB domain that is primarily responsible for the recruitment of transcriptional cofactors. Accumulating evidence indicates that ZBTB family proteins may not only act as transcriptional repressors by recruiting corepressors such as N-CoR (nuclear receptor corepressor) or SMRT (silencing mediator for retinoid and thyroid hormone receptor) (Huynh et al. *Oncogene*, 1998, PMID: 9824158), but may also serve as transcriptional activators by recruiting coactivators such as p300 (Staller et al. *Nature Cell Biology*, 2001, PMID: 11283613). For example, ZBTB17 (also known as MIZ1) functions as a transcriptional activator or repressor, depending on ZBTB17-mediated recruitment of its binding partners to specific promoters (Schneider et al. *Curr Top Microbiol Immunol.*, 1997, PMID: 9308237; Staller et al. *Nature Cell Biology*, 2001, PMID: 11283613). Thus, due to the structural similarity of the BTB domain among ZBTB family proteins, it is most likely that ZBTB11 functions as a transcriptional activator for a subgroup of target genes by binding and recruiting coactivators on their promoters or other regulatory elements. Taken together, although the mechanisms by which ZBTB11 selectively regulates specific target genes need further investigation, our current data reveal that ZBTB11 serves as a *bona fide* transcription factor in cancer cells.

5. Authors claim that SET and ZBTB11 activate genes implicated in ECM remodeling, yet they did not characterize the tumor stroma in their tumor implantation model (Fig. 4) or GEMM model (Fig. 7).

Response: We appreciate the reviewer for this constructive comment. MMP9 serves as a key enzyme that is responsible for degrading the extracellular matrix, including collagen IV, to promote cell invasion and migration. In our study, we identified *MMP9* as a cotarget gene of SET and ZBTB11, where SET facilitated ZBTB11-dependent transcriptional activation of *MMP9* by acting as a cofactor. Therefore, we used collagen IV as a marker to evaluate whether loss of ZBTB11 contributed to ECM remodeling in the GEMM tumor model, which may reflect a physiological condition in terms of SET/ZBTB11-mediated MMP9 regulation. As shown in **Revised Manuscript Extended Data Fig. 8a**, loss of ZBTB11 increased the collagen IV levels surrounding tumor cells, supporting that ZBTB11 is physiologically involved in ECM remodeling during tumor progression.

6. Authors claimed "this effect on cancer cell migration and invasion by SET deletion was abolished upon concomitant knockdown of ZBTB11 (Fig. 4b-e)" in p.10. Since

ZBTB11 knockdown dramatically reduced migration, it looks like ZBTB11 knockdown cells nearly reached baseline and it would be difficult to see the synergistic effect of double knockdown. If SET or ZBTB11 overexpression increases cell migration and invasion, authors can try to test the effect of ZBTB11 knockdown in cells overexpressing SET, and the effect of SET knockdown in cells overexpressing ZBTB11.

Response: These points were well taken. We greatly appreciate the reviewer for these constructive and critical suggestions. First, we evaluated cell migration and invasion in control or Flag-HA tagged SET-overexpressed stable cells (H1299-EV vs. H1299-FH-SET) depleted with or without ZBTB11 (**Revised Manuscript Fig. 4a**). As expected, knockdown of ZBTB11 markedly reduced cell migration and invasion (**Revised Manuscript Fig. 4b-e**). SET overexpression modestly increased cell migration and invasion, and more importantly, this effect was largely rescued by concomitant knockdown of ZBTB11 (**Revised Manuscript Fig. 4b-e**).

Next, we generated an H1299 control or SFB-tagged ZBTB11 stable cell line (H1299 EV vs. H1299 ZBTB11-SFB). Endogenous SET was depleted by siRNA in H1299 EV or ZBTB11-SFB cells to evaluate the potential effects of the interplay between SET and ZBTB11 on cell migration and invasion (**Revised Manuscript Fig. 4f**). Indeed, SET depletion decreased cell migration and invasion (**Revised Manuscript Fig. 4g-j**). Compared with SET overexpression-mediated modest increases in cell migration and invasion, ZBTB11 overexpression dramatically boosted cell migration and invasion (**Revised Manuscript Fig. 4g-j**). Notably, knockdown of SET significantly reduced ZBTB11 overexpression-induced upregulation of both cell migration and invasion (**Revised Manuscript Fig. 4g-j**). Taken together, our data indicate that the interplay between SET and ZBTB11 is critically involved in regulating cancer cell migration and invasion.

7. The authors mix siRNA-mediated transient knockdown (Fig. 1-6), shRNA-mediated stable knockdown (Fig. 4h), and inducible knockout (Fig. 5r). There is no apparent or given reason for the use of these different methods.

Response: We thank the reviewer for this comment. The reviewer was right, and we indeed used multiple methods for knockdown assays. Here, we would like to explain why we chose different methods in specific assays. First, we employed siRNA in *in vitro* assays such as RNA-seq, cell migration and invasion, since these assays only took a few days and the transient transfection of siRNA exhibited a good knockdown efficiency that is sufficient to elicit the biological effects (**Revised Manuscript Fig. 2-6**).

Next, for the *in vivo* xenograft tumor growth and metastasis assay, the experiment took 5-6 weeks, and transient knockdown by siRNA was expected to not successfully maintain a desired knockdown effect during such a long period. Therefore, we used a

shRNA strategy to generate the indicated stable knockdown cell lines for this *in vivo* assay (**Revised Manuscript Fig. 4s-v**).

Finally, to evaluate whether the recruitment of SET to *MMP9* loci is dependent on ZBTB11, we performed a ChIP assay in H1299 cells with or without ZBTB11 depletion. To increase the sensitivity of the downstream ChIP-qPCR assay, a relatively large number of cells is recommended for the initial experimental setup. To facilitate cell preparation and maintain a high ZBTB11 knockdown efficiency in a timely and economical manner, we used our ZBTB11 inducible knockout (ZBTB11-iKO) cells, in which depletion of ZBTB11 could be easily achieved by doxycycline (Doxy) administration (**Revised Manuscript Fig. 5r**). As a result, we observed a very decent decrease in SET enrichment at *MMP9* loci upon ZBTB11 depletion by using this iKO method.

8. The effect of stable knockdown of SET or ZBTB11 on cell proliferation and viability are not shown.

Response: These points were well taken. We thank the reviewer for these comments. As shown in **Revised Manuscript Extended Data Fig. 4f**, the trypan blue staining assay showed that stable knockdown of SET and/or ZBTB11 had no effect on cell viability. In addition, we monitored cell proliferation by crystal violet staining for three days. Again, stable knockdown of SET and/or ZBTB11 exhibited no obvious changes in cell proliferation (**Revised Manuscript Extended Data Fig. 4g**).

9. The paper seems to be biased for MMP among SET or ZBTB11 common targets that mediate. However, MMP9 has been long implicated in metastasis and lacks novelty. Among other SET and ZBTB11 targets, SPARC and MMP19 showed very limited effects (Fig. 5 and Extended Data Fig. 2) and the authors imply, without validation, that this is because these genes are lowly expressed. The authors also out PECAM1 without giving a reason.

Response: We sincerely accept the reviewer's comments and apologize for our unclear description of why we focused on MMP9 for extensive analysis during mechanistic study in our previous version of manuscript. First, we agree with the reviewer that multiple downstream targets may contribute together to SET-ZBTB11 complex-mediated metastatic regulation. To this end, we paid more attention to SET/ZBTB11 coregulated target genes that are potentially involved in modulating metastasis-related biological processes. Combining analyses of our ChIP-seq and RNA-seq datasets, we noticed that SET/ZBTB11 coregulated genes were highly enriched in the regulation of extracellular matrix organization, a critical biological process related to cancer cell migration and invasion. Therefore, we evaluated whether altered expression of these genes contributed to SET/ZBTB11 complex-mediated regulation of cancer cell migration and invasion.

Among this group of genes, downregulation of *MMP9*, *MMP19*, *SPARC* and *PECAM1* by both SET and ZBTB11 was likely to explain SET/ZBTB11 complex-regulated cancer cell migration and invasion (**Revised Manuscript Fig. 3g and Extended Data Fig. 2i**). However, functional analysis by knocking down *MMP19* or *SPARC* in H1299 cells showed no obvious changes in cell migration and invasion (**Revised Manuscript Extended Data Fig. 5b-c**), suggesting that *MMP19* and *SPARC* might not be the major effectors in SET/ZBTB11 complex-regulated cell migration and invasion, at least in H1299 lung cancer cells. In the previous version of the manuscript, we speculated that one explanation could be the relatively low basal expression of *MMP19* and *SPARC* in H1299 cells, based on our RNA-seq dataset (**average RPKM for *MMP19* and *SPARC* are 0.25 and 1.17, respectively**). Since this speculation was preliminary, we have removed this description in this revised manuscript. In contrast, we found that knockdown of *MMP9* or *PECAM1* in both H1299 and H1975 lung cancer cell lines significantly repressed cell migration and invasion. Thus, it is most likely that both *MMP9* and *PECAM1* work together to modulate metastatic behavior regulated by the SET-ZBTB11 complex.

MMP9 is well studied in promoting metastasis by degrading the extracellular matrix, while *PECAM1* is mainly expressed on vascular endothelial cells and contributes to intercellular junctions between endothelial cells (**Kessenbrock et al. Cell 2010, PMID: 20371345; Privratsky et al. Cell Tissue Res. 2014, PMID: 24435645**). It has been reported that *PECAM1* may act as a substrate of *MMP9*, which cleaves the extracellular portion of *PECAM1* and facilitates leukocyte migration (**Kato et al. J Hepatol. 2014, PMID: 24412604**). Thus, SET-ZBTB11 complex-mediated transcriptional activation of *MMP9* is likely involved in the functional regulation of *PECAM1* in certain biological contexts. Therefore, we focused on *MMP9* for further study and investigated whether *MMP9* serves as an effector that directly contributes to SET-ZBTB11 complex-mediated metastatic regulation. In addition, since we aimed to further investigate the hierarchical relationship between SET and ZBTB11 in terms of their regulation of their cotarget genes, we chose *MMP9*, a classical gene involved in metastatic regulation, as an example to pursue mechanistic analysis. As shown in **Revised Manuscript Fig. 5p-t**, we indeed proved that SET serves as a cofactor of ZBTB11 to facilitate ZBTB11-mediated transcriptional regulation of *MMP9*. Although *MMP9* has long been implicated in metastasis, our study revealed a previously unrecognized SET/ZBTB11-*MMP9* regulatory axis in modulating lung cancer cell migration and invasion. Taken together, our data support the notion that the SET-ZBTB11 complex regulates cancer cell metastatic behaviors via multiple cotargets, including *MMP9* and *PECAM1*, and we prove that SET acts as a cofactor of ZBTB11 for their transcriptional regulation.

10. An unclear rationale, or a clear bias, is also noticeable in the authors' decision to interrogate the SET-independent effect of ZBTB11 due to relatively strong effect of ZBTB11 knockdown on in vitro migration compared to the SET knockdown. ZBTB11 is a transcription factor that can either activate or repress gene expression.

Why did the authors focus on ZBTB11-repressed genes to test the SET-independent role of ZBTB11? There are 43 ZBTB11-repressed genes and 17 ZBTB11/SET-repressed genes. How did the authors pick PRRG2 among the 26 SET-independent repressed genes?

Response: We thank the reviewer for this constructive comment. As the reviewer mentioned, we observed a stronger effect on cell migration and invasion upon ZBTB11 knockdown than upon SET knockdown. Therefore, we speculated that there could be extra mechanisms accounting for ZBTB11-mediated metastatic regulation that are independent of the SET-ZBTB11 complex. To this end, we retrieved our RNA-seq/ChIP-seq dataset for the targets regulated by ZBTB11 alone but not by the SET-ZBTB11 complex, including both ZBTB11-activated and ZBTB11-repressive targets. Among these 84 ZBTB11 alone-regulated target genes (26 repressive targets vs. 58 activated targets), we noticed that, considering both fold change and p value, *PRRG2* conserved the most significant expressional changes upon ZBTB11 knockdown (**Revised Manuscript Extended Data Fig. 6a**). In addition, *PRRG4*, a family member of *PRRG2*, has been reported to regulate cell metastatic behaviors (**Zhang et al. *Oncogene* 2020, PMID: 33037408**). Therefore, we investigated whether *PRRG2* is involved in ZBTB11-mediated cell migration and/or invasion. Of note, although we proved our hypothesis that *PRRG2* contributes to ZBTB11-mediated regulation of cell migration and invasion in a SET/ZBTB11 complex-independent manner (**Revised Manuscript Fig. 6f-j**), we cannot exclude the possibility that other SET-independent ZBTB11 targets also participate in this regulation.

11. Does knockdown of YAP have any effect on the expression of ZBTB11, SET, or MMP9?

Response: We appreciate the reviewer for this constructive question. Following the reviewer's suggestion, we detected ZBTB11, SET and MMP9 expression in H1299 cells upon YAP1 knockdown. As shown in **Revised Manuscript Extended Data Fig. 6m**, we found that depletion of YAP1 reduced the expression of MMP9 but not ZBTB11 and SET. This result of YAP1 in MMP9 regulation is consistent with previous findings (**Zhang et al. *Mol Cancer*, 2019; PMID: 31526394**) and further supports the notion that YAP1 participates in ECM remodeling.

12. The authors show that SET and ZBTB11 expression is associated with poor prognosis in LUAD patients (Fig. 8). Was there also a correlation with metastasis-free survival or progression-free survival in these patients?

Response: We appreciate the reviewer for this constructive comment. By analyzing the same GEO dataset (GSE30219), we found that high expression of *SET* or low expression of *PRRG2* was well correlated with reduced disease-free survival in LUAD patients (**Revised Manuscript Extended Data Fig. 8i**). In addition, despite

not being statistically significant, the high expression of both *ZBTB11* and *MMP9* also displayed a tendency to be associated with reduced disease-free survival (**Revised Manuscript Extended Data Fig. 8i**). Taken together, in combination with the analyses of both overall survival and disease-free survival in LUAD cohorts, our data support the notion that dysfunction of *ZBTB11* and/or *SET* is associated with poor prognosis in LUAD patients.

13. Does the expression of *MMP9* correlate with expression of *SET* or *ZBTB11* in tumor samples?

Response: We thank the reviewer for this comment. We analyzed the LUAD dataset (GSE30219) to evaluate the relationship between *MMP9* and *SET* or *ZBTB11*. As shown in **Revised Manuscript Extended Data Fig. 8g**, we observed a positive correlation between *MMP9* and *SET* expression. However, we did not find an obvious correlation between *MMP9* and *ZBTB11* (**Revised Manuscript Extended Data Fig. 8g**). Since the expression of *MMP9* can be transcriptionally regulated by *SET* in a *ZBTB11*-dependent manner, these correlations might reflect that the overexpression of *SET* is sufficient to activate *ZBTB11*-mediated *MMP9* transcriptional upregulation by acting as a cofactor.

Minor points:

14. Various statements are not properly supported by data: “*SET* and *ZBTB11* showed the same tendency in transcriptional regulation for a given co-target (18 coactivated targets vs. 17 corepressed targets; Fig. 3e), suggesting that *SET* may functionally synergize with *ZBTB11* for transcriptional regulation (p.9)”, “knockdown of *MMP19* or *SPARC* had no obvious effect on cell migration and invasion, most likely due to the low expression of these genes in H1299 cells (p.11)”, and “More importantly, the endogenous interaction between *SET* and *ZBTB11* was easily observed in native H1299 cells, indicating their physiological interplay in cancer cells (p. 7)”

Response: These points were well taken. We thank the reviewer for his/her efforts to help us improve the manuscript. We made appropriate modifications in the text to ascertain our description of the results more clearly and precisely. In addition, we also included necessary controls to strengthen our manuscript. All changes are noted with blue font in this revised version of manuscript.

15. Ref No.35 is about the role of *MMP9* derived from endothelial cells in the premetastatic lung in lung metastasis, not the role of tumor cell-derived *MMP9* in metastasis.

Response: This point was taken. We thank the reviewer for pointing out our inaccuracy. We have now cited the appropriate literature and updated our reference list (**refs. 42 and 44** in this revised manuscript).

Reviewer #2: SET signaling, lung cancer

In this manuscript entitled "ZBTB11, a SET-associated Transcription Factor, Triggers Lung Cancer Metastasis" the authors identify and validate through a series of cell based and in vivo experiments the role of ZBTB11 and SET in regulating lung cancer development and progression. There are a number of additional experiments and edits that in my opinion would strengthen the manuscript and make it suitable for publication.

Response: We appreciate the great efforts of the reviewer to help us improve this manuscript. The insightful comments prompted us to conduct further studies that have significantly strengthened the paper.

Specific points:

1. The authors perform IP-MS to identify proteins that interact with SET and then focus specifically on one protein ZBTB11 - how many other proteins were identified in their interactive studies and what was the rationale for specifically focusing on that one protein for all subsequent studies.

Response: We appreciate the reviewer for this constructive comment. We identified 66 proteins from control purification (H1299-EV) and 225 proteins from H1299-FH-SET purification (**Revised Manuscript Extended Data Fig. 1a and Extended Data Tab. 1**). After excluding 36 proteins shared by these two purifications, we focused on the remaining 189 proteins that represent potential SET-specific binding partners. Notably, TRIM28, a known SET direct binding protein in the nucleus, was ranked No. 1 among the listed proteins (**Revised Manuscript Extended Data Fig. 1b**), suggesting that our MS result indeed reflects the cellular profiles of the SET-binding proteins. Since we particularly focused on SET-mediated transcriptional regulation, we noticed that ZBTB11, a putative transcription factor, was enriched as a major SET-binding protein with top rank No. 10 (**Revised Manuscript Extended Data Fig. 1b**) and even ranked higher than CBX3 (No. 30) and histone H1 (No. 35), two other known direct binding proteins of SET (**Revised Manuscript Extended Data Tab. 1**). Taken together, based on our TAP-MS analysis, we found that ZBTB11 is a major binding protein of SET, and we speculated that the interaction between SET and ZBTB11 could be implicated in cellular functional regulation. Therefore, we chose the SET-ZBTB11 protein complex for subsequent studies.

2. The authors should provide validation data for the specificity of the ZBTB11 antibody using either KD or KO.

Response: These points were well taken. We thank the reviewer for this constructive comment. We used a ZBTB11 antibody purchased from Bethyl (A303-240A). This antibody was applicable to western blot, Co-IP and ChIP-seq/ChIP-qPCR assays. More importantly, this antibody was quite specific to recognize endogenous ZBTB11,

since knockdown of ZBTB11 by either siRNA or shRNA (they target distinct regions of *ZBTB11* mRNA) markedly reduced the blotting signals of ZBTB11 by this Bethyl brand antibody (**Revised Manuscript Fig. 2a and Fig. 4t**).

3. For the immunocytochemistry data presented in Figure 1 in terms of sub cellular localization of SET And ZBTB11 the authors should further validate the observation that both proteins are nuclear by performing nuclear cytoplasmic fractionation.

Response: This point was well taken. We thank the reviewer for this constructive suggestion. Following the reviewer's comment, we prepared the cytoplasmic and nuclear fractions of H1299 cells. The fractionation quality was confirmed by the cytoplasmic marker β -Actin and nuclear marker HDAC1. We found that ZBTB11 was predominantly localized in the nucleus, whereas SET was distributed in both the cytoplasm and nucleus (**Revised Manuscript Fig. 1f**). This result was consistent with our observations in the immunofluorescence (IF) assay (**Revised Manuscript Fig. 1g**) and further supported that SET and ZBTB11 mainly colocalized and interacted with each other in the nucleus.

4. Is there any structural or modeling data that would further support that the postulated sites of interaction between SET and ZBTB11 that are being proposed in this manuscript.

Response: We greatly appreciate the reviewer for this critical and constructive comment. Our data revealed that the acidic domain (AD) of SET directly interacted with the linker region (313-568 aa) of ZBTB11 (**Revised Manuscript Fig. 1i-j**). Due to the nature of enrichment with asparagine (D) and glutamic acid (E), the acidic domain of SET exhibits a highly negative charge, which, in turn, is able to mediate protein-protein interactions by recognizing the lysine/arginine (K/R)-rich regions that harbor a positive charge (**Wang et al. Nature 2016, PMID: 27626385**). Since we noticed that the 522-561 aa of ZBTB11 is a K/R-rich region (**Revised Manuscript Extended Data Fig. 1d**), we then divided the linker region into two fragments (313-521 aa and 522-568 aa) and evaluated the binding affinity of each fragment to SET. As shown in **Revised Manuscript Extended Data Fig. 1e**, aa 522-568 was necessary and sufficient for ZBTB11 binding with SET, indicating that the K/R-rich region is indeed critically involved in the ZBTB11-SET interaction. To further evaluate whether the positive charge is required for K/R-rich region binding with SET, we expressed mutant K/R-rich regions (Mut) where all lysine and arginine residues were replaced with alanine residues (A) to mimic a charge-neutralized status (**Revised Manuscript Extended Data Fig. 1d**). As shown in **Revised Manuscript Extended Data Fig. 1f**, the charge-neutralized mutant fragment completely lost the ability to bind with SET, confirming that the maintenance of the positive charge within the K/R-rich region is critical for the ZBTB11 interaction with SET. Taken together, our data reveal the "charge effect" as a physical basis for mediating the SET-ZBTB11 interaction, where the negative charge within the acidic domain of SET attracts the

positive charge within the K/R-rich region of ZBTB11 (**Revised Manuscript Extended Data Fig. 1g**).

5. For Figure 2a what time point was chosen for the RNA seq analysis as given the nature of the assay it would seem that many of the identified transcriptional changes are indirect and not direct targets of the ZBTB11 or SET complexes.

Response: We greatly appreciate the reviewer for this constructive comment. We agree with the reviewer's notion that an inappropriate time point in the knockdown experiment might lead to an indirect effect for subsequent functional analysis. Thus, it is most likely that the optimal time point for successfully knocking down a given target gene may vary among cell types (e.g., regular cancer cells vs. ESCs), species (e.g., human cells vs. mouse cells) and methods (e.g., siRNA vs. shRNA). For RNA-seq analysis in the current study, we transfected siRNAs twice (the second transfection was conducted at 48 hr after the first transfection) for a total of 96 hr into H1299 cancer cells to target human *ZBTB11* before we harvested cells for subsequent applications (**Revised Manuscript Fig. 2a**). The reason we chose 96 hrs as the time point in our experiment is to achieve high knockdown efficiency of ZBTB11. We performed pretesting to compare the knockdown efficiency of ZBTB11 at different time points. Knockdown for 48 hrs resulted in an approximate 43% reduction in *ZBTB11* mRNA levels, while knockdown for 96 hrs led to an approximate 64% reduction in *ZBTB11* mRNA levels (**Rebuttal Fig. 2a**). In addition, western blotting analysis verified that knockdown for 48 hrs exhibited a relatively modest effect on ZBTB11 protein levels, whereas knockdown for 96 hrs dramatically decreased ZBTB11 protein levels (**Rebuttal Fig. 2b**). Taken together, 96 hr was chosen for ZBTB11 depletion to obtain high knockdown efficiency, which, in turn, would increase the possibility of identifying differentially expressed genes (DEGs) by subsequent RNA-seq analysis.

Rebuttal Fig. 2 Evaluation of ZBTB11 knockdown efficiency at different time points. (a) RT-qPCR analysis of *ZBTB11* expression in H1299 cells transfected with control siRNA or siRNA

targeting *ZBTB11* for 48 hr (transfection once) or 96 hr (transfection twice; the second transfection was conducted 48 hr after the first transfection). (b) Western blot analysis of ZBTB11 protein levels in H1299 cells with or without ZBTB11 knockdown conducted under the same conditions in (a).

6. For the CHIP-SEQ data it is not clear how the target genes/promoters identified were narrowed down to the ones that are presented and there is overlap between the putative direct transcriptional targets of ZBTB11 and changes in their transcription from the RNA seq data that they performed in Figure 2A.

Response: This point was well taken. We thank the reviewer for this constructive comment. By describing how we analyzed the datasets in more detail, we believe that the logic of our study as well as the readability of our revised manuscript will be significantly improved. Specifically, after mapping ChIP-seq reads to the reference genome (hg38), MACS tools were used for independent peak calling of control (IgG)- or ZBTB11-binding DNA fragments under a p value cutoff <0.05 (with other parameters default). Peak calling identified 26788 peaks for IgG and 346961 peaks for ZBTB11. Next, peak annotation was performed by using the ChIPseeker packages of R with default parameters, considering the promoter region as 3 kb upstream and 3 kb downstream (-3 kb ~ +3 kb) of the TSS. The human genome (hg38) information referenced on peak annotation steps was obtained from Bioconductor packages (TxDb.Hsapiens.UCSC.hg38.knownGene). In total, we identified 20336 ZBTB11-specific binding loci based on the ChIP-seq dataset (**Revised Manuscript Fig. 2f**). Finally, we obtained 119 putative ZBTB11 direct targets by taking the intersection of ZBTB11-specific binding loci from ChIP-seq analysis with ZBTB11-regulated DEGs (203 genes) from RNA-seq analysis (**Revised Manuscript Fig. 2g**).

7. Further validation of at least a subset of the identified gene targets from their CHIP-SEQ experiments should be explored by either CHIP-qRT-PCR and promoter luciferase assays.

Response: This point was well taken. Following the reviewer's suggestion, we performed a ChIP-qPCR assay to verify potential ZBTB11 target genes. We chose 8 genes, including *ATPAF1*, *ANKRD40*, *COQ3*, *DUS1L*, *HEMK1*, *STX16*, *TACO1* and *TTC26*, for testing. As shown in **Revised Manuscript Extended Data Fig. 2a-h**, ChIP-qPCR analysis validated that ZBTB11 was indeed markedly enriched within the promoter of these genes, consistent with our ChIP-seq results.

8. For Figure 3 - the same comment as for Figure 2 - rationale for time point chosen and again many of the identified transcriptional changes by RNA seq are most likely indirect changes based upon effects on cell phenotypes rather than direct transcriptional targets.

Response: We appreciate the reviewer for this comment. Again, we agree with the reviewer that an inappropriate time point in the knockdown experiment might lead to an indirect effect for subsequent functional analysis. Similar to the *ZBTB11* case (**please also see the response to the reviewer's comment 5**), we transfected siRNAs twice (the second transfection was conducted at 48 hr after the first transfection) for a total of 96 hr into H1299 cancer cells to target human SET before we harvested cells for subsequent RNA-seq analysis (**Revised Manuscript Fig. 3a**). Again, we also performed pretesting to compare the knockdown efficiency of SET at different time points. Notably, unlike *ZBTB11* knockdown, which displayed relatively higher knockdown efficiency at 96 hr, depletion of *SET* in H1299 cells by siRNA for both 48 hr and 96 hr yielded robust and comparable knockdown efficiency (**Rebuttal Fig. 2b**). Since the RNA-seq performed in SET-KD cells aimed to facilitate the identification of SET-regulated *ZBTB11*-dependent target genes, we needed to keep the experimental conditions consistent with those conducted in *ZBTB11* knockdown assays. Therefore, we also adopted 96 hrs as the time point for SET knockdown.

Rebuttal Fig. 3 Evaluation of SET knockdown efficiency at different time points. (a) RT-qPCR analysis of *SET* expression in H1299 cells transfected with control siRNA or siRNA targeting *SET* for 48 hr (transfection once) or 96 hr (transfection twice; the second transfection was conducted 48 hr after the first transfection). (b) Western blot analysis of SET protein levels in H1299 cells with or without SET knockdown conducted under the same conditions in (a).

9. Does SET-ZBTB11 complex together directly on any of the target promoters that the authors suggest that they regulate and do they synergize to activate transcription and expression of target genes.

Response: We thank the reviewer for these constructive comments. As a downstream target, *MMP9* was transcriptionally activated by the SET-ZBTB11 complex. By using a ChIP-qPCR assay, we found that both ZBTB11 and SET were able to bind the same region of *MMP9* loci (**Revised Manuscript Fig. 5p**). In addition, a re-ChIP-qPCR assay proved SET-ZBTB11 complex occupancy on *MMP9* loci (**Revised Manuscript**

Fig. 5q). Moreover, depletion of ZBTB11 significantly reduced SET association on *MMP9* loci, indicating that the association of SET on *MMP9* loci is dependent on the presence of ZBTB11 (**Revised Manuscript Fig. 5r**). These data collectively support that the SET-ZBTB11 complex directly resides on *MMP9* loci.

Functionally, ZBTB11 overexpression increased the activity of the *MMP9* luciferase reporter (**Revised Manuscript Fig. 5t**). Although *MMP9* luciferase activity was comparable between the H1299-EV and H1299-FH-SET stable cell lines, SET overexpression markedly elevated ZBTB11-mediated *MMP9* luciferase activity (**Revised Manuscript Fig. 5t**). More importantly, this promotive effect of SET on ZBTB11-driven *MMP9* luciferase activity was largely dependent on the interaction between SET and ZBTB11, as the acidic domain-truncated SET (SET- Δ AD) failed to boost ZBTB11-mediated *MMP9* luciferase activity (**Revised Manuscript Extended Data Fig. 5l**). Taken together, our data indicate that SET and ZBTB11 synergize to activate the transcription and expression of their downstream target *MMP9*.

10. In Figure 4A the authors state there is no effect on cell proliferation what about on the cell cycle, response to chemotherapeutic drugs and colony formation?

Response: We thank the reviewer for these constructive suggestions. We monitored the cell cycle of H1299 cells depleted of SET or ZBTB11 by FACS and found that knockdown of either SET or ZBTB11 had no effect on cell cycle progression (**Revised Manuscript Extended Data Fig. 3c**). Next, we detected whether SET or ZBTB11 has a potential effect on cells in response to chemotherapeutic drugs. To this end, we treated H1299 cells depleted of SET or ZBTB11 with camptothecin (Cpt) for 24 hr, and cell viability was then measured. As shown in **Revised Manuscript Extended Data Fig. 3d**, Cpt treatment reduced the viability of the control cells (si-Ctr) by approximately 30%. Depletion of SET or ZBTB11 did not significantly affect cell death induced by Cpt (**Revised Manuscript Extended Data Fig. 3d**). Finally, we detected any potential effect of SET or ZBTB11 on colony formation. After 2 weeks of cell expansion and colonization, we found that SET or ZBTB11 depletion exhibited a tendency to slightly reduce colony formation of H1299 cells, but these minor changes showed no statistical significance (**Revised Manuscript Extended Data Fig. 3e**).

11. Did the authors consider doing the converse experiment instead of knocking down the target gene - did they overexpress SET or ZBTB11 or both and study the migration and invasion phenotypes.

Response: We greatly appreciate the reviewer for these constructive suggestions. By using a pair of stable cell lines (H1299-EV vs. H1299-FH-SET), we observed that SET overexpression modestly increased cell migration and invasion, and this effect was largely rescued by concomitant knockdown of ZBTB11 (**Revised Manuscript Fig. 4b-e**). Next, we generated a ZBTB11 stably overexpressing cell line (H1299 EV

vs. H1299 ZBTB11-SFB). Compared with SET overexpression, which modestly increased cell migration and invasion, ZBTB11 overexpression exhibited a strong effect on boosting cell migration and invasion (**Revised Manuscript Fig. 4g-j**). More importantly, knockdown of SET largely alleviated ZBTB11 overexpression-induced upregulation of both cell migration and invasion (**Revised Manuscript Fig. 4g-j**). Furthermore, re-expression of full-length SET (SET-FL), but not acidic domain-deleted SET (SET- Δ AD), which does not bind with ZBTB11, obviously rescued the reduced cell migration and invasion caused by depletion of endogenous SET in H1299 ZBTB11-SFB stable cells (**Revised Manuscript Fig. 4k-m**), further supporting that the formation of the SET/ZBTB11 complex critically contributes to SET-enhanced, ZBTB11-mediated regulation of cancer cell migration and invasion. Taken together, our data indicate that the interplay between SET and ZBTB11 is critically involved in regulating cancer cell metastatic behaviors.

12. Did the authors attempt to CRISPR the targets instead of just performing KD experiments?

Response: This constructive point was well taken. Indeed, for several functional analyses, CrispR-mediated knockout (KO) of target genes has a unique advantage over siRNA- or shRNA-mediated knockdown (KD), especially for some “rescue experiments”. In our current study, to functionally validate our hypothesis that the interaction between SET and ZBTB11 is critical for SET-enhanced, ZBTB11-dependent transcription of *MMP9*, we generated a series of stable cell lines including 1) H1299-SET-KO (SET-KO) based on the CrispR/Cas9 technique; 2) H1299-SET-KO stably expressing ectopic Flag-HA tagged full-length SET (SET-KO+FH-SET-FL); and 3) H1299-SET-KO stably expressing ectopic Flag-HA tagged acidic domain (AD)-truncated SET (SET-KO+FH-SET- Δ AD) (**Revised Manuscript Extended Data Fig. 5I**). As expected, in the SET-KO background, overexpression of ZBTB11 slightly activated *MMP9* transcription, as evidenced in a luciferase reporter assay (**Revised Manuscript Extended Data Fig. 5I**). More importantly, re-expression of ectopic SET-FL, but not SET- Δ AD, increased ZBTB11-dependent *MMP9* transcription (**Revised Manuscript Extended Data Fig. 5I**), indicating that the association of SET with ZBTB11 critically contributes to ZBTB11-mediated transcriptional regulation.

13. The rationale for exploring PRRG2 is not well justified or explained.

Response: This point was well taken. We thank the reviewer for this comment. This point will significantly strengthen our manuscript in logic. Since we observed that knockdown of ZBTB11 displayed a more robust effect on metastatic regulation than SET depletion, we speculated that there could be an extra way to modulate metastasis independent of SET-ZBTB11 complex formation. To this end, we focused on the target genes whose expression was regulated only by ZBTB11 but not by SET. Among these genes, 26 and 58 were upregulated and downregulated upon ZBTB11

knockdown, respectively (**Revised Manuscript Extended Data Fig. 6a**). We noticed that *PRRG2*, a previously less characterized gene, exhibited the most significant changes in its expression (considering both fold change and p value) upon ZBTB11 knockdown (**Revised Manuscript Extended Data Fig. 6a**). In addition, *PRRG4*, a member of the PRRG protein family, was recently reported to be involved in metastatic regulation (**Zhang et al. Oncogene 2020, PMID: 33037408**). Therefore, we were inspired to investigate whether ZBTB11-mediated *PRRG2* transcriptional changes also contributed to ZBTB11-dependent metastatic regulation. We have modified the revised manuscript to clearly describe the rationale for exploring *PRRG2*.

14. The transition to the study of the YAP/TAZ pathways is not well explained or justified.

Response: Again, this point was well taken. We thank the reviewer for this comment. Since we identified that *PRRG2* participated in ZBTB11-mediated regulation of cancer cell migration and invasion, we were then intrigued to investigate the potential mechanism by which *PRRG2* controls cancer cell metastatic behaviors. To date, functional studies on *PRRG2* are limited. Previous literature suggested that *PRRG2* may interact with YAP. This report prompted us to investigate whether *PRRG2* contributes to regulating cancer cell migration and invasion through a YAP-involved mechanism, as dysfunction of YAP/TAZ has emerged as a key event during cancer cell metastasis. Following this notion, we confirmed the cellular interaction between *PRRG2* and YAP1 (**Revised Manuscript Fig. 6m**), and more importantly, we found that upregulation of the *PRRG2*-mediated reduction in cell migration and invasion was almost completely abrogated upon YAP1 depletion (**Revised Manuscript Fig. 6n-r**). Mechanistically, we uncovered that ZBTB11 knockdown-mediated *PRRG2* upregulation markedly promoted the phosphorylation of YAP1 at serine 127 (p-YAP1-S127), a critical modification that inactivates YAP (**Revised Manuscript Fig. 6s-t**). Together, our study supports that YAP1 is critically involved in ZBTB11-*PRRG2* axis-mediated regulation of cancer cell metastatic behaviors. We have modified our text to clarify the transition to the study of the YAP/TAZ pathways in the revised manuscript.

15. In Figure 8 what did the expression of *SET* and *ZBTB11* look like in other lung cancer subtypes like lung adenocarcinoma or small cell lung cancers.

Response: We thank the reviewer for this constructive comment. In a previous version of the manuscript, we evaluated the expression of *SET* and *ZBTB11* in lung adenocarcinoma (LUAD) and found that both *SET* and *ZBTB11* were highly expressed in LUAD (**Revised Manuscript Fig. 8a**). To investigate whether altered expression of *SET* and/or *ZBTB11* also occurred in other types of lung cancer, we retrieved the TCGA dataset for lung squamous cell carcinoma (LUSC) samples. Notably, there was no information on small cell lung cancer available in the TCGA

dataset. As shown in **Revised Manuscript Extended Data Fig. 8f**, the expression of both *SET* and *ZBTB11* also exhibited high expression in LUSC. Taken together, the bioinformatic analyses of the TCGA dataset support the notion that both *SET* and *ZBTB11* are usually dysfunctional in lung cancer.

16. The clinical data and expression pattern of *SET* and *ZBTB11* are at times inconsistent with observed phenotypes proposed by authors - how do they explain these discrepancies.

Response: We thank the reviewer for pointing out this question. We feel that the reviewer may be concerned about the correlation between the expression pattern of *SET* and/or *ZBTB11* and the stages and/or lymph node metastasis. Overall, high expression of both *SET* and *ZBTB11* was observed in primary tumor samples of lung adenocarcinoma (LUAD) (**Revised Manuscript Fig. 8a**). In addition, upon detailed analysis of *SET/ZBTB11* expression in LUAD by classifying LUAD tumors into different stages or lymph node metastatic status, the overexpression of *SET* and/or *ZBTB11* was also observed in the majority of stages (stage 1, 3 and 4 for *ZBTB11*; stage 1-4 for *SET*) and lymph node metastasis status (N0 and N2 for *ZBTB11*; N0-3 for *SET*) (**Revised Manuscript Fig. 8b-c**). These data further support that dysfunction of *SET* and/or *ZBTB11* is associated with LUAD. Although some tendency was seen (e.g., gradually increased overexpression of *SET* associated with gradually advanced lymph node metastasis; **Revised Manuscript Fig. 8c**), these data did not attempt to generate a conclusion that gradually increased expression of *SET* and/or *ZBTB11* exactly correlated with gradually advanced stages and/or lymph node metastatic status. More importantly, these expression patterns of *SET* and/or *ZBTB11* analyzed based on the TCGA dataset reflect the difference between adjacent tissues and primary, but not distant metastatic, lung tumors. Therefore, we further monitored the expression of *SET* and/or *ZBTB11* among adjacent tissues, primary lung tumors and primary lung tumor-derived distant metastatic tumors by using LUAD tissue assays. Our data validated that both *SET* and *ZBTB11* are highly overexpressed in distant metastatic tumors (**Revised Manuscript Fig. 8e**), further supporting the notion that both *SET* and *ZBTB11* contribute to promoting distant metastasis of lung tumors.

17. For the survival data have the authors accounted for differences in stage, grade and treatment status when performing this analysis.

Response: We appreciate the reviewer for this constructive comment. Since disease-free survival information (it may indirectly reflect treatment status and prognosis) is available in this cohort (GEO dataset GSE30219), we then evaluated whether distinct expressional profiles of *ZBTB11*, *SET*, *MMP9* or *PRRG2* in LUAD tissues correlate with disease-free survival of LUAD patients. As shown in **Revised Manuscript Extended Data Fig. 8i**, we found that high expression of *SET* or low expression of *PRRG2* was well correlated with reduced disease-free survival in LUAD patients. In addition, despite not being statistically significant, the high

expression of both *ZBTB11* and *MMP9* also exhibited a tendency to be associated with reduced disease-free survival (**Revised Manuscript Extended Data Fig. 8i**). Taken together, in combination with the analyses of both overall survival and disease-free survival in LUAD patients, our data support the notion that dysfunction of *ZBTB11* and/or *SET* is associated with poor prognosis in LUAD patients.

18. In the patient cohorts is their co-occurrence data with over expression of *SET* and *ZBTB11* and some of the putative targets of these genes like *MMP9* and others.

Response: We thank the reviewer for this constructive comment. We analyzed the LUAD dataset (GSE30219) to evaluate the relationships between *SET* or *ZBTB11* and its target genes, including *MMP9* and *PRRG2*. As shown in **Revised Manuscript Extended Data Fig. 8g**, we observed a positive correlation between *MMP9* and *SET* expression. However, we did not find an obvious correlation between *MMP9* and *ZBTB11*. Since the expression of *MMP9* can be transcriptionally regulated by *SET* in a *ZBTB11*-dependent manner, these correlations might reflect that the overexpression of *SET* is sufficient to activate *ZBTB11*-mediated *MMP9* transcriptional upregulation by acting as a cofactor. For *PRRG2*, we observed that the expression of both *ZBTB11* and *SET* exhibited a significant negative correlation with *PRRG2* expression (**Revised Manuscript Extended Data Fig. 8h**).

Reviewer #3: HIPPO/YAP

In this manuscript, the authors studied the function of SET and ZBTB11 in regulating lung cancer metastasis. Through a tandem affinity purification and mass spectrometry, they identified ZBTB11 as a novel SET binding protein in lung cancer cells. They further showed that SET cooperated with ZBTB11 in transcriptional regulation of a set of genes involved in extracellular matrix organization. Through regulating these genes, such as MMP9, SET and ZBTB11 promote cancer cell migration and invasion. Interestingly, the study also found that ZBTB11 may also regulate the YAP signaling pathway through a SET-independent manner in regulating metastasis. Overall, the study revealed a previously unknown function of ZBTB11 as a metastasis promoter in lung cancer and could contribute to our understanding of the underlying mechanisms of lung cancer malignant progression. The manuscript was well written with clear presentation of the results; however, it should be further improved by addressing the following concerns.

Response: We are pleased to know that the reviewer found our study interesting. We greatly appreciate the constructive suggestions of our study by the reviewer. These insightful comments prompted us to conduct many additional experiments that have significantly strengthened our manuscript.

Major points:

1. Although two mouse models have been employed to demonstrate the essential role of ZBTB11 in lung cancer metastasis, these models have not been utilized to confirm if the two main mechanistic effectors of ZBTB11, including MMP9 and PRRG2-YAP, are involved in metastasis. Therefore, if these effectors found *in vitro* also exist *in vivo* is still unclear.

Response: We greatly appreciate the reviewer for this critical and constructive comment. First, we evaluated the potential effects of MMP9 on ZBTB11-mediated regulation of metastasis *in vivo* by analyzing the expression of MMP9 in GEMM-based lung tumors by IHC. As shown in **Revised Manuscript Extended Data Fig. 8b**, the expression levels of *Mmp9* in lung tumors derived from KLE-Zbtb11^{F1/F1} mice were lower than those in lung tumors derived from control KLE mice. These data support our cell line-based conclusion that ZBTB11 transcriptionally activated *MMP9* in lung cancer cells.

Next, we investigated whether the PRRG2-YAP axis is altered upon loss of ZBTB11 *in vivo*. As shown in **Revised Manuscript Extended Data Fig. 8c**, the expression of *Prrg2* in lung tumors derived from KLE-Zbtb11^{F1/F1} mice was higher than that in lung tumors derived from control KLE mice, consistent with our *in vitro* findings that ZBTB11 negatively regulates the expression of PRRG2.

Phosphorylation of YAP1 at serine 127 (p-YAP1-S127) inactivates YAP1 and

sequesters the YAP1 protein in the cytoplasm. Thus, the levels of p-YAP1-S127 may act as a general marker for YAP activity. Our data showed that PRRG2 overexpression in lung cancer cells dramatically increased p-YAP1-S127 without obviously changing YAP1 protein levels (**Revised Manuscript Fig. 6s**). More importantly, knockdown of ZBTB11 also elevated p-YAP1-S127, and this change in YAP1 phosphorylation was abrogated upon concomitant depletion of PRRG2 (**Revised Manuscript Fig. 6t**). This evidence supports that ZBTB11-mediated PRRG2 transcriptional regulation is linked to the alteration of YAP activity. To further evaluate whether elevated PRRG2 upon loss of ZBTB11 contributes to YAP inactivation *in vivo*, we measured p-YAP1-S127 in lung tumors derived from KLE or KLE-Zbtb11^{F1/F1} mouse models. As shown in **Revised Manuscript Extended Data Fig. 8d-e**, although the total levels of YAP1 were comparable between the two groups, p-YAP1-S127 was significantly higher in lung tumors derived from KLE-Zbtb11^{F1/F1} mice than in lung tumors derived from KLE mice, indicating that loss of ZBTB11 affects the PRRG2-YAP axis *in vivo*. Taken together, our data support the notion that alterations in both MMP9 and PRRG2-YAP are closely associated with ZBTB11-mediated regulation of tumor progression.

2. The study indicated that both SET and ZBTB11 are metastasis promoters. However, all major functional tests have focused on loss of function studies through depletion of these genes. While these tests showed the requirement of them, they cannot be the sole evidence to support their promotive roles.

Response: We greatly appreciate the reviewer for this constructive comment. Following the reviewer's suggestion, we evaluated cell migration and invasion in control or Flag-HA-tagged SET-overexpressing stable cells (H1299-EV vs. H1299-FH-SET) and found that SET overexpression modestly increased cell migration and invasion (**Revised Manuscript Fig. 4b-e**). In addition, we generated an H1299 control or SFB-tagged ZBTB11 stable cell line (H1299 EV vs. H1299 ZBTB11-SFB). As shown in **Revised Manuscript Fig. 4g-j**, ZBTB11 overexpression dramatically boosted cell migration and invasion. Therefore, our data support that overexpression of both SET and ZBTB11 promotes cancer cell migration and invasion.

3. The relationship between SET and ZBTB11 is confusing. The authors used the results shown in Figure 4b-e to support the notion that "SET hierarchically resides upstream of ZBTB11". However, this evidence is vague. To support this point, the study needs to examine if SET and/or ZBTB11 overexpression can promote cell migration, invasion, and metastasis (as indicated in the above concern), followed by examining if the effects by SET overexpression can be diminished by ZBTB11 depletion, and vice versa.

Response: We greatly appreciate the reviewer for this constructive and critical suggestion. First, we evaluated cell metastatic behaviors, including migration and

invasion, in control or Flag-HA-tagged SET-overexpressing stable cells (H1299-EV vs. H1299-FH-SET) depleted with or without ZBTB11 (**Revised Manuscript Fig. 4a**). As expected, knockdown of ZBTB11 markedly reduced cell migration and invasion (**Revised Manuscript Fig. 4b-e**). SET overexpression modestly increased cell migration and invasion, and more importantly, this effect was largely rescued by concomitant knockdown of ZBTB11 (**Revised Manuscript Fig. 4b-e**).

Next, we generated an H1299 control or SFB-tagged ZBTB11 stable cell line (H1299 EV vs. H1299 ZBTB11-SFB). Endogenous SET was depleted by siRNA in H1299 EV or ZBTB11-SFB cells to evaluate the potential effects of the interplay between SET and ZBTB11 on cell metastatic behaviors (**Revised Manuscript Fig. 4f**). Indeed, SET depletion decreased cell migration and invasion (**Revised Manuscript Fig. 4g-j**). Compared with SET overexpression-mediated modest increases in cell migration and invasion, ZBTB11 overexpression dramatically boosted cell migration and invasion (**Revised Manuscript Fig. 4g-j**). Notably, knockdown of SET significantly reduced ZBTB11 overexpression-induced upregulation of both cell migration and invasion (**Revised Manuscript Fig. 4g-j**).

To further investigate the relationship between SET and ZBTB11 in terms of their interaction in the metastatic regulation of cancer cells, we evaluated whether re-expression of RNAi-resistant full-length SET (re-FH-SET-FL) or acidic domain-deleted SET (re-FH-SET- Δ AD) is able to rescue the reduced cell migration and invasion caused by depletion of endogenous SET in stable H1299 ZBTB11-SFB cells (**Revised Manuscript Fig. 4k**). As shown in **Revised Manuscript Fig. 4l-m**, re-expression of SET-FL, but not SET- Δ AD, which does not bind with ZBTB11, obviously elevated cell migration and invasion, suggesting that the interaction between SET and ZBTB11 critically contributes to SET-enhanced, ZBTB11-mediated cancer cell metastatic regulation. Taken together, our data indicate that the interplay between SET and ZBTB11 is critically involved in regulating cancer cell metastatic behaviors.

4. The rationale for studying PRRG2 as a ZBTB11 target metastasis gene is unclear. How was PRRG2 noticed among the 119 putative ZBTB11 direct target genes (Figure 2g)? PRRG4 is a metastasis-promoting gene, why PRRG2 has a metastasis inhibitory role? Another major concern regarding PRRG2 is its link to YAP. Although the study confirmed the interaction between PRRG2 and YAP as reported previously, there were no further mechanistic study to examine the effect of their interaction on YAP function in regulating metastasis. Therefore, it is quite premature to conclude that the PRRG2-YAP signaling is an effector of ZBTB11.

Response: These points raised by the reviewer are very insightful and critical. We noticed that the reviewer primarily expressed his/her concerns in three aspects of PRRG2/YAP1-related studies, including 1) the rationale for studying PRRG2; 2) the discussion on the opposite roles in regulating metastasis between PRRG2 and PRRG4;

and 3) the mechanistic insight into PRRG2-mediated regulation of YAP1. Here, we would like to address these concerns separately.

1) The rationale for studying PRRG2. Since we observed that knockdown of ZBTB11 displayed a more robust effect on metastatic regulation than that upon SET depletion, we then speculated that there will be an extra way for ZBTB11 to modulate metastasis independent of SET-ZBTB11 complex formation. To this end, we focused on the target genes whose expression was regulated only by ZBTB11 but not by SET (84 out of 119 ZBTB11-regulated genes). Among these genes, 26 and 58 were upregulated and downregulated upon ZBTB11 knockdown, respectively (**Revised Manuscript Extended Data Fig. 6a**). We noticed that, by considering both fold change and p value, a previously less characterized gene *PRRG2* exhibited the most significant changes in its expression upon ZBTB11 knockdown (**Revised Manuscript Extended Data Fig. 6a**). In addition, PRRG4, a member of the PRRG protein family, was recently reported to be involved in metastatic regulation (**Zhang et al. Oncogene 2020, PMID: 33037408**). Therefore, we were inspired to investigate whether ZBTB11-PRRG2 axis contributed to ZBTB11-mediated metastatic regulation. Of note, although we proved our hypothesis that PRRG2 participates in ZBTB11-mediated regulation of cell migration and invasion in a SET/ZBTB11 complex-independent manner (**Revised Manuscript Fig. 6f-j and Extended Data Fig. 6c-l**), we cannot exclude the possibility that other SET-independent ZBTB11 targets are also involved in this regulation. Finally, we have modified our revised manuscript to clearly describe the rationale for exploring PRRG2.

2) The discussion on the opposite roles in regulating metastasis between PRRG2 and PRRG4. The proline-rich γ -carboxyglutamic acid (PRRG) family proteins, including PRRG1, PRRG2, PRRG3 and PRRG4, are cell surface-localized transmembrane proteins that are characterized by an extracellular γ -carboxyglutamic acid (Gla) domain and intracellular tandem Pro/Leu-Pro-Xaa-Tyr (PY) motifs. The PY motif is critical for PRRG proteins to selectively interact with other protein factors, which, in turn, may determine PRRG protein functions in a context-dependent manner.

For example, a previous study using a yeast two-hybrid screen showed that the PY motif of PRRG2 physically interacts with YAP (**Kulman et al. PNAS 2007, PMID: 17502622**). Since YAP has emerged as an increasingly recognized key metastatic regulator in multiple types of tumors, this evidence prompted us to investigate whether PRRG2-mediated metastatic regulation occurs through YAP-involved mechanisms. As shown in **Revised Manuscript Fig. 6o-r**, the reduced migration and invasion of lung cancer cells induced by PRRG2 overexpression was significantly abolished by concomitant YAP knockdown, suggesting that PRRG2-regulated cancer cell metastatic actions require the presence of YAP. Mechanistically, we found that PRRG2 upregulation by ZBTB11 knockdown markedly elevated YAP1 phosphorylation on Ser127 (p-YAP1-S127), a critical modification that inactivates

YAP1 (**Revised Manuscript Fig. 6s-t**). Thus, our data support the notion that PRRG2 suppresses cancer cell metastatic actions through the inactivation of YAP1 by promoting YAP1 phosphorylation.

In contrast to PRRG2, the binding of PRRG4 with YAP1 is controversial. A recent study conducted in breast cancer cells showed that PRRG4 did not bind YAP1. Instead, PRRG4 facilitates E3 ligase NEDD4-mediated Robo1 degradation by recruiting NEDD4 with its PY motif, which consequently promotes the metastatic actions of breast cancer cells (**Zhang et al. Oncogene 2020, PMID: 33037408**). Thus, it is most likely that the opposite roles of PRRG2 and PRRG4 in metastatic regulation rely on their selective binding with different functional protein partners.

3) The mechanistic insight into PRRG2-mediated regulation of YAP1. Increasing evidence indicates that aberrant YAP activation is sufficient to induce tumor progression, including LUAD (**Piccolo et al. Nature Cancer, 2023; PMID: 36564601**). The phosphorylation of YAP mediated by upstream kinases plays critical roles in modulating YAP activity. For example, the Hippo pathway component LATS1/2-mediated phosphorylation of YAP at serine 127 residue (p-YAP1-S127) may functionally inactivate YAP and sequester YAP within the cytoplasm (**Zhao et al. Genes Dev., 2007; PMID: 17974916**). Thus, p-YAP1-S127 may serve as a reliable marker to dictate YAP activity. Since YAP1 interacted with PRRG2 and was largely required for PRRG2-mediated repression of cancer cell migration and invasion (**Revised Manuscript Fig. 6n-r**), we speculated that PRRG2 likely exerts its metastatic suppressive actions by inhibiting YAP1 activity. To this end, we monitored YAP1 phosphorylation in control or PRRG2 stably expressing lung cancer cells. As shown in **Revised Manuscript Fig. 6s**, PRRG2 overexpression markedly elevated p-YAP1-S127 without changing total YAP1 levels. Interestingly, p-YAP1-S397, a key phosphorylation that primes YAP1 for subsequent phosphorylation and degradation (**Zhao et al. Genes Dev., 2010; PMID: 20048001**), displayed no alteration upon PRRG2 overexpression (**Revised Manuscript Fig. 6s**), consistent with the observation that PRRG2 had no effect on total YAP1 protein levels. More importantly, ZBTB11 depletion-mediated elevation of p-YAP1-S127 was remarkably abrogated upon concomitant PRRG2 knockdown (**Revised Manuscript Fig. 6t**), further supporting that the elevated PRRG2 upon ZBTB11 depletion suppresses cancer cell metastatic actions through the inactivation of YAP1 by promoting YAP1 phosphorylation at S127. We understand and agree with the reviewer that the detailed mechanisms by which the ZBTB11-PRRG2 regulatory axis links the YAP signaling pathway for tumor metastatic modulation could be complex and still need further investigation. Accordingly, in this revised manuscript, we clearly describe and discuss our current mechanistic findings and appropriately modify the text in terms of “PRRG2-YAP signaling as an effector of ZBTB11”.

Minor points:

1. In Figure 1i, the author stated, “we also mapped that the regions of ZBTB11

containing amino acids 313-568 and amino acids 569-937 were critical for interacting with SET (Fig. 1i)". However, the result showed that aa 313-568 is sufficient, but aa 569-937 is not required for the interaction. Please clarify this part.

Response: This point was well taken. We thank the reviewer for this comment. We have modified the description in the text of this revised manuscript. Now we describe this as "we also mapped the linker region (amino acids 313-568) of ZBTB11 as the primary region that was critical for interacting with SET".

2. In Figure 2d, it was unknown how the TSS were defined in this study. The method for defining TSS should be provided.

Response: This point was well taken. We thank the reviewer for this comment. Since the TSS refers to the first nucleotide from which transcription is initiated, we then defined the TSS region in this study as the region ranging from 3 kb upstream to 3 kb downstream of the TSS (-3 kb - +3 kb). We obtained TSS information from Bioconductor (TxDb.Hsapiens.UCSC.hg38.knownGene). Following the reviewer's comment, we have included this information in the Methods section of this revised version of the manuscript.

3. The effect shown in Extended Data Fig. 3d-g should be confirmed in another cell line. In Figure 3g, these results should be validated using additional cell lines.

Response: These points were well taken. We thank the reviewer for these constructive suggestions. We evaluated whether PRRG2 contributes to SET-mediated regulation of cell migration and invasion in H1975, another LUAD-derived cancer cell line. Again, knockdown of SET had no obvious effect on *PRRG2* expression in H1975 cells (**Revised Manuscript Extended Data Fig. 6h**). In addition, SET knockdown-induced suppression of cell migration or invasion cannot be rescued by concomitant depletion of *PRRG2* in H1975 cells (**Revised Manuscript Extended Data Fig. 6i-l**), similar to our previous observations in H1299 cells.

Next, we measured whether SET and/or ZBTB11 regulates the transcription of extracellular matrix organization-related genes, including *MMP9*, *MMP19*, *SPARC*, *PECAMI*, *SPP1* and *COL14A1*, in H1975 cells. As shown in **Revised Manuscript Extended Data Fig. 2i**, knockdown of SET and/or ZBTB11 repressed the expression of *MMP9*, *MMP19*, *SPARC* and *PECAMI* while activating the expression of *SPP1*. Of note, the expression of *COL14A1* also displayed a tendency of upregulation upon SET and/or ZBTB11 depletion, although the statistics were not significant (**Revised Manuscript Extended Data Fig. 2i**). This result reflects that SET/ZBTB11-mediated transcriptional regulation of *COL14A1* is relatively weak. Together, these results were largely consistent with our observations in H1299 cells, validating that these extracellular matrix organization-related genes were indeed coregulated by both SET and ZBTB11 in lung cancer cells.

4. In Figure 5g-j, and 5i-o, when overexpressing MMP9, why ZBTB11 KD or SET KD can still inhibit migration and invasion? Does this indicate that other factors are also involved in the effects?

Response: We appreciate the reviewer for this insightful comment. The reviewer was right. Since we identified that a series of genes (such as *MMP9*, *PECAM1* and *PRRG2*) regulated by SET and/or ZBTB11 were potentially involved in regulating metastatic behaviors, it was most likely that multiple downstream targets of SET and/or ZBTB11 contributed together to promote the cell migration and invasion observed in cultured cells. Consistent with this notion, overexpression of MMP9 markedly, but not completely, rescued the SET or ZBTB11 depletion-mediated reduction in cell migration and invasion (**Revised Manuscript Fig. 5g-j and Fig. 5l-o**). Therefore, our data indicate that MMP9 is one of key downstream effectors involved in SET/ZBTB11-mediated cell migration and invasion and support that multiple effectors downstream of SET and/or ZBTB11 could work together in regulating cellular metastatic behaviors.

5. Based on the results and model shown in Figure 5s-t, SET can promote ZBTB11 function. Can the authors provide certain mechanism for this effect?

Response: We greatly appreciate the reviewer for this constructive suggestion. The binding affinity or intensity of TFs to their cognate DNA element plays a key role in TF-driven transcription of specific target genes. Since SET served as a cofactor that facilitated ZBTB11-mediated transcription, we were intrigued to investigate whether SET affected ZBTB11 recruitment to target genes. To this end, we detected ZBTB11 occupancy on *MMP9* loci in H1299 cells with or without SET overexpression. As expected, ZBTB11 was readily observed to bind to *MMP9* loci in control cells (**Revised Manuscript Fig. 5s**). More importantly, SET overexpression markedly enhanced ZBTB11 occupancy on *MMP9* loci (**Revised Manuscript Fig. 5s**), supporting the notion that SET promotes ZBTB11 binding with its cognate DNA element. Consistent with this binding property, SET overexpression functionally increased ZBTB11-dependent transcription of *MMP9*, as evidenced by luciferase assay (**Revised Manuscript Fig. 5t**). Taken together, our data suggest a mechanism by which the cofactor SET facilitates ZBTB11-mediated transcriptional regulation by promoting ZBTB11 binding with its cognate elements of target genes.

6. In Figure 7c-f, how were the batch 1 and 2 defined? Why not put them together for analysis? What were the statistics test used in Figure 7d and 7f? It appeared that the numbers of individuals in each genotype group are too small for a statistic study. Can the authors examine liver, kidney, and spleen individually other than grouping them together?

Response: We greatly appreciate the reviewer for this constructive comment. In our

lung cancer GEMM, there were up to 4 genes whose expression could be manipulated directly by Cre-mediated recombination. Thus, it could be difficult to obtain many enough littermates with the desired genotypes for experiments during one round of breeding. To solve this issue, we performed two independent assays with KLE and KLE-Zbtb11^{F1/F1} littermates from distinct breeding to evaluate the role of Zbtb11 in lung tumor metastasis *in vivo*. Each independent experiment was defined as an individual batch (batch #1: 3 littermates each group; batch #2: 2 littermates each group).

For statistical analysis, two-way ANOVA was performed to test both the batch effect and Zbtb11 effect. This strategy of statistical analysis actually considered all samples together for analysis (n=5, not separately analyzed by n=3 or n=2), as the batch effect between two independent assays was taken into account. As shown in **Revised Manuscript Fig. 7f**, the difference in liver metastasis of primary lung tumors was attributed to loss of Zbtb11 expression ($p_{\text{group}}=0.0256$) but not attributed to the batch effect ($p_{\text{batch}}=0.2834$) between two independent assays.

Finally, we followed the reviewer's suggestion to examine liver, kidney and spleen metastasis separately. As shown in **Revised Manuscript Fig. 7f**, loss of Zbtb11 markedly alleviated liver metastasis of primary lung tumors. However, the reduction in metastasis in the kidney, spleen or intestine showed no statistical significance (**Revised Manuscript Fig. 7g-i**).

7. Why the organs with metastatic tumors in human (Extended Data Fig. 5b) have no overlap with those in the mouse model (Figure 7e)? Does this raise a concern regarding feasibility of the mouse model?

Response: We appreciate the reviewer for this insightful comment. In general, malignant primary lung tumors in humans preferentially metastasize to distal organs, including bone, brain, liver and adrenal gland (**Altorki et al. Nature Reviews Cancer, 2019; PMID: 30532012**). Since the etiologies as well as the underlying mechanisms of lung cancer are complex, the initiation and progression of lung tumors induced by distinct genetic aberrances may result in various phenotypic properties of the tumor, such as pathological subtypes, differentiation status, growth and/or metastatic preferences. Therefore, we agree with the reviewer that a given lung tumor mouse model might not completely mimic all aspects of human lung cancers. In our current genetically engineered mouse model (GEMM), we observed the liver as one of the major organs that is metastasized by primary lung tumors (**Revised Manuscript Fig. 7f**), partially consistent with the metastatic preference of human lung cancer in the clinic.

The tissue arrays we employed in this study were commercially purchased. Compared with primary tumor samples, metastatic tumor samples are relatively rare and may not be completely available for tissue array collection. Thus, we speculated that this could

be a reason that our current metastatic tissue array did not contain liver-metastasized tumor samples. Together, based on both the GEMM model and tissue array analyses, we can conclude that dysfunction of ZBTB11 is critically involved in modulating distal metastasis of primary lung tumors.

8. Molecular weight markers should be indicated with all western blotting results to indicate the relative size of each detected protein.

Response: This point was well taken. We have added molecular weight markers to all western blotting items.

Reviewers' comments:

Reviewer #1 (Remarks to the Author):

This revised manuscript has addressed some of the previous points but still suffers from several major problems. The rigor and novelty of the revised manuscript are not greatly improved compared to the original, despite the revisions made.

1. While the authors have further elucidated the role of the SET-ZBTB11 complex, this work still heavily relies on in vitro cell migration assays to explore the effects of ZBTB11 and SET. The revised manuscript is still biased towards MMP9, even though no in vivo experiments were performed to validate the effect of MMP9. The authors tried to assess clinical relevance based on analysis of public datasets, but the results do not show full alignment with their in vitro results. For instance, i) it is unclear whether the effect of the SET-ZBTB11 complex on metastasis is solely driven by MMP9, ii) SET overexpression poorly increased cell migration, even though the authors claimed that SET expression is upregulated in metastatic tumors and high expression of SET is associated with a poor prognosis for patients with LUAD, and iii) correlation analysis in Extended Data Fig. 8g,h showed a very different pattern from in vitro observation.

2. Although the goal of this work was to define SET and relevant TF in lung cancer metastasis, as indicated in the title, the manuscript lacks a clear definition of the effect of the main downstream target of the SET-ZBTB11 complex (MMP9) on lung cancer metastasis.

3. The proposed molecular mechanism still needs much characterization to be convincing, such as i) defining whether ZBTB11 and SET affect each other's DNA binding and ii) identifying the upstream signaling pathway dictating DNA binding of this complex, taking into account their results testing the involvement of DNA in the SET-ZBTB11 interaction.

4. The interpretation and claims of several points are not convincing.

a. SET overexpression poorly increased cell migration (Revised Fig. 4c), while Fig. 4g-j showed that migration and invasion increased by ZBTB11 were entirely dependent on SET expression. These results conflict with the authors' rationale for studying PRRG2.

b. The criteria they used for "ranking" in mass spectrometry analysis were based solely on the number of peptides, which largely depended on the size of the protein.

c. Based on the results of Fig. 4 and the relevant extended data figures, it is difficult to agree with authors' claim that SET may hierarchically reside upstream of ZBTB11.

d. SET knockdown reduced expression of ZBTB11. If SET is a transcriptional regulator of ZBTB11, the entire results must be interpreted differently.

e. The explanation against point #4 and #7 was very poor. vi) The manuscript lacks proper evidence of involvement of MMP9 in the axis of SET/ZBTB11-metastasis.

There are additional points that could be raised but, this being already a second round of review of the manuscript, I shall stop here.

Reviewer #2 (Remarks to the Author):

In this manuscript entitled "ZBTB11, a SET-associated Transcription Factor, Triggers Lung Cancer Metastasis" the authors identify and validate through a series of cell based and in vivo experiments the role of ZBTB11 and SET in regulating lung cancer development and progression. The authors have done a very thorough and comprehensive job in addressing all of the reviewers comments and questions.

Reviewer #3 (Remarks to the Author):

I would like to thank the authors for addressing my previous concerns, most of which have been well resolved. However, some points have not been sufficiently addressed.

One of my previous major concerns was "if the two main mechanistic effectors of ZBTB11, including MMP9 and PRRG2-YAP, are involved in metastasis in vivo". Although the authors have provided new results showing that these effectors are regulated in KLE-Zbtb11(FL/FL) mice as predicted, these results can not sufficiently support their in vivo roles. To test these, genetic epistasis analysis of their relationship with Zbtb11 should be examined using a mouse model. For example, can ZBTB11-KD inhibition of H1299 tumor metastasis be reversed by PRRG2-KD, or by YAP-S127A mutant expression? In addition, can ZBTB11 expression promote H1299 tumor metastasis? If yes, can this promotion be reversed by MMP9-KD, PRRG2 expression, or by YAP-KD? Furthermore, can PRRG2 expression inhibit H1299 tumor metastasis? If yes, can this inhibition be reversed by YAP-S127A mutant expression?

Another previous major concern was the effect of PRRG2 on YAP function in regulating metastasis. The revised manuscript showed that PRRG2 expression or ZBTB11 KD can increase the level of p-YAP1-S127, but not p-YAP1-S397. It would be important to examine if the YAP function in regulating expression of metastasis related genes is also inhibited in these situations. In addition, could the authors test if the YAP S127A mutation is able to reverse the effect of PRRG2 expression or ZBTB11 KD on cell migration?

Point-by-Point Response

Ref: ZBTB11, a SET-associated Transcription Factor, Triggers Lung Cancer Metastasis (Manuscript #: NCOMMS-22-29298B-Z)

Reviewer #1:

The reviewer commented: “This revised manuscript has addressed some of the previous points but still suffers from several major problems. The rigor and novelty of the revised manuscript are not greatly improved compared to the original, despite the revisions made.”

Response: We sincerely express our appreciation to the reviewer for his/her careful criticisms and constructive comments that will be critical to help us further improve our manuscript. We have taken the reviewer’s comments very seriously in both rounds of revisions. Following the reviewer’s comments, we performed a large number of *in vivo* experiments and provided more detailed explanations and discussions to support our study.

Specific Points:

The reviewer commented: “While the authors have further elucidated the role of the SET-ZBTB11 complex, this work still heavily relies on *in vitro* cell migration assays to explore the effects of ZBTB11 and SET. The revised manuscript is still biased towards MMP9, even though no *in vivo* experiments were performed to validate the effect of MMP9. The authors tried to assess clinical relevance based on analysis of public datasets, but the results do not show full alignment with their *in vitro* results. For instance, i) it is unclear whether the effect of the SET-ZBTB11 complex on metastasis is solely driven by MMP9, ii) SET overexpression poorly increased cell migration, even though the authors claimed that SET expression is upregulated in metastatic tumors and high expression of SET is associated with a poor prognosis for patients with LUAD, and iii) correlation analysis in Extended Data Fig. 8g,h showed a very different pattern from *in vitro* observation.”

Response: We respect the reviewer’s comments and agree with the reviewer’s concerns about the effect of MMP9 on the regulation of metastasis *in vivo*. To solve this point, we used a xenograft tumor metastasis model to evaluate the role of MMP9 in metastasis (**Revised Manuscript Fig. 5u**). Since SET acts as a co-factor that facilitates ZBTB11-dependent transcriptional upregulation of *MMP9*, we decided to overexpress ZBTB11 to drive *MMP9* expression and potential tumor metastasis *in vivo*. As expected, overexpression of ZBTB11 readily upregulated the expression of MMP9, and this regulatory effect on MMP9 expression can be successfully blocked by shRNA targeting *MMP9* (**Revised Manuscript Fig. 5v**). Overexpression of

ZBTB11 markedly promoted distal lung metastasis of the primary tumors that were inoculated subcutaneously on the flanked back (**Revised Manuscript Fig. 5w-x**). In addition, knockdown of MMP9 potently suppressed ZBTB11-induced tumor metastasis (**Revised Manuscript Fig. 5w-x**). Thus, our data validate that MMP9 is a key downstream effector of ZBTB11/SET that promotes tumor metastasis *in vivo*.

For the other three points raised by the reviewer, we would like to respond separately below.

- 1. Whether only MMP9 is responsible for SET-ZBTB11 complex-mediated metastatic regulation.** It is most likely that multiple downstream targets contribute together to SET-ZBTB11 complex-mediated metastatic regulation. Here, we would like to further describe and explain this view. To evaluate potential downstream targets that contribute to SET-ZBTB11 complex-mediated metastatic regulation, we primarily focused on the SET-ZBTB11 complex-regulated genes that are involved in extracellular matrix organization (**Revised Manuscript Fig. 3f**), a critical process for regulating cell migration and invasion. Among this group of target genes, we found that knockdown of MMP9 or PECAM1 successfully suppressed cancer cell migration and invasion in both H1299 and H1975 lung cancer cells (**Revised Manuscript Fig. 5a-e and Extended Data Fig. 5d-k**), supporting the view that multiple downstream targets, such as MMP9 and PECAM1, may contribute together to SET-ZBTB11 complex-mediated metastatic regulation. Interestingly, PECAM1 may act as a substrate of MMP9 that cleaves the extracellular portion of PECAM1 and facilitates leukocyte migration (**Kato et al. J Hepatol. 2014, PMID: 24412604**). Thus, SET-ZBTB11 complex-mediated transcriptional activation of MMP9 is likely involved in the functional regulation of PECAM1 in certain biological contexts. Therefore, we focused on MMP9 for further study *in vivo*. As shown in **Revised Manuscript Fig. 5u-x**, depletion of MMP9 markedly inhibited ZBTB11 overexpression-induced tumor metastasis in a xenograft mouse model, indicating that MMP9 indeed serves as a key effector to promote metastasis *in vivo*. Together, our data support that MMP9 acts as one of the key downstream targets that contribute to SET-ZBTB11 complex-mediated metastatic regulation.
- 2. The extent of cell migration regulated by SET.** We agree with the reviewer's comment. Compared with ZBTB11 overexpression, SET overexpression displayed a relatively modest effect on promoting cancer cell migration, but this promotive effect on cell migration was still sound and statistically significant (**Revised Manuscript Fig. 4b-c vs. Fig. 4g-h**). On the one hand, this discrepancy might reflect the regulatory characteristics of the SET-ZBTB11 complex on cell migration and invasion, since ZBTB11 acts as a transcription factor, while SET serves as a co-factor that facilitates ZBTB11-dependent transcription of metastasis-related genes (e.g., *MMP9*). Thus, overexpression of ZBTB11 (TF) might result in a more direct and relatively robust effect than overexpression of SET (co-factor). On the other hand, the extent of the effect of SET overexpression on cell migration and invasion may vary depending on the cellular context where

different levels of SET or ZBTB11 are already maintained. For example, as noted by the reviewer, overexpression of SET in parental H1299 cells promoted cell migration and invasion up to ~33% and ~46%, respectively (**Revised Manuscript Fig. 4b-e**). However, overexpression of SET in H1299-ZBTB11 stable cells with depletion of endogenous SET can yield ~62% and ~84% enhancement of cell migration and invasion, respectively (**Revised Manuscript Fig. 4l-m**). Thus, our data indicated that overexpression of SET indeed promoted cell migration and invasion, and the extent of this promotive effect could be cellular context dependent.

3. **Correlation between SET/ZBTB11 and MMP9/PRRG2 in lung cancer.** The reviewer is correct. Our previous analysis of the lung cancer dataset (GSE30219) showed an unexpected pattern of correlation between *ZBTB11* and *MMP9*, and between *SET* and *PRRG2*. Here, we mined another lung cancer dataset (GSE37745) to analyze their correlation. As shown in **Revised Manuscript Extended Data Fig. 9g**, we observed a positive correlation between *MMP9* and *SET* expression. In addition, the expression of *ZBTB11* and *MMP9* displayed a tendency of positive correlation, although the statistical significance was not sound (**Revised Manuscript Extended Data Fig. 9g**). Since the expression of *MMP9* can be transcriptionally regulated by SET in a ZBTB11-dependent manner, these correlations might reflect that the overexpression of SET is sufficient to activate ZBTB11-mediated *MMP9* transcriptional upregulation by acting as a cofactor. For *PRRG2*, we observed that the expression of ZBTB11, but not SET, exhibited a significant negative correlation with *PRRG2* expression (**Revised Manuscript Extended Data Fig. 9h**), consistent with our data that *PRRG2* was transcriptionally repressed by ZBTB11 but not SET.

The reviewer commented: “Although the goal of this work was to define SET and relevant TF in lung cancer metastasis, as indicated in the title, the manuscript lacks a clear definition of the effect of the main downstream target of the SET-ZBTB11 complex (MMP9) on lung cancer metastasis.”

Response: We thank the reviewer for raising this point, and we would like to further describe the effect of MMP9 on SET-ZBTB11 complex-mediated metastatic regulation. Cancer cell migration and invasion represent a critical step during tumor metastasis. MMP9 is an enzyme that is involved in degrading the extracellular matrix to facilitate cancer cell migration and invasion. In our study, we identified that MMP9 acted as a downstream target whose expression was transcriptionally upregulated by the SET-ZBTB11 complex (**Revised Manuscript Fig. 5p-t and Extended Data Fig. 5l**), thereby promoting cancer cell migration and invasion (**Revised Manuscript Fig. 5f-o**). In addition, consistent with the role and regulation of MMP9 observed *in vitro*, loss of *Zbtb11* resulted in reduced levels of *Mmp9* and increased levels of collagen IV in lung tumors derived from an *in vivo* GEMM (**Revised Manuscript Extended Data Fig. 9a-b**). Furthermore, by using a xenograft model to monitor tumor metastasis regulated by MMP9, we found that ZBTB11 overexpression-induced tumor

metastasis can be markedly suppressed by the depletion of MMP9 (**Revised Manuscript Fig. 5u-x**), indicating that MMP9 is a key downstream effector that modulates tumor metastasis *in vivo*. Taken together, our data support that SET-ZBTB11 complex-mediated transcriptional activation of MMP9 contributes to cancer cell migration and invasion by degrading the extracellular matrix, which promotes distal metastasis of primary tumors.

The reviewer commented: “The proposed molecular mechanism still needs much characterization to be convincing, such as i) defining whether ZBTB11 and SET affect each other's DNA binding and ii) identifying the upstream signaling pathway dictating DNA binding of this complex, taking into account their results testing the involvement of DNA in the SET-ZBTB11 interaction.”

Response: These points were greatly appreciated and well taken. Here, based on the reviewer's questions, we would like to further explain the mechanistic characteristics of the interaction between ZBTB11 and SET as well as SET-ZBTB11 complex-mediated transcriptional regulation. We will describe these two points separately.

- 1. Whether ZBTB11 and SET affect each other's DNA binding.** First, literatures have shown that SET is not able to directly bind DNA. The association of SET with chromatin/DNA is mediated by other chromatin/DNA-binding proteins, such as histones (e.g., H3) and transcription factors (e.g., p53). Consistently, in the current study, we identified that SET interacts with the transcription factor ZBTB11 and facilitates ZBTB11-mediated transcriptional regulation of target genes (e.g., *MMP9*) by serving as a co-factor of ZBTB11. SET-ZBTB11 indeed formed a protein complex that bound to *MMP9* loci, and loss of ZBTB11 profoundly abrogated the chromatin binding of SET (**Revised Manuscript Fig. 5p-r**), indicating that the recruitment of SET to *MMP9* loci is largely dependent on the presence of ZBTB11. Second, our data showed that overexpression of SET enhanced the recruitment of ZBTB11 to *MMP9* loci (**Revised Manuscript Fig. 5 s**), supporting the notion that the interaction between SET and ZBTB11 promotes ZBTB11 binding to its cognate DNA elements.
- 2. The signaling dictating DNA binding of the SET-ZBTB11 complex and the role of DNA in the SET-ZBTB11 interaction.** It is currently unknown which upstream signaling pathways influence DNA binding of the SET-ZBTB11 complex. We feel that this is an interesting but still opening and broad question, since the DNA binding of SET-ZBTB11 complex, as well as the formation of SET-ZBTB11 complex might be regulated in different layers in response to various signals. Since the expression of SET and ZBTB11 was elevated and displayed a positive correlation in lung tumors, especially in distal metastasized lung tumors (**Revised Manuscript Fig. 9a-f**), we speculate that tumor progression may represent one of the selective pressures to promote SET-ZBTB11 complex formation, which may subsequent influence DNA binding of the complex. In addition, since the “charge effect” is critical to mediate SET-ZBTB11 interaction

(**Extended Data Fig. 1g**), the signalings or stresses inducing posttranslational modifications (PTMs) that alter local charge status of ZBTB11 or SET (e.g., neutralization of the positive charge by acetylation of lysine residues on K/R-rich region of ZBTB11) is likely to have an impact on the interaction between SET and ZBTB11, and consequently, the DNA-binding of SET-ZBTB11 complex. We have included this point in the discussion section in this revised manuscript.

Next, in terms of the involvement of DNA in the SET-ZBTB11 interaction, our data have showed that the interaction between SET and ZBTB11 did not require the presence of DNA, since complete degradation of DNA by benzonase still maintains the interaction between SET and ZBTB11 (**Revised Manuscript Extended Data Fig. 1c**).

The reviewer commented: “The interpretation and claims of several points are not convincing. **a.** SET overexpression poorly increased cell migration (Revised Fig. 4c), while Fig. 4g-j showed that migration and invasion increased by ZBTB11 were entirely depend on SET expression. These results conflict with the authors' rationale for studying PRRG2. **b.** The criteria they used for “ranking” in mass spectrometry analysis were based solely on the number of peptides, which largely depended on the size of the protein. **c.** Based on the results of Fig. 4 and the relevant extended data figures, it is difficult to agree with authors claim that SET may hierarchically reside upstream of ZBTB11. **d.** SET knockdown reduced expression of ZBTB11. If SET is a transcriptional regulator of ZBTB11, the entire results must be interpreted differently. **e.** The explanation against point #4 and #7 was very poor. **vi)** The manuscript lacks proper evidence of involvement of MMP9 in the axis of SET/ZBTB11-metastasis. There are additional points that could be raised but, this being already a second round of review of the manuscript, I shall stop here.”

Response: The comments raised by the reviewer were critical and important for us to improve our manuscript. We have taken these points very carefully, and here, we would like to respond these points separately.

1. **The effect of SET in regulating cell migration and invasion.** The reviewer is correct. Under our experimental conditions, overexpression of SET resulted in a relatively modest effect on cell migration and invasion, but this promotive effect was still statistically significant (**Revised Manuscript Fig. 4b-e**). The extent of the effect of SET overexpression on cell migration and invasion may vary depending on the cellular context where different levels of SET or ZBTB11 are already maintained. For example, as noted by the reviewer, overexpression of SET in parental H1299 cells promoted cell migration and invasion up to ~33% and ~46%, respectively (**Revised Manuscript Fig. 4b-e**). However, overexpression of SET in H1299-ZBTB11 stable cells depleted of endogenous SET yielded ~62% and ~84% enhancement of cell migration and invasion, respectively (**Revised Manuscript Fig. 4l-m**). Thus, our data indicated that overexpression of SET indeed promoted cell migration and invasion, and the extent of this promotive effect could be cellular context dependent.

In terms of the regulatory role of SET in ZBTB11-mediated cell migration and invasion, knockdown of endogenous SET inhibited ZBTB11 overexpression-induced cell migration and invasion (**Revised Manuscript Fig. 4g-j**). Notably, although this inhibitory effect was remarkable, depletion of SET did not completely rescue ZBTB11-mediated cell migration and invasion (**Revised Manuscript Fig. 4h column 2 vs. 4; Fig. 4j column 2 vs. 4**), consistent with our notion that a SET-ZBTB11 complex-independent mechanism may exist in parallel to contribute to ZBTB11-mediated metastatic regulation. Indeed, our data validated that PRRG2, a downstream effector that is regulated by ZBTB11 but not SET, participates in the ZBTB11-mediated regulation of tumor metastasis *in vivo* (**Revised Manuscript Fig. 6m-p**). Taken together, our data reveal diversified mechanisms by which the transcription factor ZBTB11 contributes to the regulation of tumor metastasis.

- 2. The rationale for ZBTB11 identification from SET-containing protein complexes.** We agree with the reviewer that multiple parameters can be taken into account for candidate validation upon receiving TAP-MS results. Peptide number and unique peptide number are the parameters that are commonly used to evaluate the possibility and/or strength of the binding of candidates to the bait. In addition, our primary goal to conduct TAP-MS is to identify nuclear factors that are involved in SET-mediated transcriptional regulation. Thus, it is reasonable for us to pay attention to the top-ranked transcription factor ZBTB11 for further investigation (**Revised Manuscript Extended Data Fig. 1b**). Indeed, this strategy led us to successfully validate both the physical and functional interplays between the bait and the candidate binding partners. As shown in **Revised Manuscript Figs. 1-5**, we successfully validated that SET interacts with ZBTB11 and regulates the ZBTB11-mediated metastatic behaviors of cancer cells. Together, our data reveal that ZBTB11 acts as a novel functional binding partner of SET.
- 3. The hierarchical relationship between SET and ZBTB11.** This point was well taken. The reviewer was correct. The description by using “hierarchical relationship” was not precise. Here, we would like to further explain it. The recognition of the relationship between SET and ZBTB11 in metastatic regulation was based the mechanism of SET-ZBTB11 complex-mediated transcription of their metastasis-related target genes. We uncovered that SET facilitated ZBTB11-dependent transcriptional activation of MMP9, a downstream target gene critically involved in inducing cell migration and invasion during tumor metastasis, by acting as a cofactor of ZBTB11 (**Revised Manuscript Fig. 5p-t**). These findings support that SET-mediated regulation of MMP9 and MMP9-induced cell migration and invasion are largely dependent on the presence of ZBTB11. To avoid potential misunderstanding, we have removed the words like “hierarchical relationship” and instead, directly described the mechanistic relationship between SET and ZBTB11 in regulating cancer cell metastatic behaviors in this revised manuscript.
- 4. The effect of SET knockdown on ZBTB11 expression.** We appreciate the

reviewer for this constructive comment. We also noticed that knockdown of SET slightly decreased the levels of ZBTB11. This phenomenon is potentially interesting and might suggest an alternative way of SET in the regulation of ZBTB11 under certain contexts. However, this phenomenon is not against the mechanism by which SET and ZBTB11 form a protein complex for transcriptional regulation of their target genes. Our data showed that SET can interact with ZBTB11 and facilitate ZBTB11-dependent transcription of *MMP9* by serving as a co-factor of ZBTB11 (**Revised Manuscript Fig. 5p-t**). In addition, among 119 putative ZBTB11 target genes, 35 genes were coregulated by both SET and ZBTB11, whereas 84 genes were only regulated by ZBTB11 but not SET (**Revised Manuscript Fig. 3d and Extended Data Fig. 6a**). Thus, if SET-mediated regulation of ZBTB11 levels is a primary way to control ZBTB11-mediated transcription, it would be observed that the transcription of the majority of ZBTB11 target genes can be regulated by SET. Together, although knockdown of SET may slightly decrease ZBTB11, we speculated that the effect of this regulation on ZBTB11-mediated transcription, at least in our current experimental context, was limited.

5. **The explanations against previous points #4 and #7.** These points were well taken. Here, we would like to further discuss these points separately.

1) **In previous point #4, the reviewer commented:** “There is a big discrepancy between ZBTB11 ChIP-seq peaks and ZBTB11-regulated genes (Fig. 2g). Less than 1% of ZBTB11 binding loci show changes in gene transcription. Why does ZBTB11 bind to a large number of loci without changing transcription? Fig. 2 shows that ZBTB11 is either a transcriptional activator or a repressor. How does ZBTB11 repress the transcription of its target genes?”

Response to previous point #4: We greatly appreciate the reviewer for these critical and insightful comments. The reviewer is correct. We also noticed that in our study, only 0.59% (119 out of 20336) of ZBTB11-binding loci was linked to transcriptional regulation. Since, to our knowledge, our study by ChIP-seq assay was the first to attempt to identify ZBTB11 target genes in human cancer cells, we aimed to screen potential ZBTB11 target genes as much as possible for subsequent investigation. Thus, we employed a general but not extremely stringent threshold to perform peak calling through the MACS tool (p value cutoff <0.05 with other parameters default). Although this modest criterion can yield, in relatively, more binding loci, the results were still highly reliable. For example, along the ChIP-seq analysis, we easily tested a group of candidate genes by ChIP-qPCR and successfully validated ZBTB11 binding to their promoters in cancer cells (**Revised Manuscript Extended Data Fig. 2a-h**). More importantly, this strategy did not affect subsequent functional and mechanistic studies, as evidenced by the identification of *MMP9* and *PRRG2* as bona fide target genes of ZBTB11 by ChIP-qPCR and luciferase assays (**Revised Manuscript Fig. 5p-t and Fig. 6k-l**).

To further evaluate the profiles of ZBTB11-mediated transcriptional regulation, we applied exactly the same strategy and threshold to analyze the genome-wide

distribution of ZBTB11 and ZBTB11-mediated transcriptional alteration in mouse embryonic stem cells (mESCs) based on published ChIP-seq and RNA-seq datasets (**Wilson et al. Nature Communications, 2020; PMID: 33122634**). We still found a relatively low percentage of ZBTB11-binding loci associated with transcriptional changes (1.27%, 87 out of 6825). Moreover, the binding motif of ZBTB11 revealed by our ChIP-seq analysis was almost completely identical to that uncovered in the published study mentioned above (**Rebuttal Fig. 1**), further supporting that our ChIP-seq results indeed reflect the DNA-binding profiles of ZBTB11. Although the detailed profiles of the genome-wide distribution of ZBTB11 as well as ZBTB11-mediated transcriptional regulation may vary depending on cell type, tissue of origin or even species, the analyses based on both our study and published datasets prompt us to speculate that such unexpected binding profiles of ZBTB11 (fewer binding loci show changes in gene transcription) might reflect intrinsic characteristics of ZBTB11 in terms of its molecular functions by acting as a chromatin/DNA-binding protein. For example, a recent study showed that the chromatin association of transcription factors (TFs) may serve as chromatin architectural regulators that are responsible for high-order chromatin structure organization (e.g., MyoD, **Wang et al. Nature Communications, 2022, PMID: 35017543**). This evidence implies that chromatin/DNA binding of TFs could be functionally diversified and beyond traditionally recognized transcriptional regulation.

Rebuttal Fig. 1 DNA-binding motif of ZBTB11. Left: DNA-binding motif of ZBTB11 identified in the current study. Right: DNA-binding motif of ZBTB11 identified by Wilson et al. published in Nature Communications in 2020.

For the reviewer's second comment, we indeed found that ZBTB11 may serve as either a transcriptional activator (e.g., for transcriptionally activating *MMP9*) or repressor (e.g., for transcriptionally repressing *PRRG2*) in cancer cells. Generally, the regulatory consequences of specific target genes by a given TF are context-dependent in eukaryotic cells, which largely rely on TF-mediated differential recruitment of cofactors to target gene promoters or other regulatory elements. ZBTB family proteins share a conserved BTB domain that is primarily responsible for the recruitment of transcriptional cofactors. Accumulating evidence indicates that ZBTB family proteins may not only act as transcriptional

repressors by recruiting corepressors such as N-CoR (nuclear receptor corepressor) or SMRT (silencing mediator for retinoid and thyroid hormone receptor) (Huynh et al. *Oncogene*, 1998, PMID: 9824158), but may also serve as transcriptional activators by recruiting coactivators such as p300 (Staller et al. *Nature Cell Biology*, 2001, PMID: 11283613). For example, ZBTB17 (also known as MIZ1) functions as a transcriptional activator or repressor, depending on ZBTB17-mediated recruitment of its binding partners to specific promoters (Schneider et al. *Curr Top Microbiol Immunol.*, 1997, PMID: 9308237; Staller et al. *Nature Cell Biology*, 2001, PMID: 11283613). Thus, due to the structural similarity of the BTB domain among ZBTB family proteins, it is most likely that ZBTB11 functions as a transcriptional activator for a subgroup of target genes by binding and recruiting coactivators on their promoters or other regulatory elements. Taken together, although the mechanisms by which ZBTB11 selectively regulates specific target genes need further investigation, our current data reveal that ZBTB11 serves as a transcription factor in cancer cells.

2) **In previous point #7, the reviewer commented:** “The authors mix siRNA-mediated transient knockdown (Fig. 1-6), shRNA-mediated stable knockdown (Fig. 4h), and inducible knockout (Fig. 5r). There is no apparent or given reason for the use of these different methods.”

Response to previous point #7: We thank the reviewer for this comment. Here, we would like to explain why we chose different methods in specific assays. First, we employed siRNA in *in vitro* assays such as RNA-seq, cell migration and invasion, since these assays only took a few days and the transient transfection of siRNA exhibited a good knockdown efficiency that is sufficient to elicit the biological effects (**Revised Manuscript Fig. 2-7**).

Next, for the *in vivo* xenograft tumor growth and metastasis assay, the experiment took 5-6 weeks, and transient knockdown by siRNA was expected to not successfully maintain a desired knockdown effect during such a long period. Therefore, we used a shRNA strategy to generate the indicated stable knockdown cell lines for this *in vivo* assay (**Revised Manuscript Fig. 4s-v, Fig. 5u-x, Fig. 6m-p and Fig. 7i-o**).

Finally, to evaluate whether the recruitment of SET to *MMP9* loci is dependent on ZBTB11, we performed a ChIP assay in H1299 cells with or without ZBTB11 depletion. To increase the sensitivity of the downstream ChIP-qPCR assay, a relatively large number of cells are recommended for the initial experimental setup. To facilitate cell preparation and maintain a high ZBTB11 knockdown efficiency in a timely and economical manner, we used our ZBTB11 inducible knockout (ZBTB11-iKO) cells, in which depletion of ZBTB11 could be easily achieved by doxycycline (Doxy) administration (**Revised Manuscript Fig. 5r**). As a result, we observed a very decent decrease in SET enrichment at *MMP9* loci upon ZBTB11 depletion by using this iKO method.

6. **The evidence of MMP9 in metastasis regulation.** Again, this point was well

taken. We generated a xenograft tumor metastasis model to evaluate the role of MMP9 in metastasis (**Revised Manuscript Fig. 5u**). Since SET acts as a co-factor that facilitates ZBTB11-dependent transcriptional upregulation of *MMP9*, we decided to overexpress ZBTB11 to drive *MMP9* expression and potential tumor metastasis *in vivo*. As expected, overexpression of ZBTB11 readily upregulated the expression of MMP9, and this regulatory effect on MMP9 expression can be successfully blocked by shRNA targeting *MMP9* (**Revised Manuscript Fig. 5v**). Overexpression of ZBTB11 markedly promoted distal lung metastasis of the primary tumors that were inoculated subcutaneously on the flanked back (**Revised Manuscript Fig. 5w-x**). In addition, knockdown of MMP9 potently suppressed ZBTB11-induced tumor metastasis (**Revised Manuscript Fig. 5w-x**). Thus, our data validate that MMP9 is a key downstream effector of ZBTB11/SET that promotes tumor metastasis *in vivo*.

Reviewer #2:

The reviewer commented: “In this manuscript entitled "ZBTB11, a SET-associated Transcription Factor, Triggers Lung Cancer Metastasis" the authors identify and validate through a series of cell based and in vivo experiments the role of ZBTB11 and SET in regulating lung cancer development and progression. The authors have done a very thorough and comprehensive job in addressing all of the reviewers comments and questions.”

Response: We are very pleased to know that our new data in the revised version of the manuscript have fully addressed the reviewer’s questions. We are grateful to the reviewer for his/her time and effort in reviewing our paper and supporting our work.

Reviewer #3:

The reviewer commented: “I would like to thank the authors for addressing my previous concerns, most of which have been well resolved. However, some points have not been sufficiently addressed.”

Response: First, we are pleased to know that our new data have addressed the majority of the reviewer’s previous concerns. Second, we deeply appreciate these new comments raised by the reviewer, since we feel that these new suggestions are very constructive, concrete and feasible. Indeed, we have performed a large number of new *in vivo* experiments to strengthen our study and resolve the remaining concerns of the reviewer. We are truly grateful to the reviewer for his/her enthusiastic help in further improving our study.

Specific Points:

The reviewer commented: “One of my previous major concerns was “if the two main mechanistic effectors of ZBTB11, including MMP9 and PRRG2-YAP, are involved in metastasis *in vivo*”. Although the authors have provided new results showing that these effectors are regulated in KLL4-Zbtb11(FL/FL) mice as predicted, these results can not sufficiently support their *in vivo* roles. To test these, genetic epistasis analysis of their relationship with Zbtb11 should be examined using a mouse model. For example, can ZBTB11-KD inhibition of H1299 tumor metastasis be reversed by PRRG2-KD, or by YAP-S127A mutant expression? In addition, can ZBTB11 expression promote H1299 tumor metastasis? If yes, can this promotion be reversed by MMP9-KD, PRRG2 expression, or by YAP-KD? Furthermore, can PRRG2 expression inhibit H1299 tumor metastasis? If yes, can this inhibition be reversed by YAP-S127A mutant expression?”

Response: We greatly appreciate these constructive and critical suggestions by the reviewer. We have performed a series of *in vivo* experiments to address the reviewer’s concerns. Here, we would like to respond to these points separately.

1. **ZBTB11-MMP9 axis in regulating tumor metastasis *in vivo*.** To evaluate the role of the ZBTB11-MMP9 regulatory axis in tumor metastasis *in vivo*, we generated H1299 stable cell lines overexpressing exogenous ZBTB11 (FH-ZBTB11) and/or depleting endogenous MMP9 (MMP9-KD), and used a xenograft tumor model to observe tumor metastasis (**Revised Manuscript Fig. 5u-v**). As expected, overexpression of ZBTB11 promoted distal lung metastasis of the primary tumors (**Revised Manuscript Fig. 5w-x**). Moreover, knockdown of MMP9 profoundly reversed the promotive effect on tumor metastasis by ZBTB11 overexpression (**Revised Manuscript Fig. 5w-x**), indicating that the ZBTB11-MMP9 regulatory axis plays a critical role in modulating tumor metastasis *in vivo*.
2. **ZBTB11-PRRG2-YAP1 axis in regulating tumor metastasis *in vivo*.** First, we

evaluated whether PRRG2 contributes to ZBTB11-mediated metastatic regulation *in vivo* by conducting a xenograft tumor model (**Revised Manuscript Fig. 6m**). As expected, knockdown of ZBTB11 markedly elevated PRRG2 expression, which was abolished upon PRRG2 knockdown with shRNA (**Revised Manuscript Fig. 6n**). Again, depletion of endogenous ZBTB11 remarkably suppressed distal lung metastasis of the primary subcutaneous tumors (**Revised Manuscript Fig. 6o-p**). More importantly, this suppressive effect can be largely reversed upon simultaneous knockdown of PRRG2 (**Revised Manuscript Fig. 6o-p**). This evidence validates that PRRG2 serves as a key downstream target involved in ZBTB11-mediated metastatic regulation *in vivo*.

Since our previous *in vitro* assays revealed that PRRG2 upregulated YAP1 phosphorylation at Ser127 (**Revised Manuscript Fig. 7g-h**), we next investigated whether PRRG2-mediated metastasis suppression can be compromised by replenishing ectopic YAP1 activity *in vivo* (**Revised Manuscript Fig. 7i**). To this end, we generated H1299 cell lines that stably express PRRG2 and/or YAP1-S127A (**Revised Manuscript Fig. 7j**). We observed that PRRG2 overexpression indeed suppressed tumor cell metastasis *in vivo* (**Revised Manuscript Fig. 7k-l**). In addition, overexpression of the YAP-S127A mutant significantly abolished PRRG2-mediated inhibition of tumor metastasis (**Revised Manuscript Fig. 7k-l**). These data indicate a metastasis-suppressive role of PRRG2 by functionally regulating YAP1.

Finally, we evaluated the roles of PRRG2-mediated YAP1 regulation in ZBTB11-controlled metastasis *in vivo*. As expected, ZBTB11 overexpression promoted distal lung metastasis of the primary tumors (**Revised Manuscript Fig. 7m-o**). Notably, overexpression of PRRG2 or depletion of YAP1 markedly attenuated ZBTB11-promoted tumor metastasis (**Revised Manuscript Fig. 7m-o**), validating a critical role of the ZBTB11-PRRG2-YAP1 axis in regulating tumor metastasis.

Taken together, our new data validate the critical roles of downstream effectors, including MMP9 and PRRG2-YAP1, in the regulation of ZBTB11-controlled metastasis *in vivo*.

The reviewer commented: “Another previous major concern was the effect of PRRG2 on YAP function in regulating metastasis. The revised manuscript showed that PRRG2 expression or ZBTB11 KD can increase the level of p-YAP1-S127, but not p-YAP1-S397. It would be important to examine if the YAP function in regulating expression of metastasis related genes is also inhibited in these situations. In addition, could the authors test if the YAP S127A mutation is able to reverse the effect of PRRG2 expression or ZBTB11 KD on cell migration?”

Response: We appreciate these critical comments by the reviewer. We measured the expression of a series of metastasis-related genes previously reported to be regulated by YAP1, including *CTGF*, *CYR61*, *FOXM1*, *THBS1*, *ITGAV* and *ARHGAP29*, upon

ZBTB11/PRRG2 knockdown. The knockdown efficiency of ZBTB11 and PRRG2 has been validated (**Revised Manuscript Fig. 6f**). As shown in **Revised Manuscript Extended Data Fig. 7b**, knockdown of ZBTB11 decreased the expression of *CTGF*, *FOXM1* and *THBS1*, while the expression of *CYR61*, *ITGAV* and *ARHGAP29* showed no obvious changes upon ZBTB11 knockdown. In addition, the repressive effect on the transcription of *CTGF*, *FOXM1* and *THBS1* by ZBTB11 knockdown can be reversed by concomitant depletion of PRRG2 (**Revised Manuscript Extended Data Fig. 7b**), revealing that PRRG2 is needed for ZBTB11-mediated regulation of these genes. Thus, our data support that the ZBTB11-PRRG2 axis may selectively regulate a subgroup of YAP1 target genes that are functionally related to metastatic regulation.

Next, we followed the reviewer's suggestion to investigate whether the YAP1-S127A mutant is able to reverse the effect of PRRG2 expression or ZBTB11 knockdown on cell migration. To this end, we generated H1299 stable cell lines that express YAP1-S127A or PRRG2 or deplete endogenous ZBTB11, as indicated (**Revised Manuscript Extended Data Fig. 7c and f**). As expected, overexpression of PRRG2 or knockdown of ZBTB11 inhibited cell migration, and more importantly, simultaneous overexpression of the YAP1-S127A mutant in cancer cells resulted in profound resistance to this suppressive effect on cell migration induced by PRRG2 overexpression or ZBTB11 knockdown (**Revised Manuscript Extended Data Fig. 7d-e and g-h**). Thus, our data further support the mechanism by which the ZBTB11-PRRG2-YAP1 axis regulates cancer cell metastatic behaviors.

REVIEWERS' COMMENTS

Reviewer #3 (Remarks to the Author):

The authors have addressed all my previous concerns in this revised version.

Point-by-Point Response

Ref: ZBTB11, a SET-associated Transcription Factor, Triggers Lung Cancer Metastasis (Manuscript #: NCOMMS-22-29298C-Z)

Reviewer #3:

The reviewer commented: “The authors have addressed all my previous concerns in this revised version.”

Response: We are very pleased to know that our new data in the revised version of the manuscript have fully addressed the reviewer’s concerns. We are grateful to the reviewer for his/her time and effort in reviewing our paper and supporting our work.